# Correcting the Mythos of KL-Regularization: Direct Alignment without Overoptimization via $\chi^2$-Preference Optimization

**Audrey Huang**[*]   **Wenhao Zhan**[†]   **Tengyang Xie**[‡]   **Jason D. Lee**[†]
**Wen Sun**[§]   **Akshay Krishnamurthy**[◇]   **Dylan J. Foster**[◇]

[*]University of Illinois Urbana-Champaign   [†]Princeton University
[‡]University of Wisconsin-Madison   [§]Cornell University   [◇]Microsoft Research

## Abstract

Language model alignment methods such as reinforcement learning from human feedback (RLHF) have led to impressive advances in language model capabilities, but are limited by a widely observed phenomenon known as *overoptimization*, where the quality of the language model degrades over the course of the alignment process. As the model optimizes performance on an offline reward model, it overfits to inaccuracies and drifts away from preferred responses covered by the data. To discourage such distribution shift, KL-regularization is widely employed in existing offline alignment methods, but overoptimization continues to harm performance. Lending theoretical insight into the source of these empirical observations, we first show that the KL-regularization is too weak to prevent overfitting, then ask: is it possible to design an efficient algorithm that is provably robust to overoptimization?

In this paper, we advance theoretical understanding of sample-efficient offline alignment and introduce a new algorithm called $\chi^2$-*Preference Optimization* ($\chi$PO). $\chi$PO is a one-line change to Direct Preference Optimization (DPO; Rafailov et al. (2023)), that modifies only the logarithmic link function in the DPO objective. Despite this minimal change, $\chi$PO implicitly implements the principle of *pessimism in the face of uncertainty* via regularization with the $\chi^2$-divergence—which quantifies uncertainty more effectively than KL-regularization—and provably alleviates overoptimization, achieving sample-complexity guarantees based on *single-policy concentrability*, the gold standard in offline reinforcement learning. This guarantee makes $\chi$PO the first simple, yet general-purpose offline alignment algorithm that is provably robust to overoptimization.

## 1 Introduction

Large language models (LLMs) trained on unsupervised text data exhibit impressive and surprising capabilities (Brown et al., 2020; Ouyang et al., 2022; Touvron et al., 2023; OpenAI, 2023; Google, 2023), but can be difficult to control without further guidance. *Reinforcement learning from human feedback (RLHF)* and other alignment methods have emerged as a central tool to align these models to human values and elicit desired behavior (Christiano et al., 2017; Bai et al., 2022; Ouyang et al., 2022; Rafailov et al., 2023). This is achieved by treating the language model as a *policy*, and using techniques from reinforcement learning to optimize for desirable outcomes under a (explicit or implicit) reward model learned from a dataset of human-labeled responses.

Alignment methods like RLHF have led to significant advances in language model capabilities, but existing techniques are limited by a widely observed phenomenon known as *reward overoptimization* or *reward hacking* (Michaud et al., 2020; Tien et al., 2022; Gao et al., 2023; Rafailov et al., 2024a). Since the reward model is an imperfect proxy for human preferences, the true quality of the language model can degrade as training proceeds, even as its performance under the reward model continues to improve. Intuitively, this occurs because the language model may drift away from the manifold covered by the human-labeled data used to train the reward model and end up in a region where the reward model is inaccurate.

Overoptimization is distinct from the classical concept of overfitting because it is a causal or counter-factual phenomenon: When the human-labeled dataset does not cover all possible alternatives, the decision maker—in this case, a language model policy—cannot directly evaluate the effect of their actions. This perspective is supported by the fact that overoptimization can be mitigated by *online* alignment techniques (Guo et al., 2024; Gao et al., 2024; Dong et al., 2024), which exploit interactive access to human or AI feedback to iteratively improve the reward model; unfortunately, gathering such feedback is costly and impractical in many settings. This raises natural theoretical questions regarding the role of overoptimization in *offline alignment*:

- Is overoptimization in offline alignment an *information-theoretic phenomenon*? This would mean that there is simply not enough information in the human-labeled (offline) preference dataset due to partial coverage, and no algorithmic intervention can avoid the overoptimization issue.

- Alternatively, is overoptimization an *algorithmic phenomenon*? This would mean that existing algorithms are not making the most of the data they have (e.g., due to optimizing the wrong objective and converging toward suboptimal solutions) and would suggest that their *sample-efficiency* can be improved, perhaps by taking more aggressive measures to avoid overfitting to the reward model.

Previous developments in the theory of offline reinforcement learning suggest that the answer may be the latter. Indeed, this literature has addressed the challenge of overoptimization—typically referred to as *distribution shift*—through the principle of *pessimism in the face of uncertainty*, which asserts that, given an offline dataset with partial coverage, a decision maker should choose their response according to the most pessimistic view of the world supported by the data. Pessimism encourages the model to avoid overfitting to the offline dataset and is supported by a rich theory offering provable robustness to overoptimization in stylized settings (Liu et al., 2020; Jin et al., 2021).

Perhaps the greatest barrier to implementing pessimism in language models is the efficient quantification of uncertainty in the offline reward, and the distillation of this information into actionable form. Most existing offline alignment methods employ KL-regularization, which penalizes the learned policy for drifting from the reference policy, but this form of uncertainty quantification is insufficient to induce pessimism (Gao et al., 2023) and is provably suboptimal in theory (Zhu et al., 2023; Song et al., 2024, see also Appendix A.1). On the other hand, offline reinforcement learning theory offers abstract pessimistic algorithms that are suitable—at least statistically—for large models (Xie et al., 2021; Uehara and Sun, 2021; Zhan et al., 2022; Chen and Jiang, 2022), but cannot be implemented directly without losing theoretical fidelity or making unrealistic modeling assumptions (Zhu et al., 2023; Zhan et al., 2023a; Li et al., 2023; Xiong et al., 2023; Liu et al., 2024; Cen et al., 2024; Fisch et al., 2024; Ji et al., 2024). Notably, the so-called "DPO+SFT" approach developed by Liu et al. (2024); Cen et al. (2024); Fisch et al. (2024) is provably suboptimal unless the language model satisfies an unrealistic convexity property (Appendix A.1). Thus we ask: *If we instead leverage the unique structure of the language modeling problem, can we develop simple, yet efficient, offline alignment methods that are certifiably robust to overoptimization?*

## 1.1 CONTRIBUTIONS

We introduce a new theoretical algorithm for offline alignment, $\chi^2$-*Preference Optimization* ($\chi$PO). $\chi$PO is simple and straightforward to implement, requiring only a single-line change to Direct Preference Optimization (Rafailov et al. (2023)), yet it is provably robust to overoptimization. Algorithmically, $\chi$PO only differs from DPO in that we replace the usual logarithmic link function in the DPO objective with a new link function that implicitly implements pessimism via regularization with the $\chi^2$-divergence—a divergence that (i) plays a fundamental role in statistics due to its ability to quantify uncertainty (Tsybakov, 2008); and (ii) penalizes off-manifold behavior more effectively than KL-regularization. Statistically, we formalize robustness to overoptimization via a sample complexity guarantee based on *single-policy concentrability*—the gold standard in offline reinforcement learning—which we establish under minimal statistical and function approximation assumptions. This result implies that, in contrast to most prior work, $\chi$PO enjoys meaningful guarantees even when the reference policy has poor coverage. Summarizing:

> $\chi$PO *is the first simple, yet general-purpose algorithm for offline alignment*
> *with provable robustness to overoptimization.*

The result above concerns the classical language model alignment formulation, which assumes the Bradley-Terry preference model (Christiano et al., 2017; Bai et al., 2022; Ouyang et al., 2022;

Rafailov et al., 2023). Turning our attention to general preference models (Munos et al., 2023; Swamy et al., 2024; Rosset et al., 2024) where the goal is to find an approximate Nash equilibrium, we show (Appendix D) that *achieving guarantees based on single-policy concentrability is impossible*. Nonetheless, we show that an iterative variant of $\chi$PO based on self-play achieves a sample complexity guarantee that scales with a new local coverage condition —a condition that is stronger than single policy concentrability, but much weaker than global concentrability and the notion of unilateral concentrability introduced by Cui and Du (2022). This result provides additional evidence for the value of regularization with $\chi^2$-divergence for obtaining sharp sample complexity guarantees.

**Technical highlights.** Our analysis of $\chi$PO leverages several new techniques. First, we show that RLHF with $\chi^2$-regularization is sufficient to achieve guarantees based on single-policy concentrability (Section 3.1 and Appendix C). Next, we show that a variant of the DPO reparameterization trick that combines $\chi^2$-regularization with KL-regularization ("mixed" $\chi^2$-regularization) can be used to reformulate our objective into a purely policy-based objective, in spite of the fact that $\chi^2$-regularization fails to satisfy certain regularity conditions found in prior work (Wang et al., 2023a). Finally, and perhaps most importantly, we use a novel analysis to show that pessimism is preserved after reparameterization. Compared to prior approaches to pessimism in offline RL (Xie et al., 2021; Uehara and Sun, 2021; Zhan et al., 2022; Chen and Jiang, 2022), $\chi^2$-regularization strikes a useful balance between generality and tractability, and we expect our techniques to find broader use.

## 2  BACKGROUND

In this section, we provide necessary background and highlight that standard algorithms in offline alignment suffer from overoptimization. We adopt standard big-oh notation, and write $f = \widetilde{O}(g)$ to denote that $f = O(g \cdot \max\{1, \operatorname{polylog}(g)\})$ and $a \lesssim b$ as shorthand for $a = O(b)$.

### 2.1  ALIGNMENT FROM HUMAN FEEDBACK

Following prior work (e.g., Rafailov et al. (2023); Ye et al. (2024)), we adopt a contextual bandit formulation of the alignment problem. We formalize the language model as a *policy* $\pi : \mathcal{X} \to \Delta(\mathcal{A})$ which maps a context (prompt) $x \in \mathcal{X}$ to an action (response) $a \in \mathcal{A}$ via $a \sim \pi(\cdot \mid x)$, and let $\rho \in \Delta(\mathcal{X})$ denote the distribution over contexts/prompts.

**Offline alignment.** In the offline alignment problem (Christiano et al., 2017; Bai et al., 2022; Ouyang et al., 2022), we assume access to a dataset $\mathcal{D}_{\mathsf{pref}} = \{(x, a_+, a_-)\}$ of $n$ prompts and labeled response pairs generated from a reference policy (language model) $\pi_{\mathsf{ref}}$, which is typically obtained through SFT. Here, $a_+$ is a positive action/response and $a_-$ is a negative action/response. Given the context/prompt $x \sim \rho$, the pair $(a_+, a_-)$ is generated by sampling a pair $(a, b)$ as $a \sim \pi_{\mathsf{ref}}(\cdot \mid x)$ and $b \sim \pi_{\mathsf{ref}}(\cdot \mid x)$, and then ordering them as $(a_+, a_-)$ based on a binary preference $y \sim \mathbb{P}(a \succ b \mid x)$. We assume that preferences follow the *Bradley-Terry* model (Bradley and Terry, 1952):

$$\mathbb{P}(a \succ b \mid x) = \frac{\exp(r^\star(x,a))}{\exp(r^\star(x,a)) + \exp(r^\star(x,b))}, \tag{1}$$

for an unknown reward function $r^\star : \mathcal{X} \times \mathcal{A} \to [0, R_{\mathsf{max}}]$ for some $R_{\mathsf{max}} \geq 1$. From the preference dataset $\mathcal{D}_{\mathsf{pref}}$, we aim to learn a policy $\widehat{\pi}$ that has high reward in the sense that $J(\pi^\star) - J(\widehat{\pi}) \leq \varepsilon$ for a small $\varepsilon > 0$, where $J(\pi) := \mathbb{E}_{x \sim \rho, a \sim \pi(\cdot \mid x)}[r^\star(x, a)]$ is the true expected reward, and $\pi^\star$ is any comparator policy of interest. We abbreviate $\mathbb{E}_\pi[\cdot] := \mathbb{E}_{x \sim \rho, a \sim \pi(\cdot \mid x)}[\cdot]$, and assume that $\rho(x) > 0$ for all $x$ and $\pi_{\mathsf{ref}}(a \mid x) > 0$ for all $x, a$ without loss of generality.

**Offline RLHF with KL-regularization.** Classical algorithms for offline alignment (Christiano et al., 2017; Ouyang et al., 2022) are based on reinforcement learning with a *KL-regularized* reward objective, defined for a regularization parameter $\beta > 0$, via

$$J_\beta^{\mathsf{KL}}(\pi) := J(\pi) - \beta \cdot D_{\mathsf{KL}}(\pi \,\|\, \pi_{\mathsf{ref}}) = \mathbb{E}_\pi \left[ r^\star(x, a) - \beta \log \frac{\pi(a \mid x)}{\pi_{\mathsf{ref}}(a \mid x)} \right], \tag{2}$$

where we adopt the shorthand $D_{\mathsf{KL}}(\pi \,\|\, \pi') = \mathbb{E}_{x \sim \rho}[D_{\mathsf{KL}}(\pi(\cdot \mid x) \,\|\, \pi'(\cdot \mid x))]$. These methods first estimate a reward function $\widehat{r}$ from $\mathcal{D}_{\mathsf{pref}}$ using maximum likelihood under the Bradley-Terry model:

$$\widehat{r} = \operatorname*{argmax}_{r \in \mathcal{R}} \sum_{(x, a_+, a_-) \in \mathcal{D}_{\mathsf{pref}}} \log \sigma(r(a_+ \mid x) - r(a_- \mid x)), \tag{3}$$

where $\sigma(x) := \frac{\exp(x)}{1 + \exp(x)}$ is the sigmoid function and $\mathcal{R}$ is a class of reward functions, which is typically parameterized by a neural network. Then, they apply standard policy optimization methods

like PPO to optimize an estimated version of Eq. (2): $\widehat{\pi} = \arg\max_{\pi \in \Pi} \mathbb{E}_\pi \left[ \widehat{r}(x, a) - \beta \log \frac{\pi(a|x)}{\pi_{\text{ref}}(a|x)} \right]$. The regularization term in Eq. (2) is intended to encourage $\widehat{\pi}$ to stay close to $\pi_{\text{ref}}$, with the hope of preventing the policy from overfitting to the potentially inaccurate reward model $\widehat{r}$.

**Direct preference optimization (DPO).** $\chi$PO is based on an alternative offline alignment approach, Direct Preference Optimization (DPO; Rafailov et al. (2023)). DPO uses the closed-form solution of the optimal KL-regularized policy under the objective Eq. (2)—which can be viewed as implicitly modeling rewards—to define a single policy optimization objective that removes the need for direct reward function estimation. Given a user specified policy class $\Pi$, DPO solves

$$\widehat{\pi}_{\text{DPO}} = \arg\max_{\pi \in \Pi} \sum_{(x, a_+, a_-) \in \mathcal{D}_{\text{pref}}} \log \left[ \sigma \left( \beta \log \frac{\pi(a_+|x)}{\pi_{\text{ref}}(a_+|x)} - \beta \log \frac{\pi(a_-|x)}{\pi_{\text{ref}}(a_-|x)} \right) \right], \quad (4)$$

with the convention that the value of the objective is $-\infty$ if $\pi$ does not satisfy $\pi \ll \pi_{\text{ref}}$.

## 2.2 Overoptimization and Insufficiency of KL-Regularization

Empirically, both classical RLHF and direct alignment methods like DPO have been observed to suffer from overoptimization (Gao et al., 2023; Guo et al., 2024; Rafailov et al., 2024a; Song et al., 2024), wherein model quality degrades during the optimization process as the learned policy drifts away from $\pi_{\text{ref}}$. This can be mitigated by *online alignment* techniques (Gao et al., 2024; Guo et al., 2024; Dong et al., 2024; Xie et al., 2024), which collect labeled preference data on-policy during training, but there are many settings where this is impractical or infeasible. As we will see, the overoptimization phenomena in offline alignment methods is an issue of sample-inefficiency, which can be understood through the lens of coverage coefficients developed in the theory of offline reinforcement learning (Liu et al., 2020; Jin et al., 2021; Rashidinejad et al., 2021). In particular, the performance of existing offline alignment algorithms depends on how well data covers all candidate policies, and degrades when coverage is inadequate or the number of samples is insufficiently large.

**Coverage coefficients.** In offline reinforcement learning theory, the sample efficiency of an algorithm refers to the number of samples required to guarantee that $J(\widehat{\pi}) \approx J(\pi^\star)$. It is typically quantified by a *coverage coefficient* (or concentrability coefficient) that measures the quality of the data collected by the reference $\pi_{\text{ref}}$ (Farahmand et al., 2010; Xie and Jiang, 2020; Zanette et al., 2021). We will utilize the $L_1$ coverage coefficient, defined for a policy $\pi$ as $\mathcal{C}^\pi := \mathbb{E}_\pi \left[ \frac{\pi(a|x)}{\pi_{\text{ref}}(a|x)} \right]$. *Single policy concentrability* is the gold standard for sample efficiency, and is obtained by an algorithm if, *for any comparator policy $\pi^\star$*, the sample size required to learn $J(\widehat{\pi}) \approx J(\pi^\star)$ scales with $\mathcal{C}^{\pi^\star}$, the coverage coefficient of $\pi^\star$. This guarantees that $\widehat{\pi}$ is competitive with the best policy that is sufficiently covered by offline data, and, importantly, also guarantees that $\widehat{\pi}$ is never much worse than $\pi_{\text{ref}}$ itself. Single policy concentrability is typically achieved by pessimistic algorithms that penalize the evaluations of candidate policies according to their uncertainty under the offline data, which prevents the learner from overfitting to inaccurate offline reward models.

In contrast, the performance of non-pessimistic algorithms typically scales with *all-policy concentrability*—meaning that sample complexity scales with $\max_{\pi \in \Pi} \mathcal{C}^\pi$ (Liu et al., 2020; Jin et al., 2021; Rashidinejad et al., 2021)— which is a guarantee achieved by even greedy algorithms that directly optimize the offline reward model without regularization. All-policy concentrability describes algorithms that require the data itself to be rich enough to prevent overfitting; as such, we will use it to identify methods that are prone to overoptimization. Single policy concentrability then serves as a theoretical certification that an algorithm is robust to poor data coverage and will not overfit.

**Pessimism in offline alignment.** Zhu et al. (2023) show that the performance of PPO and DPO scales with all-policy concentrability, $\max_\pi \mathcal{C}^\pi_\infty$, for the stylized case of alignment with linearly parameterized policies where $\pi_\theta(a \mid x) \propto \exp(\langle \phi(x, a), \theta \rangle)$ for a known feature embedding $\phi(x, a) \in \mathbb{R}^d$ (see also Zhu et al. (2024); Song et al. (2024)). They also propose a pessimistic algorithm that achieves single policy concentrability, or $J(\pi^\star) - J(\widehat{\pi}) \lesssim \sqrt{\frac{\text{poly}(\mathcal{C}^{\pi^\star}_\infty, d)}{n}}$ simultaneously for all $\pi^\star$. While encouraging, these results are restricted to linearly parameterized policies, and cannot be directly applied to large language models. Most existing theoretical algorithms for offline alignment are similar in nature, and either place restrictive assumptions on the policy class $\Pi$ (Zhu et al., 2023; Zhan et al., 2023a; Li et al., 2023; Xiong et al., 2023) or are not feasible to implement in a way that is faithful to theory (Ye et al., 2024; Ji et al., 2024).

Most relevant to our work, a series of recent papers (Liu et al., 2024; Cen et al., 2024; Fisch et al., 2024) propose implementing pessimism for general policy classes $\Pi$ by solving the "DPO+SFT" objective

$$\operatorname*{argmax}_{\pi \in \Pi} \left\{ \alpha \cdot \mathbb{E}_{\pi_{\mathsf{ref}}}[\beta \log \pi(a \mid x)] + \frac{1}{n} \sum_{(x, a_+, a_-) \in \mathcal{D}_{\mathsf{pref}}} \log \left[ \sigma \left( \beta \log \frac{\pi(a_+|x)}{\pi_{\mathsf{ref}}(a_+|x)} - \beta \log \frac{\pi(a_-|x)}{\pi_{\mathsf{ref}}(a_-|x)} \right) \right] \right\}, \quad (5)$$

which augments the DPO objective (the second term) with an additional supervised fine-tuning-like (SFT) loss (the first term). While this objective is simple to apply to general policy classes, the existing single-policy concentrability guarantees for this method assume that $\Pi$ satisfies restrictive *convexity* conditions which do not hold in practice for large language models. Perhaps surprisingly, we show (Appendix A.1) that without convexity, *the objective in Eq. (5) fails to achieve a single-policy concentrability guarantee.*[1] In other words, DPO+SFT is insufficient to mitigate overoptimization.

## 3  $\chi^2$-PREFERENCE OPTIMIZATION

This section presents our main algorithm, $\chi$PO. We begin by introducing $\chi^2$-regularization as a general framework for mitigating overoptimization in offline alignment (Section 3.1), then derive the $\chi$PO algorithm (Section 3.2) and finally present our main theoretical guarantee (Section 3.3).

### 3.1  FRAMEWORK: $\chi^2$-REGULARIZED REWARD OPTIMIZATION

The central algorithm design principle for our work is to (implicitly or explicitly) optimize a variant of the classical RLHF objective (Eq. (2)) that replaces KL-regularization with regularization via $\chi^2$-divergence, defined for a pair of probability measures $\mathbb{P}$ and $\mathbb{Q}$ with $\mathbb{P} \ll \mathbb{Q}$ via $D_{\chi^2}(\mathbb{P} \parallel \mathbb{Q}) := \frac{1}{2} \int \left( \frac{d\mathbb{P}}{d\mathbb{Q}} - 1 \right)^2 d\mathbb{Q}$. $\chi^2$-divergence is a more aggressive form of regularization than KL-divergence; we have $D_{\mathsf{KL}}(\mathbb{P} \parallel \mathbb{Q}) \leq 2 D_{\chi^2}(\mathbb{P} \parallel \mathbb{Q})$, but the converse is not true in general. We consider the following $\chi^2$-regularized RL objective:[2]

$$J_\beta^\chi(\pi) := \mathbb{E}_\pi[r^\star(x, a)] - \beta \cdot D_{\chi^2}(\pi \parallel \pi_{\mathsf{ref}}), \qquad D_{\chi^2}(\pi \parallel \pi_{\mathsf{ref}}) := \mathbb{E}_\pi \left[ \frac{\pi(a|x)}{\pi_{\mathsf{ref}}(a|x)} \right]. \quad (6)$$

Moving to a form of regularization that penalizes deviations from $\pi_{\mathsf{ref}}$ more forcefully than KL-regularization is a natural approach to mitigating overoptimization, but an immediate concern is that this may lead to overly conservative algorithms. As we will show, however, $\chi^2$-divergence is better suited to the geometry of offline alignment, as it has the unique property (not shared by KL-divergence) that its value quantifies the extent to which the accuracy of a reward model $\widehat{r}$ trained under $\pi_{\mathsf{ref}}$ will transfer to a downstream policy $\pi$ of interest (Lemma H.3). This implies that the $\chi^2$-regularized RL objective in Eq. (6) meaningfully implements a form of pessimism in the face of uncertainty, and by tuning the regularization parameter $\beta > 0$, we can keep the learned policy $\widehat{\pi}$ close to $\pi_{\mathsf{ref}}$ in the "right" (uncertainty-aware) way. As such, we view optimizing $\chi^2$-regularized rewards, i.e., $\operatorname*{argmax}_{\pi \in \Pi} J_\beta^\chi(\pi)$ as a general principle to guide algorithm design for offline alignment (as well as offline RL more broadly), which we expect to find broader use.

We now turn our attention to the matter of how to optimize this objective. One natural approach, in the vein of classical RLHF (Christiano et al., 2017; Ouyang et al., 2022), is to estimate a reward model $\widehat{r}$ using maximum likelihood (Eq. (3)), and then use PPO or other policy optimization methods to solve

$$\widehat{\pi} = \operatorname*{argmax}_{\pi \in \Pi} \mathbb{E}_\pi \left[\widehat{r}(x, a)\right] - \beta \cdot D_{\chi^2}(\pi \parallel \pi_{\mathsf{ref}}) = \operatorname*{argmax}_{\pi \in \Pi} \mathbb{E}_\pi \left[ \widehat{r}(x, a) - \beta \frac{\pi(a|x)}{\pi_{\mathsf{ref}}(a|x)} \right]. \quad (7)$$

While this indeed leads to strong statistical guarantees (cf. Appendix C), we adopt a simpler and more direct approach inspired by DPO, which removes the need for a separate reward estimation step.

### 3.2  THE $\chi$PO ALGORITHM

Our main algorithm, $\chi$PO, is described in Algorithm 1. Given a preference dataset $\mathcal{D}_{\mathsf{pref}}$ and policy class $\Pi$, the algorithm learns a policy $\widehat{\pi}$ by solving the DPO-like optimization objective Eq. (9), which replaces the usual $\log \frac{\pi(a|x)}{\pi_{\mathsf{ref}}(a|x)}$ terms in the original DPO objective (Eq. (4)) with a new link function:

$$\phi\left( \frac{\pi(a|x)}{\pi_{\mathsf{ref}}(a|x)} \right) = \frac{\pi(a|x)}{\pi_{\mathsf{ref}}(a|x)} + \log \left( \frac{\pi(a|x)}{\pi_{\mathsf{ref}}(a|x)} \right).$$

---

[1]This finding is surprising because Xie et al. (2024) show that an *optimistic online* counterpart to Eq. (5), which negates the SFT term, enjoys online RLHF guarantees without requiring analogous convexity conditions.

[2]Note the definition of $D_{\chi^2}(\pi \parallel \pi_{\mathsf{ref}})$ differs from $\mathbb{E}[D_{\chi^2}(\pi(\cdot \mid x) \parallel \pi_{\mathsf{ref}}(\cdot \mid x))]$ only by a constant scaling and shift, both of which are inconsequential when used as regularization in an optimization objective.

---

**Algorithm 1** $\chi^2$-Preference Optimization ($\chi$PO)

---

**input:** Reference policy $\pi_{\mathsf{ref}}$, preference dataset $\mathcal{D}_{\mathsf{pref}}$, $\chi^2$-regularization coefficient $\beta > 0$.

1: Define

$$\phi(z) := z + \log z. \tag{8}$$

2: Optimize $\chi^2$-regularized preference optimization objective:

$$\widehat{\pi} \leftarrow \operatorname*{argmax}_{\pi \in \Pi} \sum_{(x, a_+, a_-) \in \mathcal{D}_{\mathsf{pref}}} \log \left[ \sigma \left( \mathsf{clip}_{2R_{\mathsf{max}}} \left[ \beta \phi \left( \frac{\pi(a_+ \mid x)}{\pi_{\mathsf{ref}}(a_+ \mid x)} \right) - \beta \phi \left( \frac{\pi(a_- \mid x)}{\pi_{\mathsf{ref}}(a_- \mid x)} \right) \right] \right) \right]. \tag{9}$$

3: **return:** $\widehat{\pi}$.

---

A secondary modification is that we handle potentially unbounded density ratios by clipping to the interval $[-2R_{\mathsf{max}}, +2R_{\mathsf{max}}]$ via the operator $\mathsf{clip}_R(z) = \max\{\min\{R, z\}, -R\}$. In what follows, we will show that this simple modification to DPO—that is, incorporating an additional density ratio term outside the logarithm—implicitly implements pessimism via $\chi^2$-regularization.

**Algorithm derivation.** Recall that DPO is derived (Rafailov et al., 2023) by observing that the optimal KL-regularized policy $\pi^\star_{\beta;\mathsf{KL}} := \operatorname{argmax}_\pi \{\mathbb{E}_\pi[r^\star(x, a)] - \beta D_{\mathsf{KL}}(\pi \| \pi_{\mathsf{ref}})\}$ satisfies $r^\star(x, a) = \beta \log \frac{\pi^\star_{\beta;\mathsf{KL}}(a|x)}{\pi_{\mathsf{ref}}(a|x)} + Z_{\beta, r^\star;\mathsf{KL}}(x)$ for all $x \in \mathcal{X}$ and $a \in \mathcal{A}$ where $Z_{\beta, r^\star;\mathsf{KL}}(x)$ is a normalization constant that depends on $x$ but not $a$. This facilitates reparameterizing the reward model in the maximum likelihood estimation objective (Eq. (3)) in terms of a learned policy, yielding the DPO objective in Eq. (4).

To apply a similar reparameterization trick for $\chi^2$-divergence, a natural starting point is an observation from Wang et al. (2023a), who show that an analogous characterization for the optimal regularized policy holds for a general class of $f$-*divergences*. For a convex function $f : \mathbb{R}_+ \to \mathbb{R}$, define the induced $f$-divergence by $D_f(\mathbb{P} \| \mathbb{Q}) = \int f\left(\frac{d\mathbb{P}}{d\mathbb{Q}}\right) d\mathbb{Q} = \mathbb{E}_\mathbb{Q}\left[f\left(\frac{d\mathbb{P}}{d\mathbb{Q}}\right)\right]$. Wang et al. (2023a) show that for any differentiable $f$ that satisfies the technical condition $0 \notin \operatorname{dom}(f')$, the optimal $f$-regularized policy $\pi^\star_{\beta;f} = \operatorname{argmax}_\pi \{\mathbb{E}_\pi[r^\star(x, a)] - \beta D_f(\pi \| \pi_{\mathsf{ref}})\}$ satisfies

$$r^\star(x, a) = \beta f'\left(\frac{\pi^\star_{\beta;f}(a|x)}{\pi_{\mathsf{ref}}(a|x)}\right) + Z_{\beta, r^\star;f}(x) \tag{10}$$

for a normalization constant $Z_{\beta, r^\star;f}(x)$, allowing for a similar reparameterization. Informally, the condition $0 \notin \operatorname{dom}(f')$ means that $D_f(\cdot \| \pi_{\mathsf{ref}})$ acts as a *barrier* for the positive orthant, automatically forcing $\pi^\star_{\beta;f}$ to place positive probability mass on any action $a$ for which $\pi_{\mathsf{ref}}(a \mid x) > 0$.

The $\chi^2$-divergence is an $f$-divergence corresponding to $f(z) = \frac{1}{2}(z-1)^2$, but unfortunately does not satisfy the condition $0 \notin \operatorname{dom}(f')$, making Eq. (10) inapplicable. Indeed, the optimal $\chi^2$-regularized policy can clip action probabilities to zero in a non-smooth fashion even when $\pi_{\mathsf{ref}}(a \mid x) > 0$, which means that the identity Eq. (10) does not apply. To address this issue, we augment $\chi^2$-regularization by considering the *mixed $\chi^2$-divergence* given by $f_{\chi_{\mathsf{mix}}}(z) := \frac{1}{2}(z-1)^2 + z \log z$, which has

$$D_{f_{\chi_{\mathsf{mix}}}}(\mathbb{P} \| \mathbb{Q}) = D_{\chi^2}(\mathbb{P} \| \mathbb{Q}) + D_{\mathsf{KL}}(\mathbb{P} \| \mathbb{Q}).$$

In other words, we use *both $\chi^2$-regularization and KL-regularization*; $\chi^2$-regularization enforces pessimism, while KL-regularization enforces the barrier property and facilitates reparameterization. Indeed, the link function $\phi$ (Eq. (8)) used in $\chi$PO has $\phi(z) := f'_{\chi_{\mathsf{mix}}}(z) = z + \log z$, which satisfies $0 \notin \operatorname{dom}(f'_{\chi_{\mathsf{mix}}})$, so Eq. (10) yields the reparameterization $r^\star(x, a) = \beta \phi\left(\frac{\pi^\star_{\beta;f_{\chi_{\mathsf{mix}}}}(a|x)}{\pi_{\mathsf{ref}}(a|x)}\right) + Z_{\beta, r^\star;f_{\chi_{\mathsf{mix}}}}(x)$. Substituting this identity into the maximum likelihood estimation objective (Eq. (3)) yields the $\chi$PO algorithm.

Going forward, we define $J^{\chi_{\mathsf{mix}}}_{\beta, r}(\pi) = \mathbb{E}_\pi[r(x, a)] - \beta \cdot D_{\chi^2}(\pi \| \pi_{\mathsf{ref}}) - \beta \cdot D_{\mathsf{KL}}(\pi \| \pi_{\mathsf{ref}})$ for a reward function $r$. We use the shorthand $\pi^\star_\beta = \operatorname{argmax}_\pi J^{\chi_{\mathsf{mix}}}_{\beta, r^\star}(\pi)$ as the optimal policy under mixed $\chi^2$-regularization, and abbreviate $Z_{\beta, r}(x) := Z_{\beta, r; f_{\chi_{\mathsf{mix}}}}(x)$, so that

$$r^\star(x, a) = \beta \phi\left(\frac{\pi^\star_\beta(a|x)}{\pi_{\mathsf{ref}}(a|x)}\right) + Z_{\beta, r^\star}(x). \tag{11}$$

### 3.3 THEORETICAL GUARANTEES

To state our main sample complexity guarantee for $\chi$PO, we begin by making standard statistical assumptions. Let the regularization parameter $\beta > 0$ in $\chi$PO be fixed. We first make a *realizability* assumption, which states that the policy class $\Pi$ used in $\chi$PO is sufficiently expressive to represent the optimal policy under mixed $\chi^2$-regularization (Eq. (11)); recall that in the context of language modeling, $\Pi$ represents a class of language models with fixed architecture and varying weights.

**Assumption 3.1** (Policy realizability)**.** *The policy class $\Pi$ satisfies $\pi_\beta^\star \in \Pi$, where $\pi_\beta^\star$ is the optimal policy under mixed $\chi^2$-regularization (Eq. (11)).*

Policy realizability is a standard assumption for sample-efficient reinforcement learning (Agarwal et al., 2019; Lattimore and Szepesvári, 2020; Foster and Rakhlin, 2023), and is equivalent to reward model realizability in our setting via reparameterization. Next, our second assumption asserts that the implicit reward models induced by the policy class $\Pi$ in $\chi$PO have bounded range.

**Assumption 3.2** (Bounded implicit rewards)**.** *For a parameter $V_{\max} \geq R_{\max}$, it holds that for all $\pi \in \Pi$, $x \in \mathcal{X}$, and $a, b \in \mathcal{A}$, $\left| \beta\phi\left(\frac{\pi(a|x)}{\pi_{\mathsf{ref}}(a|x)}\right) - \beta\phi\left(\frac{\pi(b|x)}{\pi_{\mathsf{ref}}(b|x)}\right) \right| \leq V_{\max}$.*

In practice, $V_{\max}$ can be measured and directly controlled (e.g., via clipping), and our guarantees scale polynomially in this parameter. Assumption 3.2 generalizes analogous assumptions from analyses of DPO-like methods (Rosset et al., 2024; Xie et al., 2024); see Appendix B.4 for detailed comparison.

**Example 3.1** (Policy classes induced by reward models)**.** A natural setting in which both Assumption 3.1 and Assumption 3.2 hold is when the policy class $\Pi$ is induced by a class of bounded reward function $\mathcal{R} \subset (\mathcal{X} \times \mathcal{A} \to [0, R_{\max}])$ through the mixed-$\chi^2$ parameterization, for $\beta > 0$:

$$\Pi_{\mathcal{R},\beta} := \left\{ \pi(a \mid x) = \pi_{\mathsf{ref}}(a \mid x) \cdot \phi^{-1}(\beta^{-1}(r(x,a) - Z_{\beta,r}(x))) \mid r \in \mathcal{R} \right\}. \tag{12}$$

Here, Assumption 3.1 holds whenever $r^\star \in \mathcal{R}$, and Assumption 3.2 holds with $V_{\max} \leq 2R_{\max}$. ◁

Finally, recall the definition of the $L_1$ concentrability coefficient, $\mathcal{C}^\pi := \mathbb{E}_\pi\left[\frac{\pi(a|x)}{\pi_{\mathsf{ref}}(a|x)}\right]$, which is equivalent to the $\chi^2$-divergence up to a constant shift, i.e., $\mathcal{C}^\pi = 1 + 2D_{\chi^2}(\pi \parallel \pi_{\mathsf{ref}})$. We use $L_1$ concentrability to quantify how well the offline preference dataset $\mathcal{D}_{\mathsf{pref}}$, generated by $\pi_{\mathsf{ref}}$, covers a policy $\pi$, and the following result is our main sample complexity guarantee for $\chi$PO.

**Theorem 3.1** (Sample complexity bound for $\chi$PO)**.** *Suppose Assumptions 3.1 and 3.2 hold for some $\beta > 0$. With probability at least $1 - \delta$, $\chi$PO (Algorithm 1) produces a policy $\widehat{\pi}$ such that for all policies $\pi^\star$ simultaneously, we have*

$$J(\pi^\star) - J(\widehat{\pi}) \lesssim V_{\max}e^{2R_{\max}} \cdot \sqrt{\frac{\mathcal{C}^{\pi^\star}\log(|\Pi|/\delta)}{n}} + \beta \cdot \mathcal{C}^{\pi^\star} + \beta^{-1} \cdot \frac{V_{\max}^2 e^{4R_{\max}}\log(|\Pi|/\delta)}{n}. \tag{13}$$

*Given any comparator policy $\pi^\star$, we can choose the regularization parameter $\beta$ to achieve*

$$J(\pi^\star) - J(\widehat{\pi}) \lesssim V_{\max}e^{2R_{\max}} \cdot \sqrt{\frac{\mathcal{C}^{\pi^\star}\log(|\Pi|/\delta)}{n}}. \tag{14}$$

Theorem 3.1 shows that $\chi$PO achieves a sample complexity guarantee that scales only with the single-policy concentrability parameter $\mathcal{C}^{\pi^\star}$ for the comparator policy $\pi^\star$, for all policies $\pi^\star$ simultaneously. In particular, roughly $n = O\left(\frac{\mathcal{C}^{\pi^\star}\log(|\Pi|/\delta)}{\varepsilon^2}\right)$ examples are sufficient to learn a policy that is $\varepsilon$-suboptimal relative to $\pi^\star$. As a result, $\chi$PO is robust to overoptimization since the learned policy is as good as any $\pi^\star$ that is sufficiently covered by $\pi_{\mathsf{ref}}$ (in the sense that $\mathcal{C}^{\pi^\star} = O(1)$), which is effectively the best one can hope for in the purely offline setting. In contrast, naive offline alignment methods like DPO have sample complexity that scales with *all-policy concentrability* (roughly, $\max_\pi \mathcal{C}^\pi$), even when the comparator policy $\pi^\star$ is sufficiently covered (Zhu et al., 2023; Song et al., 2024). To highlight this, in Figure 1 (see Appendix B for details) we give a concrete example in which $\chi$PO allows the user to tune $\beta$ to achieve tight statistical rates, yet no choice of $\beta$ for DPO leads to comparable performance. Effectively, any choice of $\beta$ for DPO is either susceptible to overoptimization, or is unacceptably conservative. All prior works that achieve similar sample complexity guarantees based on single-policy concentrability are either impractical, or require more restrictive statistical assumptions on the policy class (Ye et al., 2024; Liu et al., 2024; Cen et al., 2024; Fisch et al., 2024; Ji et al., 2024).

Regarding the parameter $V_{\max}$, we observe that since the policy $\pi_\beta^\star$ satisfies $\left|\beta\phi\left(\frac{\pi_\beta^\star(a|x)}{\pi_{\mathrm{ref}}(a|x)}\right) - \beta\phi\left(\frac{\pi_\beta^\star(b|x)}{\pi_{\mathrm{ref}}(b|x)}\right)\right| \leq 2R_{\max}$, information-theoretically we can always achieve $V_{\max} = 2R_{\max}$ by pre-filtering the policy class $\Pi$ to remove all policies in violation of this inequality. Since this may be non-trivial computationally, we enforce this range via clipping in Eq. (9). Lastly, $\chi^2$-regularized methods that utilize an explicit reward model, such as $\chi^2$-RLHF (Appendix C) or Corollary 3.1, avoid dependence on $V_{\max}$, which we discuss in greater depth in Section 4.3.

**Tuning the regularization parameter.** To achieve optimal dependence on $\mathcal{C}^{\pi^\star}$, Theorem 3.1 requires tuning $\beta > 0$ as a function of this parameter, similar to other pessimistic schemes (Liu et al., 2024). With no prior knowledge, setting $\beta \propto \sqrt{\frac{V_{\max}^2 e^{4R_{\max}}\log(|\Pi|/\delta)}{n}}$ suffices to ensure that, simultaneously for all comparator policies $\pi^\star$, we have $J(\pi^\star) - J(\widehat{\pi}) \lesssim V_{\max}e^{2R_{\max}} \cdot \sqrt{\frac{(\mathcal{C}^{\pi^\star})^2\log(|\mathcal{R}|/\delta)}{n}}$. This guarantee achieves a slightly worse rate than Eq. (14) but holds simultaneously for all comparator policies rather than the specific one that was used to tune $\beta$. The following result, specializing to the setting in Example 3.1, shows that there exists an optimal parameter $\beta^\star > 0$ that recovers the rate in Eq. (14) and holds simultaneously for all comparator policies.

**Corollary 3.1** (Sample complexity bound for $\chi$PO with a reward model). Consider the setting in Example 3.1, where the policy class $\Pi_{\mathcal{R},\beta}$ is the set of mixed $\chi^2$-regularized policies induced by a reward model class $\mathcal{R}$ with $r^\star \in \mathcal{R}$ and $\beta > 0$. For any $\delta \in (0,1)$, there exists a choice[3] for $\beta^\star > 0$ such that with probability at least $1 - \delta$, $\chi$PO (Algorithm 1), with class $\Pi_{\mathcal{R},\beta^\star}$, produces a policy $\widehat{\pi}$ such that for all policies $\pi^\star$ simultaneously, we have $J(\pi^\star) - J(\widehat{\pi}) \lesssim R_{\max}e^{2R_{\max}} \cdot \sqrt{\frac{\mathcal{C}^{\pi^\star}\log(|\mathcal{R}|/\delta)}{n}}$.

### 3.3.1 EXPERIMENTS IN OFFLINE LANGUAGE MODEL ALIGNMENT

We perform preliminary evaluations of $\chi$PO for offline language model alignment on the TL;DR dataset (Stiennon et al., 2020) using DPO as our baseline; see Appendix E for full results and details. Table 1 displays the final-checkpoint winrates of $\chi$PO and DPO for different regularization parameters $\beta$ and number of training epochs. Smaller $\beta$ and increased epochs reflect the regime where overoptimization is a concern, but more policy improvement is available (existing works treat $\beta = 0.05$ and 1 training epoch as standard choices for DPO (Gao et al., 2024; Guo et al., 2024; Rafailov et al., 2024a)). Over all choices of $\beta$ and epochs, $\chi$PO achieves a higher average winrate than DPO. The performance gap grows as the number of epochs increases and $\beta$ decreases, reflecting the favorable bias-overoptimization tradeoff for $\chi$PO from our theoretical analysis; moreover, $\chi$PO displays robust performance of all parameters while DPO degrades completely for $\beta = 0.005$.

Table 1: Winrate on TL;DR Summarization of $\chi$PO vs. DPO, for several choices of regularization parameter $\beta$ and number of training epochs. Standard error over 3 seeds is also reported.

| $\beta$ | Epochs | $\chi$PO winrate (%) | DPO winrate (%) |
|---|---|---|---|
| | 1 | **56.5 ± 1.3** | 55.8 ± 2.1 |
| 0.05 | 2 | **56.1 ± 0.6** | 50.3 ± 0.8 |
| | 4 | **48.0 ± 1.6** | 38.0 ± 0.7 |
| | 1 | **50.6 ± 1.6** | 14.7 ± 3.9 |
| 0.005 | 2 | **52.8 ± 2.3** | 3.4 ± 1.5 |
| | 4 | **51.6 ± 0.8** | 0.5 ± 0.2 |

## 4 UNDERSTANDING $\chi$PO: THE BIAS-OVEROPTIMIZATION TRADEOFF

Having derived $\chi$PO from the mixed $\chi^2$-regularized RLHF objective and analyzed its performance, we now take a moment to better understand the statistical properties of the policies the algorithm learns. We focus on the tradeoff between overoptimization and bias (i.e., underoptimization) achieved by the regularization parameter $\beta > 0$, highlighting through examples how this leads to statistical benefits over naive alignment methods like DPO. **See Appendix B for full discussion.**

---

[3]It is unclear how to select $\beta^\star$ in a data-driven manner, as it depends on the functionals $\pi \mapsto C^\pi$, $\pi \mapsto J(\pi)$.

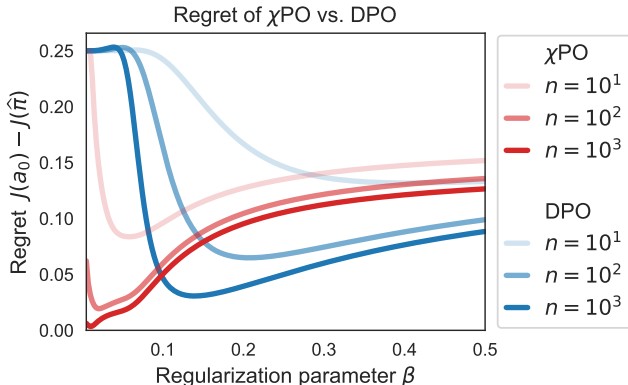

Figure 1: The regret $J(a_0) - J(\widehat{\pi})$ of $\chi$PO and DPO for different values of $n$. For DPO, the error from overoptimization dominates when $\beta \leq (2 \log n)^{-1}$ (as discussed in Appendix B.3), and the error from bias dominates when $\beta > (2 \log n)^{-1}$. Taking the best choice of $\beta$ for each method, DPO converges at an exponentially slower rate $(\frac{1}{\log n})$ than $\chi$PO $(\frac{1}{\sqrt{n}})$; see Proposition A.1 for formal statement and Appendix B.3 for further discussion.

### 4.1 PROPERTIES OF OPTIMAL POLICY UNDER MIXED $\chi^2$-REGULARIZATION

We begin by deriving a (nearly) closed form solution for the optimal mixed $\chi^2$-regularized policy in Eq. (11), which is the $\chi$PO solution in the limit of infinite data. The link function $\phi(\cdot)$ is strictly increasing over $\mathbb{R}_+$, and its inverse is given by $\phi^{-1}(z) = W_0(\exp(z))$, where $W_0(y)$ is the Lambert W-function (Corless et al., 1996) defined as the inverse of $x \mapsto xe^x$ for $y \geq -e^{-1}$. Consequently, for any $x$, the optimal policy under mixed $\chi^2$-regularization satisfies

$$\pi_\beta^\star(a \mid x) = \pi_{\mathsf{ref}}(a \mid x) \cdot W_0\big(\exp\big(\beta^{-1}(r^\star(x, a) - Z_{\beta, r^\star}(x))\big)\big),$$

where $Z_{\beta, r^\star}(x)$ is chosen such that $\sum_a \pi_\beta^\star(a \mid x) = 1$.

Compared to KL-regularization, which leads to softmax policies that satisfy $\pi_{\beta;\mathsf{KL}}^\star(a \mid x) = \pi_{\mathsf{ref}}(a \mid x) \cdot \exp\big(\beta^{-1}(r^\star(x, a) - Z_{\beta, r^\star;\mathsf{KL}}(x))\big)$, the inverse link function $\phi^{-1}(z) = W_0(\exp(z))$ for mixed $\chi^2$-regularization satisfies $\phi^{-1}(z) \approx z$ for $z \geq 1$, leading to a more heavy-tailed action distribution for $\pi_\beta^\star$. On the other hand, for $z \leq 1$ the inverse link behaves like the exponential function (i.e., $\phi^{-1}(z) \approx e^z$ for $z \leq 1$); see Figure 2 for an illustration, and Proposition B.1 for a formal statement. Using these properties, we derive the following upper and lower bounds on the density ratio between $\pi_\beta^\star$ and $\pi_{\mathsf{ref}}$.

**Proposition 4.1.** *For all $x \in \mathcal{X}, a \in \mathcal{A}$, the optimal policy $\pi_\beta^\star$ under mixed $\chi^2$-regularization satisfies*

$$\exp\Big(-\frac{R_{\max}}{\beta}\Big) \lesssim \frac{\pi_\beta^\star(a|x)}{\pi_{\mathsf{ref}}(a|x)} \lesssim 1 + \frac{R_{\max}}{\beta}. \tag{15}$$

The upper bound in Eq. (15), which arises from the $\chi^2$ term in the mixed-$\chi^2$ objective, scales inversely with the regularization parameter $\beta$, and reflects the heavy-tailed, pessimistic behavior this regularizer induces; in contrast, the optimal policy under pure KL-regularization only satisfies $\exp\Big(-\frac{R_{\max}}{\beta}\Big) \lesssim \frac{\pi_{\beta;\mathsf{KL}}^\star(a|x)}{\pi_{\mathsf{ref}}(a|x)} \lesssim \exp\Big(\frac{R_{\max}}{\beta}\Big)$ in general. The lower bound in Eq. (15) arises from the KL term in the mixed-$\chi^2$ objective, but is not important for our analysis (outside of allowing DPO-like reparameterization).

### 4.2 THE BIAS-OVEROPTIMIZATION TRADEOFF

We are now well equipped to understand how $\chi$PO modulates the tradeoff between overoptimization and bias using the regularization parameter $\beta$, and how this tradeoff compares to vanilla DPO. To showcase this, we take a reward modeling perspective, and consider the setting in which the policy class $\Pi$ is induced by a given reward model class $\mathcal{R}$, similar to Example 3.1.

Suppose we start with a reward model class $\mathcal{R}$ such that $r^\star \in \mathcal{R}$. If we use the induced policy class

$$\Pi_{\mathsf{DPO}, \beta} := \big\{\pi(a \mid x) = \pi_{\mathsf{ref}}(a \mid x) \cdot \exp(\beta^{-1}(r(x, a) - Z_{\beta, r;\mathsf{KL}}(x))) \mid r \in \mathcal{R}\big\}, \tag{16}$$

then DPO can be viewed as first fitting a reward model $\widehat{r}$ (Eq. (3)), then outputting the policy $\widehat{\pi}_{\mathsf{DPO}}(a \mid x) = \pi_{\mathsf{ref}}(a \mid x) \cdot \exp(\beta^{-1}(\widehat{r}(x, a) - Z_{\beta, \widehat{r};\mathsf{KL}}(x)))$. Meanwhile, if we use the induced policy class

$$\Pi_{\chi\mathsf{PO}, \beta} := \big\{\pi(a \mid x) = \pi_{\mathsf{ref}}(a \mid x) \cdot \phi^{-1}(\beta^{-1}(r(x, a) - Z_{\beta, r}(x))) \mid r \in \mathcal{R}\big\}, \tag{17}$$

then $\chi$PO can be interpreted as fitting a reward model $\widehat{r}$ with the exact same maximum likelihood objective, but instead outputting the policy $\widehat{\pi}_{\chi\mathsf{PO}}(a \mid x) = \pi_{\mathsf{ref}}(a \mid x) \cdot \phi^{-1}(\beta^{-1}(\widehat{r}(x, a) - Z_{\beta, \widehat{r}}(x)))$.

The policies $\widehat{\pi}_{\chi\mathsf{PO}}$ and $\widehat{\pi}_{\mathsf{DPO}}$ are induced by the same reward model $\widehat{r}$ and parameter $\beta$, but exhibit different bias-overoptimization tradeoffs. For both, large $\beta$ means the policy avoids overfitting to

errors in the reward model (e.g., when $\beta \to \infty$ both policies become $\pi_{\text{ref}}$), while small $\beta$ means the policy has low *bias*, i.e., low error in when the model is correct and $\widehat{r} = r^\star$ (e.g. when $\beta \to 0$, both policies become $x \mapsto \operatorname{argmax}_{a:\pi_{\text{ref}}(a|x)>0} \widehat{r}(x,a)$). Yet, for the same choice of $\beta$, $\widehat{\pi}_{\chi\text{PO}}$ is significantly more heavy-tailed than $\widehat{\pi}_{\text{DPO}}$, a consequence of the pessimism induced by $\chi^2$-regularization; see Figure 3, which plots the action distribution for both policies as a function of $\beta$.

**An illustrative example.** Building on the intuition above, Figure 1 gives a construction in which $\chi\text{PO}$ achieves $\frac{1}{\sqrt{n}}$ regret with an appropriate choice for $\beta$, yet DPO suffers an exponentially worse rate of $\frac{1}{\log n}$ regardless of $\beta$. Intuitively, DPO overfits severely when $\beta$ is small, but suffers high bias when $\beta$ is larger. $\chi\text{PO}$, however, strikes a better tradeoff because small values of $\beta$ are sufficient to prevent overoptimization, which means the policy is also less biased. The "DPO+SFT" algorithm of Liu et al. (2024); Cen et al. (2024); Fisch et al. (2024) also fails in this construction (see Appendix A.1).

### 4.3 Nontriviality and Role of $V_{\text{max}}$ Parameter

We close this section by discussing the role of the $V_{\text{max}}$ parameter (Assumption 3.2) used in the analysis of $\chi\text{PO}$ (Theorem 3.1), motivating it using the induced policy class $\Pi_{\chi\text{PO},\beta}$ from Section 4.2.

Assumption 3.2 implies that all policies $\pi \in \Pi$ satisfy $\left\| \frac{\pi}{\pi_{\text{ref}}} \right\|_\infty \lesssim \frac{V_{\text{max}}}{\beta}$, i.e., that *all-policy $L_\infty$-concentrability* with $\max_{\pi \in \Pi} \mathcal{C}_\infty^\pi \lesssim \frac{V_{\text{max}}}{\beta}$ holds. This might seem to trivialize the offline alignment problem, since such a policy class would enable plug-in regret bounds for even greedy algorithms. We will show that this is not the case, because the $\frac{V_{\text{max}}}{\beta}$ bound is uniquely induced by $\chi^2$-regularization.

Recall that $\chi\text{PO}$ requires the realizability assumption that $\pi_\beta^\star \in \Pi$ (Assumption 3.1), where $\pi_\beta^\star$ is the optimal $\chi^2$-regularized policy that satisfies $r^\star(x,a) = \beta\phi\left(\frac{\pi_\beta^\star(a|x)}{\pi_{\text{ref}}(a|x)}\right) + Z_{\beta,r^\star}(x)$. From Proposition B.2 we have $\left\| \frac{\pi_\beta^\star}{\pi_{\text{ref}}} \right\|_\infty \lesssim \frac{R_{\text{max}}}{\beta}$, so from a statistical perspective, we can take Assumption 3.2 to hold w.l.o.g. by removing any policy that violates this bound. Further, as highlighted in Example 3.1, if we begin from a class of bounded reward models $\mathcal{R} \ni r^\star$, Assumption 3.2 holds with $V_{\text{max}} \lesssim R_{\text{max}}$ for the induced class $\Pi_{\chi\text{PO},\beta}$ defined in Eq. (17), even though knowledge of such a reward model class is a mild statistical assumption that clearly does not trivialize the learning problem.

On the other hand, for DPO, a minimal assumption is that $\pi_{\beta;\text{KL}}^\star \in \Pi$ (Xie et al., 2024), where $\pi_{\beta;\text{KL}}^\star$ is the optimal KL-regularized policy that satisfies $r^\star(x,a) = \beta \log \frac{\pi_{\beta;\text{KL}}^\star(a|x)}{\pi_{\text{ref}}(a|x)} + Z_{\beta,r^\star;\text{KL}}(x)$. Unlike the optimal mixed $\chi^2$-regularized policy, $\pi_{\beta;\text{KL}}^\star$ has $\frac{\pi_{\beta;\text{KL}}^\star(a|x)}{\pi_{\text{ref}}(a|x)} \gtrsim \exp\left(\frac{R_{\text{max}}}{\beta}\right)$. This means that it is impossible to find a policy class that simultaneously (a) realizes $\pi_{\beta;\text{KL}}^\star$, and (b) satisfies all-policy concentrability with $\max_{\pi \in \Pi} \mathcal{C}_\infty^\pi \ll \exp\left(\frac{R_{\text{max}}}{\beta}\right)$. As the bias of DPO is unacceptably large unless $\beta = \text{poly}(1/n)$ (the "small-$\beta$" regime), this leads to vacuous guarantees.

As a result, our analysis of $\chi\text{PO}$ can be viewed as showing that, for any bounded reward class $\mathcal{R}$, there exists a policy class $\Pi$ (precisely, the class $\Pi_{\chi\text{PO},\beta}$ in Eq. (17)) such that the following properties hold:

1. **Bounded bias.** For all $r \in \mathcal{R}$, there exists $\pi_r \in \Pi$ such that for all $\pi^\star$, $J_r(\pi^\star) - J_r(\pi_r) \lesssim \beta \cdot \mathcal{C}^{\pi^\star}$.

2. **Bounded overoptimization.** For all $\pi \in \Pi$, $\left\| \frac{\pi}{\pi_{\text{ref}}} \right\|_\infty \lesssim \frac{R_{\text{max}}}{\beta}$.

We view this as an interesting and non-trivial contribution in its own right.

## 5 Discussion

Our work gives the first general-purpose algorithm for offline alignment with provable robustness to overoptimization, and sample complexity guarantees based on single-policy concentrability. Our analysis and algorithm design techiques offer an example of fruitful interplay between RL theory and language modeling, and we expect they will find broader use. Natural technical directions raised by our paper include (i) understanding the tightest sample complexity guarantees for offline alignment with *general preference models*; (ii) extending our techniques to reinforcement learning settings beyond offline alignment (e.g., general MDPs). We look forward to studying these questions in future work.

**Additional results.** *Results deferred to the appendix for space include (i) Guarantees for RLHF with $\chi^2$-regularization (Appendix C), (ii) Guarantees for general preference models (Appendix D), and (iii) Experiments in language models demonstrating that $\chi\text{PO}$ mitigates overoptimization (Appendix E).*

ACKNOWLEDGEMENTS

We thank Qinghua Liu, Zhaolin Gao, and Yuda Song for several helpful discussions. WS acknowledges funding support from NSF IIS-2154711, NSF CAREER 2339395, DARPA LANCER: LeArning Network CybERagents.

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

# Contents of Appendix

# Part I

# Additional Results

## A  ADDITIONAL RELATED WORK

**Theoretical algorithms for offline alignment.**  Much of prior theoretical work on offline alignment considers algorithms that are tailored to linearly parameterized policies (Zhu et al., 2023; Li et al., 2023; Xiong et al., 2023), while others are not efficiently implementable, e.g., as they require solving min-max problems over a version space (Zhan et al., 2023a). For general policy classes, Ye et al. (2024) provide an algorithm that achieves sample complexity guarantees based on single-policy

concentrability, but the algorithm requires computation of an uncertainty bonus which cannot be implemented faithfully for large language models. Ji et al. (2024) provide an algorithm that achieves single-policy concentrability using self-play, but their approach requires the non-standard realizability assumption that for all $\pi \in \Pi$, there exists $\pi' \in \Pi$ such that $r(x, a) = \beta \log \frac{\pi(a|x)}{\pi'(a|x)} - Z_{\pi,\pi'}(x)$ for some function $Z_{\pi,\pi'}(x)$ that depends on $x$, but not the action $a$. In addition, their algorithm is iterative, and requires solving a DPO-like objective many times (roughly $1/\varepsilon^2$ iterations are required to achieve accuracy $\varepsilon$). Most relevant to our work, Liu et al. (2024); Cen et al. (2024); Fisch et al. (2024) propose solving the appealingly simple DPO + SFT objective in Eq. (5). As we discuss in detail in Appendix A.1, this objective fails to achieve single-policy concentrability unless non-standard convexity assumptions on the policy class or reward model class hold.

A number of other works consider the *hybrid* setting for alignment where—in addition to offline preference data from $\pi_{\text{ref}}$, the algorithm has access to online feedback (Xiong et al., 2023; Gao et al., 2024; Chang et al., 2024; Song et al., 2024). While it is straightforward to achieve guarantees based on single-policy concentrability in this setting, this is a stronger feedback model than what we consider, and is not always realistic. Our work is also complementary to fully online alignment, which dispenses with coverage conditions entirely but requires active exploration (Xu et al., 2020; Novoseller et al., 2020; Pacchiano et al., 2021; Wu and Sun, 2023; Zhan et al., 2023b; Chen et al., 2022; Wang et al., 2023b; Du et al., 2024; Das et al., 2024; Ye et al., 2024; Xie et al., 2024; Cen et al., 2024).

**Generalizations of DPO.**  Wang et al. (2023a) provide a generalization of the DPO reparameterization trick which supports general $f$-divergences that satisfy certain regularity conditions. Their work does not provide sample complexity guarantees or theoretical guidance on which choices of $f$-divergence are preferable, but our main algorithm χPO, can be derived as a special case of their technique with a novel choice of $f$-divergence. Tang et al. (2024) also provide a general framework for deriving DPO variants with general loss functions, but our algorithm does not appear to be a special case of their framework.

**Offline reinforcement learning theory.**  The theory of *offline reinforcement learning* addresses challenges similar to overoptimization, which is typically describes through the language of *distribution shift*. Many of these works, using pessimism and related algorithmic techniques, provide guarantees that are robust to partial coverage of the data collection policy $\pi_{\text{ref}}$, which is reflected in sample complexity guarantees based on single-policy concentrability and similar coverage conditions. While this line of work provides efficient algorithms for simple (e.g., tabular or linear) settings (Liu et al., 2020; Jin et al., 2021; Rashidinejad et al., 2021), existing approaches that support general function approximation (Xie et al., 2021; Uehara and Sun, 2021; Zhan et al., 2022; Chen and Jiang, 2022) cannot be implemented efficiently for language models without non-trivial modifications. See also closely related research on policy optimization and evaluation in statistics and econometrics (Athey and Wager, 2021; Chernozhukov et al., 2019; Kallus and Uehara, 2020).

**$\chi^2$-divergence in reinforcement learning.**  Our work contributes to a growing body of research that uses $\chi^2$-divergence to derive reinforcement learning algorithms with novel statistical guarantees.[4] Notably, our work is inspired by Wang et al. (2024) (see also Gabbianelli et al. (2024)), who use a regularizer similar to $\chi^2$-divergence to derive single-policy concentrability guarantees for contextual bandits. Compared to the $\chi^2$-regularizer $\mathcal{C}^\pi = \mathbb{E}_\pi \left[ \frac{\pi(a|x)}{\pi_{\text{ref}}(a|x)} \right]$ we use, their regularizer takes the form $\mathbb{E}_\pi \left[ \frac{1}{\pi_{\text{ref}}(a|x)} \right]$, which is always larger. As a result of this diference, their regularizer is not suitable for large action spaces. By addressing this shortcoming, we expect our $\chi^2$-regularization approach to find further use in offline RL.

Other related works include (i) Duan et al. (2020) show that $\chi^2$-divergence plays a fundamental role in offline RL with linear function approximation; (ii) Zhan et al. (2022) use $\chi^2$-regularization to provide guarantees based on single-policy concentrability for an offline RL method based on weight function learning; and (iii) Amortila et al. (2024) provide online RL algorithms that explore by directly minimizing an exploration objective based on $\chi^2$-divergence. We mention in passing that a number of recent empirical works apply $\chi^2$-regularization (Zhu et al., 2020; Lee et al., 2021; Ma et al.,

---

[4]More classically, $\chi^2$-divergence is known to play a fundamental role in asymptotic statistics (Tsybakov, 2008; Duchi and Namkoong, 2019).

2022a;b; Zhu and Zhang, 2024) to reinforcement learning in embodied domains. Lastly, Cesa-Bianchi et al. (2017) prove lower bounds against the softmax policy distribution, but in the context of online exploration for online RL. While this is different problem setting than ours, their construction may be in similar in spirit to our lower bound against KL-regularization in offline reinforcement learning (Proposition A.1).

**Empirical research on offline alignment.**   Our work uses DPO (Rafailov et al., 2023) as a starting point. Many prior works have built upon DPO with the aim of addressing specific shortcomings, including Liu et al. (2023); Tang et al. (2024); Azar et al. (2024); Rosset et al. (2024); Chen et al. (2024); Wu et al. (2024); Tajwar et al. (2024). Closely related, there is a large body of research that attempts to understand and mitigate overoptimization in offline alignment from a purely empirical perspective (Michaud et al., 2020; Tien et al., 2022; Coste et al., 2023; Dong et al., 2023; Eisenstein et al., 2023; Gao et al., 2023; Moskovitz et al., 2023; Pal et al., 2024; Rita et al., 2024; Rafailov et al., 2024a; Zhang et al., 2024).

### A.1   DETAILED COMPARISON TO DPO + SFT

In this section, we give additional background on the suboptimality of the DPO + SFT objective in Eq. (5). Let $\beta > 0$ be the KL-regularization parameter and $\alpha > 0$ be an optimism parameter. Consider the setting in which $\Pi = \left\{ \pi_r(a \mid x) = \pi_{\mathsf{ref}}(a \mid x) \exp(\beta^{-1}(r(x,a) - Z_r(x))) \mid r \in \mathcal{R} \right\}$ for a reward class $\mathcal{R} \subset (\mathcal{X} \times \mathcal{A} \to \mathbb{R})$. Liu et al. (2024); Cen et al. (2024); Fisch et al. (2024) propose solving (variants of) the objective

$$\widehat{\pi}_{\mathsf{max\text{-}min}} = \operatorname*{argmax}_{\pi} \min_{r \in \mathcal{R}} \left\{ \alpha \big( \mathbb{E}_{x \sim \rho, a \sim \pi(\cdot|x), b \sim \pi_{\mathsf{ref}}(\cdot|x)}[r(a) - r(b)] - \beta D_{\mathsf{KL}}(\pi \,\|\, \pi_{\mathsf{ref}}) \big) + \mathcal{L}(r) \right\},$$
(18)

where the max ranges over the space of all policies, and where $\mathcal{L}(r) := -\frac{1}{n} \sum_{(x,a_+,a_-) \in \mathcal{D}_{\mathsf{pref}}} \log \sigma[r(x, a_+) - r(x, a_-)]$ is the negative log-likelihood under the Bradley-Terry model. Liu et al. (2024) show that for general policy classes, this algorithm attains sample complexity guarantees scaling with single-policy concentrability; Cen et al. (2024) provide similar results for the special case of linearly parameterized policies.

The objective in Eq. (18) is non-trivial to implement for language models. To derive the DPO + SFT objective in Eq. (5), Liu et al. (2024) observe that if $\mathcal{R}$ is convex, the minimax theorem implies that the objective value in Eq. (18) is equivalent to the value for the min-max objective

$$\min_{r \in \mathcal{R}} \max_{\pi} \left\{ \alpha \big( \mathbb{E}_{x \sim \rho, a \sim \pi(\cdot|x), b \sim \pi_{\mathsf{ref}}(\cdot|x)}[r(a) - r(b)] - \beta D_{\mathsf{KL}}(\pi \,\|\, \pi_{\mathsf{ref}}) \big) + \mathcal{L}(r) \right\}.$$
(19)

This leads to a natural algorithmic strategy adopted by (Liu et al., 2024; Cen et al., 2024; Fisch et al., 2024): Let $\widehat{r}_{\mathsf{min\text{-}max}}$ be the minimizing reward function in Eq. (19) and let $\pi_{\widehat{r}_{\mathsf{min\text{-}max}}}$—the optimal policy in the KL-regularized MDP with reward function $\widehat{r}_{\mathsf{min\text{-}max}}$—be the final policy returned by the algorithm. After standard manipulations, one can then show that $\pi_{\widehat{r}_{\mathsf{min\text{-}max}}}$ is equivalent to

$$\operatorname*{argmax}_{\pi \in \Pi} \left\{ \alpha \cdot \mathbb{E}_{\pi_{\mathsf{ref}}}[\beta \log \pi(a \mid x)] + \frac{1}{n} \sum_{(x,a_+,a_-) \in \mathcal{D}_{\mathsf{pref}}} \log \left[ \sigma \left( \beta \log \frac{\pi(a_+ \mid x)}{\pi_{\mathsf{ref}}(a_+ \mid x)} - \beta \log \frac{\pi(a_- \mid x)}{\pi_{\mathsf{ref}}(a_- \mid x)} \right) \right] \right\}.$$
(20)

We call this policy $\widehat{\pi}_{\mathsf{DPO+SFT}}$. The sample complexity analyses for the $\widehat{\pi}_{\mathsf{DPO+SFT}}$ policy (Eq. (20)) in (Liu et al., 2024; Cen et al., 2024) rely on showing that the objective value in Eq. (19) is equivalent to the value in Eq. (18), which is not guaranteed to hold if $\mathcal{R}$ is non-convex (e.g., if $\mathcal{R}$ is a class of neural networks).[5] Indeed, the following proposition shows that, for non-convex reward classes $\mathcal{R}$, the DPO + SFT objective in Eq. (20) fails to achieve a statistical guarantee based on single-policy concentrability, even when Eq. (18) succeeds.

**Proposition A.1.** *Let $n \in \mathbb{N}$ with $n \geq 2$ be given. There exists a reward class $\mathcal{R}$ with $|\mathcal{R}| = 2$, a problem instance $(\rho, r)$ satisfying realizability ($r \in \mathcal{R}$) and $r \in [0, 1]$, a data collection policy $\pi_{\mathsf{ref}}$, and universal constants $c_1 \in (0, 1)$ and $c_2, c_3 > 0$ such that the following hold:*

---

[5]Precisely, Liu et al. (2024) provide guarantees for $\widehat{\pi}_{\mathsf{max\text{-}min}}$ with general reward class $\mathcal{R}$ and establish equivalence of $\widehat{\pi}_{\mathsf{max\text{-}min}}$ and $\widehat{\pi}_{\mathsf{min\text{-}max}}$ when $\mathcal{R}$ is convex, while Cen et al. (2024) consider linear function approximation, which yields the required convexity.

1. *There exists a policy $\widetilde{\pi}$ such that $\|\widetilde{\pi}/\pi_{\text{ref}}\|_\infty \leq 2$; yet*

2. *For any $\beta \leq (2\log(n))^{-1}$ and $\alpha \geq 0$, the minimax policy $\widehat{\pi}_{\text{min-max}}$ (Eq. (19)) and DPO+SFT policy $\widehat{\pi}_{\text{DPO+SFT}}$ (Eq. (20)) derived from a dataset $\mathcal{D}_{\text{pref}}$ of $n$ samples from $\pi_{\text{ref}}$ incur suboptimality*

$$J(\widetilde{\pi}) - J(\widehat{\pi}_{\text{DPO+SFT}}) = J(\widetilde{\pi}) - J(\widehat{\pi}_{\text{min-max}}) \geq c_2,$$

*with probability at least $c_1$.*

3. *For any $\beta \geq (2\log(n))^{-1}$ and $\alpha \geq 0$, the minimax policy $\widehat{\pi}_{\text{min-max}}$ (Eq. (19)) and DPO+SFT policy $\widehat{\pi}_{\text{DPO+SFT}}$ (Eq. (20)) derived from a dataset $\mathcal{D}_{\text{pref}}$ of $n$ samples from $\pi_{\text{ref}}$ incur suboptimality*

$$J(\widetilde{\pi}) - J(\widehat{\pi}_{\text{DPO+SFT}}) = J(\widetilde{\pi}) - J(\widehat{\pi}_{\text{min-max}}) \geq \frac{c_3}{\log(n)},$$

*with probability at least $c_1$.*

On the other hand, we observe that for the instance in Proposition A.1, $\chi$PO (via Theorem 3.1) with $\beta \propto 1/\sqrt{n}$ and the class $\Pi = \{\pi(a \mid x) = \pi_{\text{ref}}(a \mid x) \cdot \phi^{-1}(\beta^{-1}(r(x,a) - Z_r(x))) \mid r \in \mathcal{R}\}$ achieves

$$J(\widetilde{\pi}) - J(\widehat{\pi}) \lesssim \sqrt{\frac{(\mathcal{C}^{\widetilde{\pi}})^2}{n}} \lesssim \sqrt{\frac{1}{n}},$$

highlighting the fact that $\chi$PO meaningfully adapts to single-policy concentrability even when the technical conditions required by DPO+SFT do not hold; see also Appendix B. We find this conclusion to be somewhat surprising, as Xie et al. (2024) show that an *optimistic* counterpart to Eq. (20), which negates the SFT term, enjoys strong guarantees for online alignment with general policy classes without requiring convexity.

Although our construction does not establish inconsistency in the $\beta \geq (2\log(n))^{-1}$ regime, in general, DPO+SFT will incur $O(\beta)$ bias if one aims to compete with the optimal policy. Due to restriction that $\beta$ must be rather large, this results in an exponentially slower rate of convergence than $\chi$PO.

**Proof of Proposition A.1.** Let $n \in \mathbb{N}$ with $n \geq 2$ be given. We consider a problem instance with $\mathcal{X} = \{x_1, x_2\}$ and $\mathcal{A} = \{a_0, a_1, a_2, a_3\}$, so that $|\mathcal{A}| = 4$. We define a reward class with two reward functions $\mathcal{R} := \{r_1, r_2\}$ as follows. For $i \in \{1, 2\}$:

$$r_i(x_1, a_0) = \zeta, \quad \text{and} \quad r_i(x_1, a_1) = r_i(x_1, a_2) = r_i(x_1, a_3) = 0$$
$$r_i(x_2, a_0) = 1/2, \quad r_i(x_2, a_i) = 1, \quad \text{and} \quad r_i(x_2, a_j) = 0 \ \forall j \neq i.$$

Here $\zeta \in [0, 1]$ will be chosen at the end of the proof. The context distribution is $\rho = \text{unif}(\mathcal{X})$, and we define $\pi_{\text{ref}}$ for each $x_i \in \{x_1, x_2\}$ via

$$\pi_{\text{ref}}(a_0 \mid x_i) = 1/2, \quad \pi_{\text{ref}}(a_1 \mid x_i) = \pi_{\text{ref}}(a_2 \mid x_i) = 1/(2n), \quad \text{and} \quad \pi_{\text{ref}}(a_3 \mid x_i) = (n-2)/(2n).$$

Let $r_1$ be the true reward function. Recall that $\mathcal{D}_{\text{pref}} = \{(x, a_+, a_-)\}$ consists of $n$ tuples $(x, a_+, a_-)$ obtained by sampling $x \sim \rho$ and a pair of actions $(a, b) \sim \pi_{\text{ref}}$ and labeling them as $(a_+, a_-)$ via the Bradley-Terry model in Eq. (1) with reward $r_1$. Define a "bad" event under this process:

$$\mathcal{E} := \{\text{No tuples in } \mathcal{D}_{\text{pref}} \text{ contain } a_1 \text{ or } a_2\}.$$

We can lower bound the probability of $\mathcal{E}$ as follows:

$$\mathbb{P}[\mathcal{E}^c] \leq \mathbb{P}[a_1 \text{ in } \mathcal{D}_{\text{pref}}] + \mathbb{P}[a_2 \text{ in } \mathcal{D}_{\text{pref}}]$$
$$= 2(1 - (1 - 1/2n)^n) \leq 2(1 - e^{-1/2}(1 - 1/(4n))) \leq 2(1 - 7e^{-1/2}/8) \leq 0.94,$$

where the first inequality uses that $(1 - x/n)^n \geq e^{-x}(1 - x^2/n)$ for $n \geq 1$ and $|x| < n$. We conclude that

$$\mathbb{P}[\mathcal{E}] \geq 0.06 =: c_1.$$

Let $\mathcal{L}(r; \mathcal{D}_{\text{pref}}) := -\frac{1}{n} \sum_{(x, a_+, a_-) \in \mathcal{D}_{\text{pref}}} \log \sigma[r(x, a_+) - r(x, a_-)]$ denote the DPO loss. Observe that conditioned on $\mathcal{E}$, we have that $\mathcal{L}(r_1; \mathcal{D}_{\text{pref}}) = \mathcal{L}(r_2; \mathcal{D}_{\text{pref}})$. Noting that

$$\max_\pi \{\mathbb{E}_\pi[r] - \mathbb{E}_{\pi_{\text{ref}}}[r] - \beta D_{\text{KL}}(\pi \parallel \pi_{\text{ref}})\} = \mathbb{E}_{\pi_r}[r] - \mathbb{E}_{\pi_{\text{ref}}}[r] - \beta D_{\text{KL}}(\pi_r \parallel \pi_{\text{ref}}),$$

is the same for both $r \in \mathcal{R}$, we see that both $r_1$ and $r_2$ optimize the minimax objective in Eq. (19). Thus, breaking ties adversarially, we can choose $\widehat{\pi}_{\text{min-max}} = \pi_{r_2}$ under $\mathcal{E}$ for all values of $\beta > 0$ and $\alpha \geq 0$. By the equivalence between the minimax objective in Eq. (19) and the DPO+SFT objective in Eq. (20) (Liu et al., 2024; Cen et al., 2024; Fisch et al., 2024), for $\Pi = \{\pi_{r_1}, \pi_{r_2}\}$, we can choose $\widehat{\pi}_{\text{DPO+SFT}} = \pi_{r_2}$ in Eq. (20) under $\mathcal{E}$. Indeed, under $\mathcal{E}$, the DPO+SFT objective is equivalent to $\arg\max_{\pi \in \Pi} \mathbb{E}_{\pi_{\text{ref}}}[\log \pi(a)]$, and $\pi_{r_1}$ and $\pi_{r_2}$ have the same value for this objective.

To conclude we choose $\widetilde{\pi}(\cdot) = a_0$, which has $\|\widetilde{\pi}/\pi_{\text{ref}}\|_\infty = 2$. It remains to calculate the suboptimality gap.
$$J(\widetilde{\pi}) - J(\widehat{\pi}_{\text{DPO+SFT}}) = J(\widetilde{\pi}) - J(\widehat{\pi}_{\text{min-max}}) = J(\widetilde{\pi}) - J(\pi_{r_2})$$
under $\mathcal{E}$. Note that $J(\widetilde{\pi}) = \zeta/2 + 1/4$. We decompose the reward for $\pi_{r_2}$ on instance $r_1$ into two components, corresponding to the two contexts $x_1, x_2$:

$$J(\pi_{r_2}) = \frac{1}{2}\left(\mathbb{E}_{a \sim \pi_{r_2}}[r_1(x_1, a)] + \mathbb{E}_{a \sim \pi_{r_2}}[r_1(x_2, a)]\right) =: \frac{1}{2}(J_1(\beta) + J_2(\beta))$$

$$J_1(\beta) = \frac{r_1(x_1, a_0)\pi_{\text{ref}}(a_0 \mid x_1)\exp(r_2(x_1, a_0)/\beta)}{Z(r_2, x_1)} = \frac{\zeta/2\exp(\zeta/\beta)}{1/2\exp(\zeta/\beta) + 1/2}$$

$$J_2(\beta) = \frac{r_1(x_2, a_0)\pi_{\text{ref}}(a_0 \mid x_2)\exp(r_2(x_2, a_0)/\beta) + r_1(x_1, a_1)\pi_{\text{ref}}(a_1 \mid x_2)\exp(r_2(x_2, a_1)/\beta))}{Z(r_2, x_2)}$$

$$= \frac{1/4e^{1/2\beta} + 1/(2n)}{1/2e^{1/2\beta} + e^{1/\beta}/(2n) + (n-1)/(2n)},$$

where $Z(r_2, x) := \sum_{a \in \mathcal{A}} \pi_{\text{ref}}(a \mid x)\exp(r_2(x, a)/\beta)$.

We first consider the small $\beta$ regime. Here we use the upper bound $J_1(\beta) \leq \zeta$ and focus on $J_2(\beta)$. Note that $J_2(\beta)$ is increasing with $\beta$ for $\beta \leq 1/(2\log(n))$. In particular, if we consider $\beta = 1/(c\log(n))$ for $c \geq 2$, then the expression above is equal to

$$J_2(\beta) = \frac{n^{c/2}/4 + 1/(2n)}{n^{c/2}/2 + n^{c-1}/2 + (n-1)/(2n)} \leq \frac{n^{c/2}/4 + 1/(2n)}{n^{c/2} + (n-1)/(2n)} \leq 1/4 + \frac{1}{2n^{c/2+1}} \leq 3/8,$$

where the last inequality holds when $c \geq 2$ and $n \geq 2$. We set $c = 2$, so that as long as $n \geq 2$, $J(\pi_{r_2}) \leq \frac{3}{8}$. Thus, the suboptimality is

$$J(\widetilde{\pi}) - J(\pi_{r_2}) \geq \frac{\zeta}{2} + \frac{1}{4} - \left(\frac{\zeta}{2} + \frac{3}{16}\right) \geq \frac{1}{16} =: c_2.$$

Next consider the regime where $\beta \geq 1/(2\log(n))$. Analogously to before, note that $J_2(\beta) \leq 1/2$. On the other hand, $J_1(\beta)$ is monotonically decreasing with $\beta$, so using $\beta \geq 1/(2\log(n))$ we obtain the bound

$$J_1(\beta) \leq \frac{\zeta \exp(2\zeta \log(n))}{\exp(2\zeta \log(n)) + 1} = \zeta \cdot \frac{n^{2\zeta}}{n^{2\zeta} + 1}.$$

So in this case, the suboptimality is

$$J(\widetilde{\pi}) - J(\pi_{r_2}) \geq \frac{\zeta}{2} \cdot \left(1 - \frac{n^{2\zeta}}{n^{2\zeta} + 1}\right) \geq \frac{\zeta}{4} \cdot \frac{1}{n^{2\zeta}} = \frac{\log(2)}{16\log(n)},$$

if we set $\zeta = \log(2)/(2\log(n))$ which is in $[0, 1]$ under the assumption that $n \geq 2$. $\qquad\square$

## B DETAILED DISCUSSION: $\chi$PO AND THE BIAS-OVEROPTIMIZATION TRADEOFF

Having derived $\chi$PO from the mixed $\chi^2$-regularized RLHF objective and analyzed its performance, we now take a moment to better understand the statistical properties of the policies the algorithm learns. We focus on the tradeoff between overoptimization and bias (i.e., underoptimization) achieved by the regularization parameter $\beta > 0$, highlighting through examples how this leads to statistical benefits over naive alignment methods like DPO.

## B.1 Properties of Optimal Policy under Mixed $\chi^2$-Regularization

We begin by deriving a (nearly) closed form solution for the optimal mixed $\chi^2$-regularized policy in Eq. (11); recall that we expect $\chi$PO to converge to this policy in the limit of infinite data.

We first observe that the link function $\phi(\cdot)$ is strictly increasing over $\mathbb{R}_+$, and its inverse is given by $\phi^{-1}(z) = W_0(\exp(z))$; here, $W_0(y)$ denotes the Lambert W-function (Corless et al., 1996), defined for $y \geq -e^{-1}$ as the inverse of the function $x \mapsto xe^x$. Consequently, for any $x$, the optimal policy under mixed $\chi^2$-regularization satisfies

$$\pi_\beta^\star(a \mid x) = \pi_{\mathsf{ref}}(a \mid x) \cdot W_0\big(\exp\big(\beta^{-1}(r^\star(x,a) - Z_{\beta,r^\star}(x))\big)\big),$$

where $Z_{\beta,r^\star}(x)$ is chosen such that $\sum_a \pi_\beta^\star(a \mid x) = 1$. We can better understand how this policy behaves using the following simple upper and lower bounds on the inverse link function $\phi^{-1}(z) = W_0(\exp(z))$.

**Proposition B.1.** *The link function $\phi(z) = z + \log z$ is strictly increasing over $(0, \infty)$, and its inverse $\phi^{-1}(z) = W_0(\exp(z))$ is strictly increasing over $(-\infty, \infty)$. The inverse link function $\phi^{-1}$ satisfies*

$$\frac{z}{2} \leq \phi^{-1}(z) \leq z \quad \forall z \in [1, \infty), \quad \text{and} \quad e^{z-e} \leq \phi^{-1}(z) \leq e^z \quad \forall z \in (-\infty, 1].$$

Compared to KL-regularization, which leads to softmax policies that satisfy $\pi_{\beta;\mathsf{KL}}^\star(a \mid x) = \pi_{\mathsf{ref}}(a \mid x) \cdot \exp\big(\beta^{-1}(r^\star(x,a) - Z_{\beta,r^\star;\mathsf{KL}}(x))\big)$, we see that the inverse link function $\phi^{-1}(z) = W_0(\exp(z))$ for mixed $\chi^2$-regularization satisfies $\phi^{-1}(z) \approx z$ for $z \geq 1$, leading to a more heavy-tailed action distribution for $\pi_\beta^\star$. On the other hand, for $z \leq 1$ the inverse link behaves like the exponential function (i.e., $\phi^{-1}(z) \approx e^z$ for $z \leq 1$); see Figure 2 for an illustration. Using these properties, we can derive the following upper and lower bounds on the density ratio between $\pi_\beta^\star$ and $\pi_{\mathsf{ref}}$.

**Proposition B.2** (Proposition 4.1 restated)**.** *For all $x \in \mathcal{X}$ and $a \in \mathcal{A}$, the optimal policy $\pi_\beta^\star$ under mixed $\chi^2$-regularization satisfies*

$$\exp\left(-\frac{R_{\max}}{\beta}\right) \lesssim \frac{\pi_\beta^\star(a \mid x)}{\pi_{\mathsf{ref}}(a \mid x)} \lesssim 1 + \frac{R_{\max}}{\beta}. \tag{21}$$

*Both inequalities are tight in general (up to absolute constants).*

The upper bound in Eq. (21), which arises from the $\chi^2$ term in the mixed-$\chi^2$ objective, scales inversely with the regularization parameter $\beta$, and reflects the heavy-tailed, pessimistic behavior this regularizer induces; in contrast, the optimal policy under pure KL-regularization only satisfies

$$\exp\left(-\frac{R_{\max}}{\beta}\right) \lesssim \frac{\pi_{\beta;\mathsf{KL}}^\star(a \mid x)}{\pi_{\mathsf{ref}}(a \mid x)} \lesssim \exp\left(\frac{R_{\max}}{\beta}\right) \tag{22}$$

in general. The lower bound in Eq. (21) arises from the KL term in the mixed-$\chi^2$ objective, but is not important for our analysis (outside of allowing for DPO-like reparameterization).

## B.2 The Bias-Overoptimization Tradeoff

We are now well equipped to understand how $\chi$PO modulates the tradeoff between overoptimization and bias using the regularization parameter $\beta$, and how this tradeoff compares to vanilla DPO. To showcase this, we take a reward modeling perspective, and consider the setting in which the policy class $\Pi$ is induced by a given reward model class $\mathcal{R}$, similar to Example 3.1.

Suppose we start with a reward model class $\mathcal{R} \subset (\mathcal{X} \times \mathcal{A} \to [0, R_{\max}])$ such that $r^\star \in \mathcal{R}$. If we use the induced policy class

$$\Pi_{\mathsf{DPO},\beta} := \big\{\pi(a \mid x) = \pi_{\mathsf{ref}}(a \mid x) \cdot \exp(\beta^{-1}(r(x,a) - Z_{\beta,r;\mathsf{KL}}(x))) \mid r \in \mathcal{R}\big\}, \tag{23}$$

then DPO can be interpreted as fitting a reward model $\widehat{r}$ using maximum likelihood (Eq. (3)) and then outputting the policy $\widehat{\pi}_{\mathsf{DPO}}(a \mid x) = \pi_{\mathsf{ref}}(a \mid x) \cdot \exp(\beta^{-1}(\widehat{r}(x,a) - Z_{\beta,\widehat{r};\mathsf{KL}}(x)))$. Meanwhile, if we use the induced policy class

$$\Pi_{\chi\mathsf{PO},\beta} := \big\{\pi(a \mid x) = \pi_{\mathsf{ref}}(a \mid x) \cdot \phi^{-1}(\beta^{-1}(r(x,a) - Z_{\beta,r}(x))) \mid r \in \mathcal{R}\big\}, \tag{24}$$

then $\chi$PO can be interpreted as fitting a reward model $\widehat{r}$ with the exact same maximum likelihood objective, but instead outputting the policy $\widehat{\pi}_{\chi\mathsf{PO}}(a \mid x) = \pi_{\mathsf{ref}}(a \mid x) \cdot \phi^{-1}(\beta^{-1}(\widehat{r}(x,a) - Z_{\beta,\widehat{r}}(x)))$.

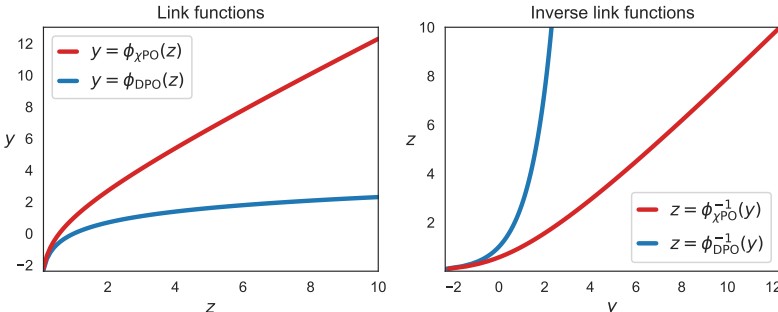

Figure 2: Behavior of the mixed $\chi^2$-regularization link function $\phi_{\chi\text{PO}}(z) = z + \log z$ and inverse $\phi_{\chi\text{PO}}^{-1}(z) = W_0(\exp(z))$, compared to the KL-regularization link function $\phi_{\text{DPO}}(z) = \log z$ and inverse $\phi_{\text{DPO}}^{-1}(z) = \exp(z)$. $\phi_{\chi\text{PO}}^{-1}(z) \approx z$ for $z \geq 1$, leading to favorable heavy-tailed, pessimistic behavior.

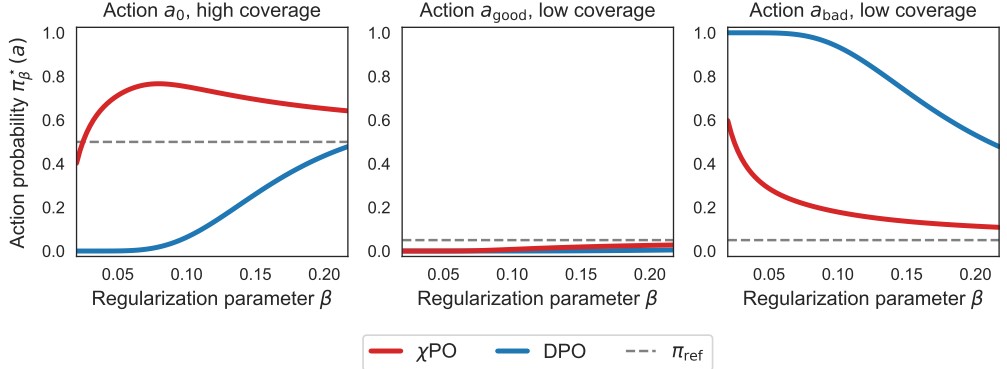

Figure 3: Action probabilities for policies learned by $\chi\text{PO}$ and DPO on the example from Appendix B.3, under the "bad" event $\mathcal{E}$ in which the true reward model is $r^\star = r_1$ but the estimated reward model is $\widehat{r} = r_2$ ($n = 10$). Here, $r^\star(a_{\text{good}}) = 1$ and $r^\star(a_{\text{bad}}) = 0$, but $\widehat{r}(a_{\text{good}}) = 0$ and $\widehat{r}(a_{\text{good}}) = 1$; both reward functions have $r^\star(a_0) = \widehat{r}(a_0) = 1/2$, and the goal is to compete with a comparator policy that deterministically plays $a_0$.

**Overoptimization.** The DPO policy is greedier with respect to the incorrect reward model and places much larger mass on the bad action $a_{\text{bad}}$ for all $\beta \in (0, \frac{1}{2\log n}]$ (Right). As a result, the DPO policy places much smaller mass on the baseline action $a_0$, suffering significantly more overoptimization error compared to $\chi\text{PO}$ (Left; see also Figure 1).

**Bias.** Compared to DPO, $\chi\text{PO}$ has a higher probability of taking both the optimal action $a_{\text{good}}$ and the reference action $a_0$. As a result, it strikes a better bias-overoptimization tradeoff than DPO, and is competitive with respect to the comparator $a_0$ even when DPO fails to converge.

The policies $\widehat{\pi}_{\chi\text{PO}}$ and $\widehat{\pi}_{\text{DPO}}$ are induced by the same reward model $\widehat{r}$, and both use the parameter $\beta$ to balance bias and overoptimization. For both policies, large $\beta$ means the policy avoids overfitting to errors in the reward model (the extreme case is $\beta \to \infty$, in which case both policies become $\pi_{\text{ref}}$), while small $\beta$ means the policy has low *bias*, i.e., low error in the case where the model is correct in the sense that $\widehat{r} = r^\star$ (the extreme case is $\beta \to 0$, in which case both policies become $x \mapsto \arg\max_{a:\pi_{\text{ref}}(a|x)>0} \widehat{r}(x, a)$). Yet, for the same choice of $\beta$, $\widehat{\pi}_{\chi\text{PO}}$ is significantly more heavy-tailed than $\widehat{\pi}_{\text{DPO}}$, a consequence of the pessimism induced by $\chi^2$-regularization; see Figure 3, which plots the action distribution for both policies as a function of $\beta$.

### B.3 AN ILLUSTRATIVE EXAMPLE

We now give a concrete example in which $\chi\text{PO}$ allows the user to tune $\beta$ to achieve tight statistical rates, yet no choice of $\beta$ for DPO leads to comparable performance (effectively, any choice of $\beta$ is

either susceptible to overoptimization, or has unacceptably high bias). This illustrates the favorable tradeoff between bias and overoptimization achieved by $\chi$PO.

Let $n \in \mathbb{N}$ with $n \geq 2$ be given. We consider a problem instance with $\mathcal{X} = \{\varnothing\}$ and $\mathcal{A} = \{a_0, a_1, a_2, a_3\}$. We define $\pi_{\mathsf{ref}}$ via

$$\pi_{\mathsf{ref}}(a_0) = \tfrac{1}{2}, \quad \pi_{\mathsf{ref}}(a_1) = \pi_{\mathsf{ref}}(a_2) = \tfrac{1}{2n}, \quad \text{and} \quad \pi_{\mathsf{ref}}(a_3) = \tfrac{n-2}{2n}.$$

We define a reward class with two reward functions $\mathcal{R} := \{r_1, r_2\}$ as follows. For $i \in \{1, 2\}$:

$$r_i(a_0) = 1/2, \quad r_i(a_i) = 1, \quad r_i(a_j) = 0, \quad \forall j \neq i.$$

Let $\beta > 0$ be fixed. To compare $\chi$PO and DPO, we consider their behavior when invoked with the induced policy classes $\Pi_{\chi\mathsf{PO},\beta}$ and $\Pi_{\mathsf{DPO},\beta}$ defined above. Recall that with this choice, the two algorithms can be interpreted as fitting a reward model $\widehat{r}$ using maximum likelihood (Eq. (3)) and returning the policies $\widehat{\pi}_{\chi\mathsf{PO}}(a \mid x) = \pi_{\mathsf{ref}}(a \mid x) \cdot \phi^{-1}(\beta^{-1}(\widehat{r}(x, a) - Z_{\beta, \widehat{r}}(x)))$ and $\widehat{\pi}_{\mathsf{DPO}}(a \mid x) = \pi_{\mathsf{ref}}(a \mid x) \cdot \exp(\beta^{-1}(\widehat{r}(x, a) - Z_{\beta, \widehat{r}; \mathsf{KL}}(x)))$, respectively.

Suppose that $r_1$ is the true reward function. It is hopeless (information-theoretically) to compete with the unconstrained optimal action $a_1$, as we are in a sample-starved regime where $\mathcal{C}^{a_1} = 2n$ (in the language of Eq. (13)). Indeed, one can show (see proof of Proposition A.1 in Appendix A) that with constant probability, none of the examples in the offline dataset $\mathcal{D}_{\mathsf{pref}}$ contain actions $a_1$ or $a_2$. Under this event, which we denote by $\mathcal{E}$, the value for the maximum likelihood objective in Eq. (3) is identical for $r_1$ and $r_2$, so we may obtain $\widehat{r} = r_2$ (due to adversarial tie-breaking). However, in spite of the fact that the policies $\widehat{\pi}_{\chi\mathsf{PO}}$ and $\widehat{\pi}_{\mathsf{DPO}}$ are induced by the same (incorrect) reward function $\widehat{r} = r_2$, they produce very different action distributions, as highlighted in Figure 3.

To understand this, note that even in the sample-starved regime, we can still hope to compete with the "baseline" action $a_0$; Figure 1 shows that $\chi$PO has low regret against this action, while DPO has high regret. In particular, since $\mathcal{C}^{a_0} = 2$, Theorem 3.1 (Eq. (13)) implies that $\chi$PO achieves

$$J(a_0) - J(\widehat{\pi}_{\chi\mathsf{PO}}) \lesssim \sqrt{\frac{1}{n}} + \beta + \beta^{-1}\frac{1}{n},$$

and setting $\beta \propto \sqrt{\frac{1}{n}}$ leads to $J(a_0) - J(\widehat{\pi}_{\chi\mathsf{PO}}) \lesssim \sqrt{\frac{1}{n}}$. This is a consequence of the pessimistic, heavy-tailed nature of $\widehat{\pi}_{\chi\mathsf{PO}}$ (cf. Proposition B.2), which places no more than $\beta^{-1}/n$ probability mass on the (incorrect) greedy action $a_2$ for $\widehat{r} = r_2$, thereby correctly capturing the inherent uncertainty in the reward for this action.

On the other hand, it is straightforward to show that for all possible values $\beta \leq (2 \log n)^{-1}$, the DPO policy $\widehat{\pi}_{\mathsf{DPO}}$ has regret

$$J(a_0) - J(\widehat{\pi}_{\mathsf{DPO}}) \geq \frac{1}{2}\left(1 - \frac{1}{1 + \frac{1}{n}e^{\frac{1}{2}} + (1 - \frac{1}{n})e^{-\frac{1}{2\beta}}}\right) - \frac{1}{2n} \geq \Omega(1)$$

whenever $n \geq 2$. This is because when $\beta \leq (2 \log n)^{-1}$, $\widehat{\pi}_{\mathsf{DPO}}$ assigns excessively high probability to the incorrect greedy action $a_2$, an instance of overoptimization. Meanwhile, larger choices for $\beta$ lead to excessively large bias in general (see Appendix A.1 for a more sophisticated construction which extends this lower bound to all possible $\beta$). In other words, as illustrated in Figure 1, no choice of $\beta$ gives a favorable tradeoff between overoptimization and bias.

To summarize, for DPO, large values of $\beta$ are required to avoid overfitting to the reward function, incurring high bias. Meanwhile, $\chi$PO avoids overoptimization using comparatively small values for $\beta$, yet has bias no worse than that of DPO, thereby striking a better tradeoff. We mention that the "DPO+SFT" algorithm of Liu et al. (2024); Cen et al. (2024); Fisch et al. (2024) also fails on the construction above; see Proposition A.1 in Appendix A.1 for details.

**Remark B.1** (DPO decreases probabilities of preferred and rejected responses). *Various recent works have noted an empirical phenomenon in which DPO decreases the probabilities for both preferred and rejected responses throughout training (Yuan et al., 2024; Pal et al., 2024; Rafailov et al., 2024b). Interestingly, we observe that the example above exhibits this phenomenon. Notably, if $\beta < (2 \log n)^{-1}$, then under the event $\mathcal{E}$ in which the offline dataset $\mathcal{D}_{\mathsf{pref}}$ does not contain the actions*

$a_1$ or $a_2$ (so that $\widehat{r} = r_2$), we observe that $\widehat{\pi}_{\text{DPO}}(a_0) = \frac{\frac{1}{2}e^{\frac{1}{2\beta}}}{\frac{1}{2}e^{\frac{1}{2\beta}} + \frac{1}{2n}e^{\frac{1}{\beta}} + \frac{n-1}{2n}} < \frac{1}{2} = \pi_{\text{ref}}(a_0)$, and for

all $i > 2$, $\widehat{\pi}_{\text{DPO}}(a_i) = \frac{\frac{1}{2n}}{\frac{1}{2}e^{\frac{1}{2\beta}} + \frac{1}{2n}e^{\frac{1}{\beta}} + \frac{n-1}{2n}} < \frac{1}{2n} = \pi_{\text{ref}}(a_i)$. We conclude that for all $a \in \mathcal{D}_{\text{pref}}$,

$$\widehat{\pi}_{\text{DPO}}(a) < \pi_{\text{ref}}(a).$$

*We emphasize that this behavior arises due to the use of function approximation. When the reward class $\mathcal{R}$ (equivalently, the policy class $\Pi_{DPO,\beta}$) is restricted, the algorithm can aggressively (and incorrectly) extrapolate rewards for actions outside the dataset and, in doing so, inadvertently decrease the probabilities for preferred responses in the dataset. Meanwhile, in the same parameter range, $\chi$PO satisfies (see Figure 3)*

$$\widehat{\pi}_{\chi\text{PO}}(a_0) > \pi_{\text{ref}}(a_0),$$

*highlighting that pessimism can mitigate this phenomenon.*

### B.4 NONTRIVIALITY AND ROLE OF $V_{\text{max}}$ PARAMETER

To close this section, we discuss the role of the $V_{\text{max}}$ parameter (Assumption 3.2) used in the analysis of $\chi$PO (Theorem 3.1) in depth, motivating it from the perspective of the induced policy class $\Pi_{\chi\text{PO},\beta}$ from Appendix B.2.

Assumption 3.2 effectively implies that all policies $\pi \in \Pi$ satisfy $\left\|\frac{\pi}{\pi_{\text{ref}}}\right\|_\infty \lesssim \frac{V_{\text{max}}}{\beta}$; in other words, the policy class we use in $\chi$PO satisfies *all-policy $L_\infty$-concentrability* with $\max_{\pi \in \Pi} \mathcal{C}_\infty^\pi \lesssim \frac{V_{\text{max}}}{\beta}$. At first glance, this might seem to trivialize the offline alignment problem, since it would suffice to prove a generalization guarantee based on all-policy concentrability, and then plug this bound in. We will show that this is not the case, and that this is actually an intrinsic feature of $\chi^2$-regularization.

In more detail, recall that for $\chi$PO, we require the realizability assumption that $\pi_\beta^\star \in \Pi$ (Assumption 3.1), where $\pi_\beta^\star$ is the optimal mixed $\chi^2$-regularized policy that satisfies $r^\star(x,a) = \beta\phi\left(\frac{\pi_\beta^\star(a|x)}{\pi_{\text{ref}}(a|x)}\right) + Z_{\beta,r^\star}(x)$. This policy, via Proposition B.2, satisfies $\left\|\frac{\pi_\beta^\star}{\pi_{\text{ref}}}\right\|_\infty \lesssim \frac{R_{\text{max}}}{\beta}$, so from a statistical perspective, we can take Assumption 3.2 to hold without loss of generality by removing any policy that violates this bound. In addition, as highlighted by Example 3.1, if we begin from a class of bounded reward models $\mathcal{R}$ with $r^\star \in \mathcal{R}$, Assumption 3.2 holds with $V_{\text{max}} \lesssim R_{\text{max}}$ for the induced class $\Pi_{\chi\text{PO},\beta}$ defined in Eq. (24), even though knowledge of such a reward model class is a mild statistical assumption that clearly does not trivialize the learning problem.

On the other hand, for DPO, a minimal assumption is that $\pi_{\beta;\text{KL}}^\star \in \Pi$ (Xie et al., 2024), where $\pi_{\beta;\text{KL}}^\star$ is the optimal KL-regularized policy that satisfies $r^\star(x,a) = \beta\log\frac{\pi_{\beta;\text{KL}}^\star(a|x)}{\pi_{\text{ref}}(a|x)} + Z_{\beta,r^\star;\text{KL}}(x)$. Unlike the optimal mixed $\chi^2$-regularized policy, $\pi_{\beta;\text{KL}}^\star$ has $\frac{\pi_{\beta;\text{KL}}^\star(a|x)}{\pi_{\text{ref}}(a|x)} \gtrsim \exp\left(\frac{R_{\text{max}}}{\beta}\right)$. This means that it is impossible to find a policy class that simultaneously (1) realizes $\pi_{\beta;\text{KL}}^\star$, and (2) satisfies all-policy concentrability with $\max_{\pi \in \Pi} \mathcal{C}_\infty^\pi \ll \exp\left(\frac{R_{\text{max}}}{\beta}\right)$. As the bias of DPO is unacceptably large unless $\beta = \text{poly}(1/n)$ (the "small-$\beta$" regime), this leads to vacuous guarantees.

In view of these observations, our analysis of $\chi$PO can be interpreted as (implicitly) showing that for any bounded reward class $\mathcal{R}$, there exists a policy class $\Pi$ (precisely, the class $\Pi_{\chi\text{PO},\beta}$ defined in Eq. (24)) such that the following properties hold:

1. **Bounded bias.** For every $r \in \mathcal{R}$, there exists $\pi_r \in \Pi$ such that for all policies $\pi^\star$, $J_r(\pi^\star) - J_r(\pi_r) \lesssim \beta \cdot \mathcal{C}^{\pi^\star}$.

2. **Bounded overoptimization.** For all $\pi \in \Pi$, $\left\|\frac{\pi}{\pi_{\text{ref}}}\right\|_\infty \lesssim \frac{R_{\text{max}}}{\beta}$.

We view this as an interesting and non-trivial contribution in its own right. We mention in passing that while it is indeed possible to analyze $\chi$PO by first proving a sample complexity guarantee based on all-policy concentrability and then using that $\max_{\pi \in \Pi} \mathcal{C}_\infty^\pi \lesssim \frac{V_{\text{max}}}{\beta}$, this would lead to a loose bound relative to Theorem 3.1.

---

**Algorithm 2** $\chi^2$-RLHF

---

**input:** Reference policy $\pi_{\mathsf{ref}}$, preference dataset $\mathcal{D}_{\mathsf{pref}}$, unlabeled context dataset $\mathcal{D}_{\mathsf{x}}$, $\chi^2$-regularization coefficient $\beta > 0$, smoothing parameter $\eta \geq 0$.

1: **Estimate reward model via maximum likelihood:**

$$\widehat{r} \leftarrow \operatorname*{argmax}_{r \in \mathcal{R}} \sum_{(x, a_+, a_-) \in \mathcal{D}_{\mathsf{pref}}} \log \left[ \sigma \left( r(x, a_+) - r(x, a_-) \right) \right]. \tag{26}$$

2: Define $\chi^2$-regularized RLHF objective:

$$\widehat{J}_{\beta, \eta}(\pi) := \frac{1}{n_{\mathsf{x}}} \sum_{x \in \mathcal{D}_{\mathsf{x}}} \left( \mathbb{E}_{a \sim \pi(\cdot | x)}[\widehat{r}(x, a)] - \beta \sum_a \frac{\pi^2(a|x)}{\pi_{\mathsf{ref}}(a|x) + \eta\pi(a|x)} \right).$$

3: **Policy optimization:** Compute $\widehat{\pi} \in \Pi$ such that

$$\widehat{J}_{\beta, \eta}(\widehat{\pi}) \geq \max_{\pi \in \Pi} \widehat{J}_{\beta, \eta}(\pi) - \varepsilon_{\mathrm{opt}}.$$

4: **return:** $\widehat{\pi}$.

---

## C   SAMPLE COMPLEXITY GUARANTEES FOR $\chi^2$-RLHF

The $\chi^2$-regularization framework we consider (Section 3.1) can be used to derive algorithms beyond just $\chi$PO, and we expect it to find broader use. To highlight this, in this section we analyze the algorithm that directly optimizes a variant of the $\chi^2$-regularized RLHF objective in Eq. (6); this can be accomplished via policy optimization methods such as PPO, in the vein of classical RLHF approaches to offline alignment (Christiano et al., 2017; Bai et al., 2022; Ouyang et al., 2022; von Werra et al., 2020). As we will show, a benefit of directly optimizing the RLHF objective is that it allows us to provide guarantees that avoid dependence on the $V_{\mathsf{max}}$ parameter in Theorem 3.1, which may lead to improvement when $\Pi$ includes policies with very large or very small density ratios $\frac{\pi}{\pi_{\mathsf{ref}}}$.

**Algorithm.**   Our algorithm, $\chi^2$-RLHF is displayed in Algorithm 2. At the population level, the algorithm aims to optimize a variant of Eq. (7) that incorporates a small but important modification that allows us to avoid dependencies on $\frac{\pi}{\pi_{\mathsf{ref}}}$. Given *smoothing parameter* $\eta > 0$, define the *smoothed $\chi^2$-divergence* $D_{\chi^2;\eta}(\pi \parallel \pi_{\mathsf{ref}}) := \mathbb{E}_\pi \left[ \frac{\pi(a|x)}{\pi_{\mathsf{ref}}(a|x) + \eta\pi(a|x)} \right]$. We aim to find

$$\operatorname*{argmax}_\pi J_{\beta, \eta}(\pi) := \mathbb{E}_\pi \left[ r^\star(x, a) \right] - \beta D_{\chi^2;\eta}(\pi \parallel \pi_{\mathsf{ref}}) \tag{25}$$

$$= \operatorname*{argmax}_\pi \mathbb{E}_\pi \left[ r^\star(x, a) - \beta \frac{\pi(a \mid x)}{\pi_{\mathsf{ref}}(a \mid x) + \eta\pi(a \mid x)} \right].$$

The smoothing parameter $\eta$ effectively clips the policy ratio in $D_{\chi^2;\eta}(\pi \parallel \pi_{\mathsf{ref}})$ where $\pi_{\mathsf{ref}}(a|x) \ll \eta\pi(a|x)$; $D_{\chi^2}(\cdot \parallel \cdot)$ corresponds to the special (non-clipped) case where $\eta = 0$. In particular, clipping ensures a uniform bound of the form $D_{\chi^2;\eta}(\pi \parallel \pi_{\mathsf{ref}}) \leq \eta^{-1}$, whereas the best bound we can hope for with the unclipped $\chi^2$-divergence is $D_{\chi^2}(\pi \parallel \pi_{\mathsf{ref}}) = \mathbb{E}_\pi \left[ \frac{\pi(a|x)}{\pi_{\mathsf{ref}}(a|x)} \right] \leq \mathcal{C}_\infty^\pi$. For this reason, smoothing will allow us to obtain guarantees that avoid dependence on all-policy concentrability or parameters similar to $V_{\mathsf{max}}$.

To optimize Eq. (25), Algorithm 2 takes two datasets as input, along with a user-specified reward model class $\mathcal{R}$ and policy class $\Pi$. The first dataset, $\mathcal{D}_{\mathsf{pref}}$, is labeled with human preferences, and is used to learn a reward model $\widehat{r}$ via maximum likelihood estimation in Line 1. The second, $\mathcal{D}_{\mathsf{x}}$, contains *only unlabeled contexts* sampled from $\rho$, and is utilized in Line 3 to learn a policy that approximately maximizes an empirical version of Eq. (25). Importantly, because Line 3 involves an empirical expectation over only contexts, it is a purely computational problem that we can solve using algorithms like PPO; we allow for tolerance $\varepsilon_{\mathrm{opt}}$ in Line 3 to accommodate optimization error from such algorithms. By using unlabeled contexts in Line 3, we can obtain tighter guarantees when $\mathcal{D}_{\mathsf{x}}$ is large. This is often the case in practice, where unlabeled contexts are cheap to obtain, but preferences can be expensive to query.

**Theoretical guarantees.**   To analyze $\chi^2$-RLHF, we make similar assumptions to those utilized in Theorem 3.1 for $\chi$PO. Since $\chi^2$-RLHF utilizes separate reward and policy classes, we require

realizability conditions for both. Namely, $\mathcal{R}$ must be able to express the true reward function $r^\star$, and $\Pi$ must include the optimal policy for the regularized RLHF objective in Eq. (25).

**Assumption C.1.** *The reward function class satisfies $r^\star \in \mathcal{R}$, and is bounded so that $r(x, a) \in [0, R_{\max}]$ for all $r \in \mathcal{R}$ and $(x, a) \in \mathcal{X} \times \mathcal{A}$.*

**Assumption C.2.** *The policy class $\Pi$ satisfies $\pi^\star_{\beta,\eta} \in \Pi$, where $\pi^\star_{\beta,\eta}$ is the optimal policy for Eq. (25).*

Below is our main sample complexity guarantee for $\chi^2$-RLHF. While it is stated for a fixed, $\beta$-dependent smoothing parameter for compactness, the general version of this result (Theorem K.1) allows for general $\eta$.

**Theorem C.1.** *Let $\beta > 0$ be given, and suppose Assumptions C.1 and C.2 hold any $\eta \in \left[0, \frac{\beta}{8R_{\max}}\right]$. With probability at least $1 - \delta$, $\chi^2$-RLHF (Algorithm 2) produces a policy $\widehat{\pi}$ such that for all policies $\pi^\star$ simultaneously, we have*
$$J(\pi^\star) - J(\widehat{\pi})$$
$$\lesssim R_{\max} e^{2R_{\max}} \cdot \sqrt{\frac{\mathcal{C}^{\pi^\star} \log(|\mathcal{R}|/\delta)}{n}} + \beta \cdot \mathcal{C}^{\pi^\star} + \beta^{-1} \cdot \frac{R_{\max}^2 e^{4R_{\max}} \log(|\mathcal{R}|/\delta)}{n} + R_{\max} \sqrt{\frac{\log(|\Pi|/\delta)}{n_{\mathsf{x}}}} + \varepsilon_{\mathrm{opt}}.$$

*In particular, given any comparator policy $\pi^\star$, we can choose the regularization parameter $\beta$ to achieve*
$$J(\pi^\star) - J(\widehat{\pi}) \lesssim R_{\max} e^{2R_{\max}} \cdot \sqrt{\frac{\mathcal{C}^{\pi^\star} \log(|\mathcal{R}|/\delta)}{n}} + R_{\max} \sqrt{\frac{\log(|\Pi|/\delta)}{n_{\mathsf{x}}}} + \varepsilon_{\mathrm{opt}}. \qquad (27)$$

Above, we see that $\chi^2$-RLHF, like $\chi$PO, has sample complexity that scales only with the single-policy concentrability coefficient $\mathcal{C}^{\pi^\star}$, and holds for all comparator policies $\pi^\star$ simultaneously. Since the choice of $\beta$ induces a similar bias-overoptimization tradeoff in the first statement of Theorem C.1 as it did in Theorem 3.1 for $\chi$PO, we focus our discussion on the guarantee for a tuned choice of $\beta$ (Eq. (27)). The first term in Eq. (27) accounts for the reward estimation error (Line 1) and scales with $\mathcal{C}^{\pi^\star}$; as before, this accounts for how well rewards estimated from $\pi_{\mathrm{ref}}$ transfer to other candidate policies. The second term in Eq. (27) accounts for the statistical error from sampled contexts used in Line 3 for policy optimization. In particular, it is possible to drive this term to be much smaller than the first by using a larger unlabeled context dataset, which is typically far cheaper to acquire.

**Computationally efficiency.** Theorem C.1 bounds the sample complexity of $\chi^2$-RLHF under the assumption that we can solve Line 3 up to $\varepsilon_{\mathrm{opt}}$-accuracy. This is a purely computational problem, and in practice it can be solved using policy gradient methods such as PPO.

**Comparison to $\chi$PO.** Unlike $\chi$PO (Theorem 3.1), Theorem C.1 has no dependence on the parameter $V_{\max}$ or quantities such as $\frac{\pi}{\pi_{\mathrm{ref}}} \le \max_\pi \mathcal{C}^\pi_\infty$. We primarily attribute this to the fact that $\chi^2$-RLHF uses an explicit reward function class $\mathcal{R}$, and normalizing or clipping it to the reward range $R_{\max}$ is both natural and routinely done in practice (Shah et al., 2015; Christiano et al., 2017; Ouyang et al., 2022). In comparison, the implicit reward models induced by the policy class $\Pi$ in $\chi$PO can have larger range, and clipping the policy class in $\chi$PO directly, e.g., so that $|\beta\phi(\frac{\pi}{\pi_{\mathrm{ref}}})|$ is bounded, is misguided, because the policy class may lose realizability (Assumption 3.1). This is because $r^\star(x, a) = \beta\phi\left(\frac{\pi^\star_\beta(a|x)}{\pi_{\mathrm{ref}}(a|x)}\right) + Z_{\beta,r^\star}(x)$, and the normalization factor $Z_{\beta,r^\star}$ cannot be reasonably accounted for when clipping $\Pi$. While the $V_{\max}$ (Assumption 3.2) parameter involves pairs of action probabilities, and thereby sidesteps the normalization constant issue, it may not always be practical to modify $\Pi$ so that $V_{\max}$ is bounded, since this would require checking all pairs of each policy's action probabilities.

However, using an explicit reward function class alone is not enough. As discussed previously, when we move from implicit to explicit $\chi^2$-regularization, incorporating the smoothing parameter $\eta$ in Eq. (25) is essential to avoid statistical errors due to policies with large density ratios when we approximate the $\chi^2$-regularizer with empirical data. A careful choice of $\eta = \beta/R_{\max}$ in Theorem C.1 balances the benefits of clipping against the bias it introduces. Without smoothing (i.e., $\eta = 0$), a guarantee that depends on $\max_\pi \mathcal{C}^\pi_\infty$ for $\chi^2$-RLHF would be unavoidable, since the sample complexity must scale with the range of the problem, which grows with the magnitude of the regularizer. See Corollary K.2 in Appendix K for a guarantee in the case where $\eta = 0$, which highlights this.

# D  $\chi$PO FOR GENERAL PREFERENCE MODELS

All of our results so far concern the Bradley-Terry model (Eq. (1)), which, as highlighted in prior work, is somewhat restrictive. Thus, in this section, we turn our attention to offline alignment under a *general preference model* which does not assume transitivity (Munos et al., 2023; Wang et al., 2023b; Swamy et al., 2024; Rosset et al., 2024; Ye et al., 2024). The setup is the same as Section 2, but we assume that for a given context $x$ and pair of actions $(a, b)$, the preference $y \in \{0, 1\}$ is generated via a Bernoulli Distribution

$$y \sim \text{Ber}(\mathcal{P}^\star(a \succ b \mid x)), \tag{28}$$

where $\mathcal{P}^\star(a \succ b \mid x) \in [0, 1]$ is a general preference distribution. For a pair of policies $\pi, \pi'$, let $\mathcal{P}^\star(\pi \succ \pi') := \mathbb{E}_{x \sim \rho}[\mathcal{P}^\star(\pi(x) \succ \pi'(x) \mid x)]$. Following Wang et al. (2023b); Munos et al. (2023); Swamy et al. (2024), we consider the *minimax winner* (Kreweras, 1965; Simpson, 1969; Kramer, 1973; Fishburn, 1984) or *von Neumann winner* (Dudík et al., 2015) as a solution concept:

$$\pi_{\text{MW}} := \underset{\pi \in \Pi}{\text{argmax}} \min_{\pi' \in \Pi} \mathcal{P}^\star(\pi \succ \pi').$$

It will be useful to slightly reparameterize this formulation by introducing the preference function $\ell^\star(x, a, b) := 2\mathcal{P}^\star(a \succ b \mid x) - 1$. Note that for any well-defined preference model, we have $\mathcal{P}^\star(a \succ b \mid x) + \mathcal{P}^\star(b \succ a \mid x) = 1$ for all $x, a, b$, which indicates that $\ell^\star$ satisfies skew symmetry:

$$\ell^\star(x, a, a) = 0, \qquad \ell^\star(x, a, b) + \ell^\star(x, b, a) = 0, \qquad \forall x \in \mathcal{X}, a, b \in \mathcal{A}.$$

Furthermore, the minimax winner above is equivalent to

$$\pi_{\text{MW}} := \underset{\pi \in \Pi}{\text{argmax}} \min_{\pi' \in \Pi} \ell^\star(\pi, \pi'), \tag{29}$$

where $\ell^\star(\pi, \pi') := \mathbb{E}_{x \sim \rho, a \sim \pi(x), b \sim \pi'(x)}[\ell^\star(x, a, b)]$. Concretely, our goal is to use the logged preference data $\mathcal{D}_{\text{pref}} = \{(x, a_+, a_-)\}$ (with $(a_+, a_-)$ labeled according to Eq. (28)) to compute a policy $\widehat{\pi}$ that is an $\varepsilon$-approximate minimax winner, in the sense that

$$\text{DG}(\widehat{\pi}) := \max_{\pi \in \Pi} \ell^\star(\pi, \widehat{\pi}) - \min_{\pi \in \Pi} \ell^\star(\widehat{\pi}, \pi) \leq \varepsilon. \tag{30}$$

## D.1  IMPOSSIBILITY OF SINGLE-POLICY CONCENTRABILITY UNDER GENERAL PREFERENCES

While the general preference framework above is more powerful than the Bradley-Terry model, we now show that there is a statistical cost for this generality. In particular, our first result in this section shows that in contrast to the Bradley-Terry model, it is not possible to achieve sample complexity guarantees that scale with single-policy concentrability under general preferences, even when the learner has access to a small class of preference models $\mathscr{P}$ that contains the true preference model $\mathcal{P}$ (i.e., $\mathcal{P}^\star \in \mathscr{P}$).

**Theorem D.1** (Impossibility of single-policy concentrability under general preferences)**.** *There exists two problem instances* $\theta_1 = (\rho, \mathcal{P}_1^\star, \Pi)$ *and* $\theta_2 = (\rho, \mathcal{P}_2^\star, \Pi)$ *differing only in their ground truth preference model, a data collection policy* $\pi_{\text{ref}}$*, and a preference model class* $\mathscr{P} = \{\mathcal{P}_1^\star, \mathcal{P}_2^\star\}$ *with* $|\mathscr{P}| = 2$ *such that the following hold:*

1. *For both instances, the single-policy* $L_\infty$*-concentrability coefficient for a minimax winner is bounded:* $\min_{\pi_{\text{MW}}} \mathcal{C}_\infty^{\pi_{\text{MW}}} \leq 2$.[6]

2. *For any* $n \in \mathbb{N}$ *and any algorithm* Alg *which derives a policy* $\widehat{\pi}$ *from a dataset* $\mathcal{D}_{\text{pref}}$ *of* $n$ *samples, there exists an instance* $\theta \in \{\theta_1, \theta_2\}$ *such that* $\pi_{\text{ref}}$ *incurs constant suboptimality:*

$$\min_{\text{Alg}} \max_{i \in \{1, 2\}} \mathbb{E}_{\mathcal{D}_{\text{pref}} \sim \theta_i}[\text{DG}(\text{Alg}(\mathcal{D}_{\text{pref}}); \theta_i)] \geq \frac{1}{8},$$

   *where* $\text{DG}(\pi; \theta)$ *is the duality gap for policy* $\pi$ *on instance* $\theta$*.*

This lower bound is inspired by similar results in the literature on offline RL in two-player zero-sum Markov games (Cui and Du, 2022). However, the lower bound constructions in Cui and Du (2022) cannot be directly applied as-is, because they do not satisfy the skew-symmetry property required by the general preference alignment framework. Our lower bound highlights that even under skew-symmetry, it is impossible to achieve single-policy concentrability for offline learning in two-player zero-sum games.

---

[6]In general, the minimax winner may not be unique. We compete against the minimax winner with the *best* possible single-policy concentrability coefficient.

---

**Algorithm 3** Iterative $\chi$PO for General Preferences

---

1: **Input**: labeled preference dataset $\mathcal{D}_{\mathsf{pref}}$, preference model class $\mathcal{L}$, regularization coefficient $\beta$, stepsize $\eta$, total number of iterations $T$.
2: **Initialize**: $\pi^1 = \pi_{\mathsf{ref}}$.
3: Learn a preference model $\widehat{\ell}$ via least-squares regression:

$$\widehat{\ell} = \underset{\ell \in \mathcal{L}}{\arg\min} \sum_{(x,a_+,a_-) \in \mathcal{D}_{\mathsf{pref}}} (\ell(x, a_+, a_-) - 1)^2 .$$

4: Collect $m$ samples $\mathcal{D}_{\mathsf{x}} = \{(x, a, b)\}$ where each sample is drawn i.i.d. from $x \sim \rho, a \sim \pi_{\mathsf{ref}}(x), b \sim \pi_{\mathsf{ref}}(x)$.
5: **for** $t = 1, \cdots, T$ **do**
6:     Sample $b_t \sim \pi^t(x)$ and let $\widehat{r}^t(x, a) = \widehat{\ell}(x, a, b_t)$ for all $x \in \mathcal{X}, a \in \mathcal{A}$.
7:     Compute

$$\pi^{t+1} = \underset{\pi \in \Pi}{\arg\min} \sum_{(x,a,b) \in \mathcal{D}_{\mathsf{x}}} \left( \mathsf{clip}_4 \left( f_{\pi,\pi^t}^{\beta,\eta}(x, a, b) \right) - (\widehat{r}^t(x, a) - \widehat{r}^t(x, b)) \right)^2, \qquad (32)$$

    where $f_{\pi,\pi^t}^{\beta,\eta}(x, a, b)$ is defined in Eq. (31).
8: **Output**: $\widehat{\pi} = \mathsf{unif}(\{\pi^t\}_{t=1}^T)$.

---

## D.2    Iterative $\chi$PO for General Preferences

In spite of the hardness in the prequel, we now show that an iterative variant of $\chi$PO—based on self-play—can learn a near-optimal minimax winner under the general preference model under a new local coverage condition—a condition that is stronger than the single policy concentrability but much weaker than global/all-policy concentrability and the notion of unilateral concentrability introduced by Cui and Du (2022).

Our algorithm, Iterative $\chi$PO, is described in Algorithm 3, and consists of two main steps.

**Preference model estimation via least squares regression on $\mathcal{D}_{\mathsf{pref}}$.** We first (Line 3) learn a preference model from the offline preference dataset $\mathcal{D}_{\mathsf{pref}}$. We assume access to a preference function class $\mathcal{L}$ which is realizable in the sense that $\ell^\star \in \mathcal{L}$ and where all $\ell \in \mathcal{L}$ satisfy skew-symmetryc, and we will estimate $\ell^\star$ rather than $\mathcal{P}^\star$. We perform least-squares regression on $\mathcal{D}_{\mathsf{pref}}$ with $\mathcal{L}$ to learn $\ell^\star$:

$$\widehat{\ell} = \underset{\ell \in \mathcal{L}}{\arg\min} \sum_{(x,a_+,a_-) \in \mathcal{D}_{\mathsf{pref}}} (\ell(x, a_+, a_-) - 1)^2 .$$

**Policy optimization with iterative $\chi$PO update.** Given the estimated model $\widehat{\ell}$, we compute an approximate minimax winner using an iterative regression scheme inspired by Gao et al. (2024). We proceed in $T$ iterations (Line 5), where at each iteration $t$, we define an iteration-dependent reward function $\overline{r}^t(x, a)$ based on the current policy $\pi^t$ as

$$\overline{r}^t(x, a) = \mathbb{E}_{b \sim \pi^t(x)}[\widehat{\ell}(x, a, b)], \qquad \forall x \in \mathcal{X}, a \in \mathcal{A}.$$

Then, for all $\pi, \pi' \in \Pi$, we define a policy-dependent predictor $f_{\pi,\pi'}^{\beta,\eta}(x, a, b)$, whose motivation will be described in detail momentarily, as follows:

$$\begin{aligned} f_{\pi,\pi'}^{\beta,\eta}(x, a, b) &:= \left( 1 + \frac{1}{\eta} \right) \cdot \left( \beta\phi \left( \frac{\pi(a \mid x)}{\pi_{\mathsf{ref}}(a \mid x)} \right) - \beta\phi \left( \frac{\pi(b \mid x)}{\pi_{\mathsf{ref}}(b \mid x)} \right) \right) \\ &\quad - \frac{1}{\eta} \left( \beta\phi \left( \frac{\pi'(a \mid x)}{\pi_{\mathsf{ref}}(a \mid x)} \right) - \beta\phi \left( \frac{\pi'(b \mid x)}{\pi_{\mathsf{ref}}(b \mid x)} \right) \right) \end{aligned} \qquad (31)$$

Using $f_{\pi,\pi^t}^{\beta,\eta}(x, a, b)$ as a policy-parameterized regression function, we (Line 7) compute the next policy $\pi^{t+1}$ by solving a least-squares regression problem in which the Bayes optimal solution is the *relative reward* $\overline{r}^t(x, a) - \overline{r}^t(x, b)$ for iteration $t$.

Let us now explain the intuition behind the the predictor $f_{\pi,\pi'}^{\beta,\eta}(x,a,b)$. Suppose that the regression step in Line 7 learns a predictor that can perfectly model the relative reward, i.e.,

$$\forall x,a,b, \quad f_{\pi^{t+1},\pi^t}^{\beta,\eta}(x,a,b) = \overline{r}^t(x,a) - \overline{r}^t(x,b),$$

In this case, we can show that the returned policy $\pi^{t+1}$ is the optimal policy for the following mixed $\chi^2$-regularized RL objective:

$$\pi^{t+1}(x) = \underset{p \in \Delta(\mathcal{X})}{\operatorname{argmax}}\left\{\mathbb{E}_{a\sim p}\left[\overline{r}^t(x,a)\right] - \beta D_{f_{\chi_{\mathrm{mix}}}}(p \,\|\, \pi_{\mathrm{ref}}(x)) - \frac{\beta}{\eta}B_x(p,\pi^t)\right\}, \qquad \forall x \in \mathcal{X}, \quad (33)$$

where $B_x(p,\pi^t)$ is the Bregman divergence induced by the regularizer $p \mapsto D_{f_{\chi_{\mathrm{mix}}}}(p \,\|\, \pi_{\mathrm{ref}}(x))$, i.e.,

$$B_x(p,q) := D_{f_{\chi_{\mathrm{mix}}}}(p \,\|\, \pi_{\mathrm{ref}}(x)) - D_{f_{\chi_{\mathrm{mix}}}}(q \,\|\, \pi_{\mathrm{ref}}(x)) - \left\langle \nabla D_{f_{\chi_{\mathrm{mix}}}}(q \,\|\, \pi_{\mathrm{ref}}(x)), p - q\right\rangle, \qquad \forall x \in \mathcal{X}.$$

Thus, the algorithm can be understood as running mirror descent on the iteration-dependent loss function $-\overline{r}^t$, with $p \mapsto D_{f_{\chi_{\mathrm{mix}}}}(p \,\|\, \pi_{\mathrm{ref}}(x))$ as a per-context regularizer. This technique draws inspiration from Chang et al. (2024), in which the authors apply a similar regularized mirror descent algorithm to learn the optimal policy for the *reward-based* setting. The motivation for using mixed-$\chi^2$ regularization is exactly the same as in $\chi$PO: we want to ensure that $\frac{\pi^{t+1}(a|x)}{\pi_{\mathrm{ref}}(a|x)} \leq 1 + \frac{1}{\beta}$, thereby mitigating overoptimization.

### D.3 THEORETICAL ANALYSIS OF ITERATIVE $\chi$PO

We now present our main theoretical guarantees for Iterative $\chi$PO. We begin by stating a number of statistical assumptions. We first assume that the preference model class contains the ground truth preference function $\ell^\star$.

**Assumption D.1** (Preference function realizability). *The model class $\mathcal{L}$ satisfies $\ell^\star \in \mathcal{L}$ where $\ell^\star$ is the ground truth preference function.*

In addition, since Algorithm 3 iteratively applies an $\chi$PO update, we require that a policy realizability assumption analogous to Assumption 3.1 holds for each of the sub-problems in Eq. (33). Concretely, we make the following assumption.

**Assumption D.2** (Policy realizability for general preferences). *For any policy $\pi \in \Pi$ and $\ell \in \mathcal{L}$, the policy class $\Pi$ contains the minimizer of the following regularized RL objective:*

$$\overline{\pi}(x;\ell,\pi) := \underset{p \in \Delta(\mathcal{X})}{\operatorname{argmax}}\left\{\mathbb{E}_{a\sim p, b\sim\pi(x)}[\ell(x,a,b)] - \beta D_{f_{\chi_{mix}}}(p \,\|\, \pi_{\mathrm{ref}}(x)) - \frac{\beta}{\eta}B_x(p,\pi)\right\}, \qquad \forall x \in \mathcal{X}.$$

Finally, we require that the implicit reward functions in Eq. (32) are bounded, analogous to Assumption 3.2.

**Assumption D.3** (Bounded implicit rewards for general preferences). *For a parameter $V_{\mathrm{max}} \geq 2$, it holds that for all $\pi,\pi' \in \Pi$, $x \in \mathcal{X}$, and $a,b \in \mathcal{A}$,*

$$\left|f_{\pi,\pi'}^{\beta,\eta}(x,a,b)\right| \leq V_{\mathrm{max}}. \tag{34}$$

Our main guarantee for Algorithm 3 is as follows.

**Theorem D.2.** *Fix any $\delta \in (0,1]$. Suppose Algorithm 3 is invoked with $T = \frac{mn}{nV_{\mathrm{max}}^2+m}$, $\beta = \frac{1}{\sqrt{T}}$, and $\eta = \frac{1}{T}$. Then under Assumption D.1, Assumption D.2 and Assumption D.3, we have that probability at least $1-\delta$,*

$$\mathsf{DG}(\widehat{\pi}) \lesssim \min_{C\geq 1}\left\{\mathsf{subopt}(\widehat{\pi},C) + C\left(\frac{V_{\mathrm{max}}\log(|\Pi|/\delta)}{\sqrt{m}} + \frac{\log(|\Pi||\mathcal{L}|/\delta)}{\sqrt{n}}\right)\right\},$$

*where $\mathsf{subopt}(\widehat{\pi},C) := \max_{\pi\in\Pi} \ell^\star(\pi,\widehat{\pi}) - \max_{\pi\in\Pi_C} \ell^\star(\pi,\widehat{\pi})$ and $\Pi_C := \{\pi : \max_{x\in\mathcal{X}} D_{\chi^2}(\pi(x) \,\|\, \pi_{\mathrm{ref}}(x)) \leq C\}$. In particular, if we define the unilateral concentrability coefficient as*

$$C_{\mathrm{uni}} := \max_{\pi\in\Pi, x\in\mathcal{X}, a,b\in\mathcal{A}} \frac{\pi(a\mid x)\pi_{\mathrm{MW}}(b\mid x)}{\pi_{\mathrm{ref}}(a\mid x)\pi_{\mathrm{ref}}(b\mid x)},$$

*then the bound above implies that*

$$\mathsf{DG}(\widehat{\pi}) \lesssim C_{\mathrm{uni}} \cdot \left(\frac{V_{\mathrm{max}}\log(|\Pi|/\delta)}{\sqrt{m}} + \frac{\log(|\Pi||\mathcal{L}|/\delta)}{\sqrt{n}}\right).$$

The first result gives a tradeoff between the statistical error and the approximation error $\mathrm{subopt}(\widehat{\pi}, C)$, which is modulated by the parameter $C$. This tradeoff is analogous to, but more subtle, than the one for $\chi$PO in the reward-based setting. In the reward-based setting, $\chi$PO has low regret to the best policy covered $\pi_{\mathrm{ref}}$. In the general preference setting, Algorithm 3 has small duality gap if, for any policy, there is an approximate best response that is covered by $\pi_{\mathrm{ref}}$ (this implies that $\mathrm{subopt}(\widehat{\pi}, C)$ is small for small $C$). Crucially, Algorithm 3 does not require that all policies are covered by $\pi_{\mathrm{ref}}$, which is a distinctive feature of mixed $\chi^2$-regularization and reflects the algorithms robustness to overoptimization.

The second result concerns the setting where all policies are covered by $\pi_{\mathrm{ref}}$ and is easier to interpret. Indeed, if all $\pi \in \Pi$ satisfy $D_{\chi^2}(\pi \parallel \pi_{\mathrm{ref}}) \leq C^\star$, then $\mathrm{subopt}(\widehat{\pi}, C^\star) = 0$, which implies that we can learn an $\varepsilon$-approximate minimizer using $\widetilde{O}(C^\star/\varepsilon^2)$ samples. Thus, we obtain a guarantee based on unilateral concentrability (Cui and Du, 2022), which is a stronger condition, i.e., we always have $\max_\pi D_{\chi^2}(\pi \parallel \pi_{\mathrm{ref}}) \leq C_{\mathrm{uni}}$. However, per the above discussion, the first part of Theorem D.2 is stronger than results based on unilateral concentrability and hints at a new notion of coverage for general preferences. Lastly, we remark that the parameter $V_{\mathrm{max}}$ only affects $\sqrt{1/m}$ term in Theorem D.2, so dependence on this parameter can be mitigated using unlabeled data.

Theorem D.2 is closely related to recent work of Ye et al. (2024), which uses pessimism to learn a regularized minimax winner, and achieves polynomial sample complexity with a concentrability assumption similar to Theorem D.2. However, there are two key differences. First, their learning objective is the KL-regularized minimax winner, while we study the unregularized objective and use $\chi^2$-regularization. More importantly, their theoretical algorithm is computationally inefficient as it constructs an explicit confidence set for the preference model and performs max-min-style policy optimization. In contrast, our algorithm only requires solving standard supervised learning problems.

# E   EXPERIMENTS IN OFFLINE LANGUAGE MODEL ALIGNMENT

## E.1   TL;DR SUMMARIZATION

We perform preliminary evaluations of $\chi$PO for offline language model alignment on the TL;DR dataset (Stiennon et al., 2020), using DPO as our comparison baseline. The reference policy $\pi_{\mathrm{ref}}$ is the Pythia-1b model (Biderman et al., 2023) pre-trained on SFT data (`cleanrl/EleutherAI_pythia-1b-deduped__sft__tldr` from Huang et al. (2022)), and performance is measured via winrate against a baseline, as judged by GPT-4o. All parameters that are not algorithm-specific, such as the learning rate, are shared by both $\chi$PO and DPO in order to ensure a fair comparison (see Appendix E.2 for details).

In Table 1 we display the winrates of $\chi$PO and DPO over several choices of training epochs, as well as regularization parameter $\beta$. The winrate corresponds to the final checkpoint learned by each algorithm for each set of hyperparameters. We consider $\beta = 0.05$ and 1 epoch of training to be a standard setup for DPO (Gao et al., 2024; Guo et al., 2024; Rafailov et al., 2024a), and, as we are particularly concerned with regimes where overoptimization is of concern, we additionally analyze performance when epochs are increased, and/or $\beta$ is decreased (corresponding to less regularization).

Over all choices of $\beta$ and epochs, $\chi$PO achieves a higher average winrate than DPO. While the difference is not significant for $\beta = 0.05$ and 1 epoch, the performance gap grows significantly as the number of epochs increases, demonstrating the robustness of $\chi$PO to overoptimization. Further, while DPO degrades completely for $\beta = 0.005$, $\chi$PO is robust over two orders of magnitude of $\beta$, reinforcing trends seen earlier in Figure 1 and the more favorable bias-overoptimization tradeoff from our theoretical analysis.

In addition, $\chi$PO exhibits better performance and robustness longitudinally throughout training, as shown in Appendix E.1. While DPO peaks early with high variance around 0.5 epochs and degrades thereafter, $\chi$PO continues to improve smoothly then plateaus over the last epoch. Further, for the same regularization parameter $\beta$, the $\chi$PO policy has significantly lower KL-divergence relative to $\pi_{\mathrm{ref}}$, demonstrating that the $\chi^2$-regularization is both a stronger regularizer and one that effectively mitigates overoptimization.

## E.2   EXPERIMENT DETAILS

**Dataset and models.**   For training, we use `trl-internal-testing/tldr-preference-trl-style`, with 92.9K train samples and 83.8K validation samples. The reference pol-

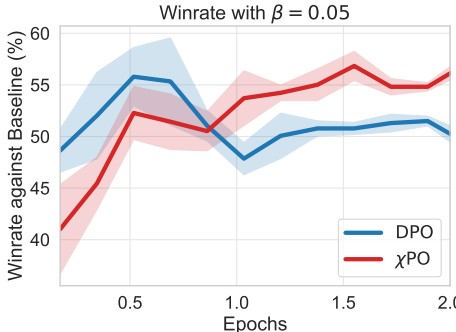 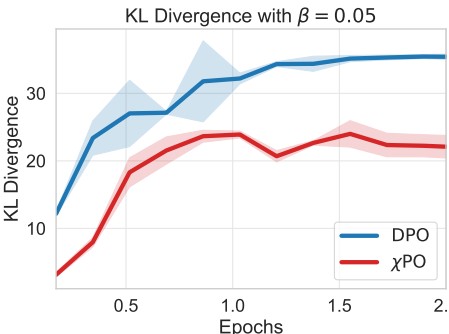

Figure 4: (Left) TL;DR Summarization winrate recorded every 250 steps, over 2 epochs of training. Shaded area displays $\pm 1$ standard error over 3 seeds. At 1 epoch $\chi$PO already obtains better performance, and continues to improve over the course of training, while DPO degrades over time. (Right) KL divergence $D_{\mathsf{KL}}(\widehat{\pi} \,\|\, \pi_{\mathsf{ref}})$ averaged over 2 of the seeds. For the same $\beta$, $\chi$PO constrains the learned policy to be significantly closer to $\pi_{\mathsf{ref}}$, thereby striking a better bias-variance tradeoff.

icy $\pi_{\mathsf{ref}}$ is the Pythia-1b model (Biderman et al., 2023) pre-trained on SFT data (`cleanrl/EleutherAI_pythia-1b-deduped__sft__tldr` from Huang et al. (2022)), and performance is measured via winrate against a baseline, as judged by GPT-4o. All parameters that are not algorithm-specific, such as the learning rate, are shared by both $\chi$PO and DPO in order to ensure a fair comparison.

**Training details.** Our implementation of $\chi$PO is built upon the DPO trainer from Transformer Reinforcement Learning (TRL) (von Werra et al., 2020). $\chi$PO comes with strong robustness and theoretical properties, but the policy ratios can sometimes introduce instability in training. In practice, we have observed that better stability and performance can be achieved by utilizing the (more general form) link function $\widetilde{\phi}(z) := \exp\left(\mathsf{clip}_{[-88,20]}(\alpha \cdot \log z)\right) + \gamma \cdot \log z$ in Algorithm 1, and performing a small grid search over additional parameters $\alpha = \{\frac{1}{4}, 1\}$ and $\gamma = \{0.1, 1\}$ for a fixed $\beta$.

We briefly discuss each parameter in turn. The mixing parameter $\gamma$ controls the relative ratios of KL- and $\chi^2$-regularization, our analysis in Appendix H.1 shows that Theorem 3.1 holds more generally for $\gamma \in (0, 1]$ (see Theorem H.1). Next, ignoring clipping, $\alpha \in (0, 1]$ in $\widetilde{\phi}$ implements regularization with the $(1 + \alpha)$-divergence (or Renyi divergence), which is an $f$-divergence that is stronger than KL-regularization but weaker than $\chi^2$-regularization (Van Erven and Harremos, 2014), and also carries single-policy concentrability guarantees (although with a slower-rate dependence on sample size $n$). For example, $\alpha = \frac{1}{4}$ corresponds to the link function $\phi(z) = (z)^{1/4} + \gamma \log z$, which is easier to optimize than the link function $\phi(z) = z + \gamma \log z$ (corresponding to $\alpha = 1$) induced by $\chi^2$-regularization, given the potentially large magnitude of $z = \frac{\pi}{\pi_{\mathsf{ref}}}$. Though we do not write out the analysis here, the methods used to prove the sample complexity of $\chi$PO (Theorem 3.1) can be used to prove analogous guarantees for regularization with $\alpha$-divergences, which will have slightly worse statistical rates.

Lastly, we provide some additional explanation for the clipping operation. We observed that `torch.exp` is prone to underflow when $\log \frac{\pi}{\pi_{\mathsf{ref}}}$ is very negative, and clipping the upper range to 20 can help reduce numerical instabilities. Clipping in such a manner is supported by our analysis in Proposition 4.1, which shows that $\frac{\pi^\star}{\pi_{\mathsf{ref}}} \leq 1 + \frac{R_{\max}}{\beta}$ (though technically we do not know $R_{\max}$). The parameters for all experiments are displayed in Table 2.

**Generation details.** For winrate evaluation, we use greedy, temperature 0, decoding. For computation of the KL divergence, we sample from the model with temperature 1. The maximum prompt length is 512, and the maximum response length is 200. We use the standard generation prompt "TL;DR:" (Gao et al., 2024).

**Evaluation of performance.** The performance of each algorithm is measured via winrate against responses in the SFT dataset, as measured by GPT-4o (global standard). The winrate is computed on a subset of 512 prompts from the SFT validation set

Table 2: Parameter settings in TL;DR summarizion

| Algorithm | Parameters |
|---|---|
| DPO | batch size: 64 |
| | learning rate: 1e-6 |
| | scheduler: cosine |
| | optimizer: adamw |
| $\chi$PO | batch size: 64 |
| | clip range: [-88, 20] |
| | learning rate: 1e-6 |
| | scheduler: cosine |
| | optimizer: adamw |
| $\beta = 0.05$, 1 epoch | $\alpha : 1.25, \gamma : 1.0$ |
| $\beta = 0.05$, 2 epochs | $\alpha : 2.00, \gamma : 1.0$ |
| $\beta = 0.05$, 4 epochs | $\alpha : 1.25, \gamma : 0.1$ |
| $\beta = 0.005$, all epochs | $\alpha : 1.25, \gamma : 0.1$ |

(`trl-internal-testing/tldr-preference-sft-trl-style`), and the order of the model and reference responses are randomized each round.

# Part II

# Proofs

## F  PRELIMINARIES

Recall that for a pair of probability measures $\mathbb{P}$ and $\mathbb{Q}$ with a common dominating measure $\omega$, Hellinger distance is defined via

$$D_{\mathsf{H}}^2(\mathbb{P}, \mathbb{Q}) = \int \left( \sqrt{\frac{\mathrm{d}\mathbb{P}}{\mathrm{d}\omega}} - \sqrt{\frac{\mathrm{d}\mathbb{Q}}{\mathrm{d}\omega}} \right)^2 \mathrm{d}\omega. \tag{35}$$

**Lemma F.1** (MLE for conditional density estimation (e.g., Wong and Shen (1995); de Geer (2000); Zhang (2006); Agarwal et al. (2020))). *Consider a conditional density $p^\star : \mathcal{X} \to \Delta(\mathcal{Y})$, where $\mathcal{X}$ is the instance space and $\mathcal{Y}$ is the target space. Let $\mathcal{D} = \{(x^i, y^i)\}_{i=1}^n$ be a dataset in which $(x^i, y^i)$ are drawn i.i.d. as $x^i \sim \rho \in \Delta(\mathcal{X})$ and $y^i \sim p^\star(y \mid x)$. Suppose we have a finite function class $\mathcal{P}$ such that $p^\star \in \mathcal{P}$, where $p(\cdot \mid x) \in \Delta(\mathcal{Y})$ for all $p \in \mathcal{P}$ and $x \in \mathcal{X}$. Define the maximum likelihood estimator*

$$\widehat{p} := \underset{p \in \mathcal{P}}{\operatorname{argmax}} \sum_{(x,y) \in \mathcal{D}} \log p(y \mid x).$$

*Then with probability at least $1 - \delta$,*

$$\mathbb{E}_{x \sim \rho}\left[ D_{\mathsf{H}}^2(\widehat{p}(\cdot \mid x), p^\star(\cdot \mid x)) \right] \leq \frac{2 \log(|\mathcal{P}|\delta^{-1})}{n}.$$

## G  ANALYSIS OF $\chi$PO: PROOF SKETCH FOR THEOREM 3.1

In this section, we sketch the proof of the main guarantee for $\chi$PO, Theorem 3.1, with the full proof deferred to Appendix H. A central object in the proof is the *implicit* reward model induced by the $\chi$PO policy $\widehat{\pi}$, which we define via

$$\widehat{r}(x, a) := \beta\phi\left( \frac{\widehat{\pi}(a \mid x)}{\pi_{\mathsf{ref}}(a \mid x)} \right). \tag{36}$$

As we will show, this reward model is a natural bridge between $\chi$PO and the corresponding mixed $\chi^2$-regularized RLHF objective in Section 3.1, and allows us to view $\chi$PO from a reward-based perspective. In particular, note that if we analogously define an induced reward model class $\mathcal{R}_\Pi :=$ $\left\{ r(x,a) = \beta\phi\left( \frac{\pi(a|x)}{\pi_{\text{ref}}(a|x)} \right) : \pi \in \Pi \right\}$, then Line 2 of $\chi$PO can be viewed as performing maximum likelihood estimation over this class (in the sense of Eq. (3)) under the Bradley-Terry model. Under Assumption 3.1, $\mathcal{R}_\Pi$ realizes the true reward function $r$ up to an action-independent shift. As a result, if we define $\Delta^r(x,a,b) := r(x,a) - r(x,b)$, then using a fairly standard generalization bound for maximum likelihood estimation (e.g., Wong and Shen (1995); Zhang (2006); de Geer (2000); see Lemma H.1), we can show that

$$\varepsilon_{\text{stat}}^2 := \mathbb{E}_{x\sim\rho, a\sim\pi_{\text{ref}}, b\sim\pi_{\text{ref}}}\left[ \left| \Delta^{\widehat{r}}(x,a,b) - \Delta^{r^\star}(x,a,b) \right|^2 \right] \leq O\left( V_{\max} e^{2R_{\max}} \cdot \frac{\log(|\Pi|/\delta)}{n} \right). \quad (37)$$

In other words, the estimated reward model $\widehat{r}$ is accurate under the action distribution induced by $\pi_{\text{ref}}$. However, $\widehat{r}$ may still be inaccurate for policies that select different actions from $\pi_{\text{ref}}$, raising concerns of overoptimization. To address this issue, we use the following lemma, which shows that $\chi^2$-divergence bounds the extent to which the accuracy of a reward model $\widehat{r}$ trained under $\pi_{\text{ref}}$ will transfer to a downstream policy $\pi$ of interest; this will motivate our use of $\chi^2$-regularization.

**Lemma G.1** (Informal version of Lemma H.3). *For any policy $\pi : \mathcal{X} \to \Delta(\mathcal{A})$, it holds that*

$$\mathbb{E}_{x\sim\rho, a\sim\pi(\cdot|x), b\sim\pi_{\text{ref}}(\cdot|x)}\left[ \left| \Delta^{\widehat{r}}(x,a,b) - \Delta^{r^\star}(x,a,b) \right| \right] \lesssim \sqrt{(1 + D_{\chi^2}(\pi \| \pi_{\text{ref}})) \cdot \varepsilon_{\text{stat}}^2}.$$

Going forward, let us abbreviate $\mathbb{E}_{\pi,\pi_{\text{ref}}}[\cdot] = \mathbb{E}_{x\sim\rho, a\sim\pi(\cdot|x), b\sim\pi_{\text{ref}}(\cdot|x)}[\cdot]$. Let $\pi^\star$ be an arbitrary policy. Noting that $\mathcal{C}^\pi = 1 + 2D_{\chi^2}(\pi \| \pi_{\text{ref}})$ and that

$$J(\pi^\star) - J(\widehat{\pi}) \lesssim \mathbb{E}_{\pi^\star,\pi_{\text{ref}}}\left[ \left| \Delta^{\widehat{r}}(x,a,b) - \Delta^{r^\star}(x,a,b) \right| \right] + \mathbb{E}_{\widehat{\pi},\pi_{\text{ref}}}\left[ \left| \Delta^{\widehat{r}}(x,a,b) - \Delta^{r^\star}(x,a,b) \right| \right],$$

it follows immediately from Lemma G.1 that $\chi$PO obtains a crude guarantee scaling with all-policy concentrability, i.e. $J(\pi^\star) - J(\widehat{\pi}) \lesssim \sqrt{(\mathcal{C}^{\pi^\star} + \mathcal{C}^{\widehat{\pi}})\varepsilon_{\text{stat}}^2} \leq \sqrt{(\mathcal{C}^{\pi^\star} + \max_{\pi\in\Pi}\mathcal{C}^\pi)\varepsilon_{\text{stat}}^2}$. This inequality is tight for non-pessimistic algorithms like DPO, which reflects their sensitivity to overoptimization. To obtain the improved guarantee for $\chi$PO in Theorem 3.1, which scales only with *single-policy concentrability* $\mathcal{C}^{\pi^\star}$, the crux of the remaining proof will be to show that $\chi$PO implicitly implements pessimism via mixed $\chi^2$-regularization. For this, we appeal to the following central technical lemma, which we expect to find broader use.

**Lemma G.2** (Informal version of Lemma H.2). *Let $f$ be a convex function with $\text{dom}(f) = \mathbb{R}_+$ that is differentiable over its domain. Given any parameter $\beta > 0$ and policy $\bar{\pi} : \mathcal{X} \to \Delta(\mathcal{A})$ with $\bar{\pi}(a \mid x) \in \text{dom}(f')$ for all $x, a$, define the reward model $\bar{r}(x,a) = \beta f'\left( \frac{\bar{\pi}(a|x)}{\pi_{\text{ref}}(a|x)} \right)$. Then*

$$\bar{\pi} \in \underset{\pi}{\arg\max}\, \mathbb{E}_\pi[\bar{r}(x,a)] - \beta \cdot D_f(\pi \| \pi_{\text{ref}}).$$

Under Assumption 3.2 we have $\widehat{\pi} \in \text{dom}(f'_{\chi_{\text{mix}}})$. Then recalling that $\widehat{r}(x,a) := \beta\phi\left( \frac{\widehat{\pi}(a|x)}{\pi_{\text{ref}}(a|x)} \right) = \beta f'_{\chi_{\text{mix}}}\left( \frac{\widehat{\pi}(a|x)}{\pi_{\text{ref}}(a|x)} \right)$ and that $f_{\chi_{\text{mix}}}$ is convex, Lemma G.2 implies that the policy $\widehat{\pi}$ produced by $\chi$PO satisfies

$$\widehat{\pi} \in \underset{\pi\in\Pi}{\arg\max}\, J_{\beta,\widehat{r}}^{\chi_{\text{mix}}}(\pi) := \mathbb{E}_\pi[\widehat{r}] - \beta D_{\chi^2}(\pi \| \pi_{\text{ref}}) - \beta D_{\text{KL}}(\pi \| \pi_{\text{ref}}). \quad (38)$$

In other words,

*The $\chi$PO policy $\widehat{\pi}$ optimizes the mixed $\chi^2$-regularized RLHF objective under its own implicit reward model.*

This formally justifies the claim that $\chi$PO implicitly implements pessimism via $\chi^2$-regularization. With this result in hand, we are now ready to prove Theorem 3.1. Let $\pi^\star$ be an arbitrary policy. Since $J_{\beta,\widehat{r}}^{\chi_{\text{mix}}}(\widehat{\pi}) \geq J_{\beta,\widehat{r}}^{\chi_{\text{mix}}}(\pi^\star)$ by Eq. (38), we can decompose the regret $J(\pi^\star) - J(\widehat{\pi})$ as

$$J(\pi^\star) - J(\widehat{\pi}) \leq J(\pi^\star) - J_{\beta,\widehat{r}}^{\chi_{\text{mix}}}(\pi^\star) + J_{\beta,\widehat{r}}^{\chi_{\text{mix}}}(\widehat{\pi}) - J(\widehat{\pi})$$

$$= \underbrace{J(\pi^\star) - J(\pi_{\mathsf{ref}}) - J_{\beta,\widehat{r}}^{\chi_{\mathsf{mix}}}(\pi^\star) + J_{\beta,\widehat{r}}^{\chi_{\mathsf{mix}}}(\pi_{\mathsf{ref}})}_{\text{(I)}} + \underbrace{J_{\beta,\widehat{r}}^{\chi_{\mathsf{mix}}}(\widehat{\pi}) - J_{\beta,\widehat{r}}^{\chi_{\mathsf{mix}}}(\pi_{\mathsf{ref}}) - J(\widehat{\pi}) + J(\pi_{\mathsf{ref}})}_{\text{(II)}}.$$

In the second line, we have added or subtracted the baselines $J(\pi_{\mathsf{ref}})$ and $J_{\beta,\widehat{r}}^{\chi_{\mathsf{mix}}}(\pi_{\mathsf{ref}})$ to center the objectives with the performance of the reference policy. Up to statistical errors, the first term (I) corresponds to error from how much $J_{\beta,\widehat{r}}^{\chi_{\mathsf{mix}}}(\pi^\star)$ *underestimates* the return of $\pi^\star$ (bias), and the second term (II) corresponds to error from how much $J_{\beta,\widehat{r}}^{\chi_{\mathsf{mix}}}(\widehat{\pi})$ *overestimates* the return of $\widehat{\pi}$ (overoptimization). As we will see shortly, these two sources of error are directly controlled (in opposing ways) by the strength of the regularization parameter $\beta$ in Eq. (38).

First, expanding the definition of $J_{\beta,\widehat{r}}^{\chi_{\mathsf{mix}}}(\pi^\star)$ and centering the returns using the reference policies, we have

$$\text{(I)} = J(\pi^\star) - J_{\beta,\widehat{r}}^{\chi_{\mathsf{mix}}}(\pi^\star) - J(\pi_{\mathsf{ref}}) + J_{\beta,\widehat{r}}^{\chi_{\mathsf{mix}}}(\pi_{\mathsf{ref}})$$

$$= \mathbb{E}_{\pi^\star}[r^\star(x,a)] - \mathbb{E}_{\pi^\star}[\widehat{r}(x,a)] + \beta D_{\chi^2}(\pi^\star \parallel \pi_{\mathsf{ref}}) + \beta D_{\mathsf{KL}}(\pi^\star \parallel \pi_{\mathsf{ref}}) - \mathbb{E}_{\widehat{\pi}}[r^\star(x,a)] + \mathbb{E}_{\pi_{\mathsf{ref}}}[\widehat{r}(x,a)]$$

$$= \mathbb{E}_{\pi^\star,\pi_{\mathsf{ref}}}[\Delta^{r^\star}(x,a,b) - \Delta^{\widehat{r}}(x,a,b)] + \beta D_{\chi^2}(\pi^\star \parallel \pi_{\mathsf{ref}}) + \beta D_{\mathsf{KL}}(\pi^\star \parallel \pi_{\mathsf{ref}})$$

$$\leq \sqrt{(1 + D_{\chi^2}(\pi^\star \parallel \pi_{\mathsf{ref}})) \cdot \varepsilon_{\mathsf{stat}}^2} + \underbrace{\beta \cdot D_{\chi^2}(\pi^\star \parallel \pi_{\mathsf{ref}})}_{\text{bias}}.$$

Above, we have used that $D_{\mathsf{KL}}(\pi \parallel \pi_{\mathsf{ref}}) \leq D_{\chi^2}(\pi \parallel \pi_{\mathsf{ref}})$ for any policy $\pi$, along with the bound on reward estimation error from Lemma G.1. Next, expanding $J_{\beta,\widehat{r}}^{\chi_{\mathsf{mix}}}(\widehat{\pi})$ and centering the returns in a similar fashion,

$$\text{(II)} = J_{\beta,\widehat{r}}^{\chi_{\mathsf{mix}}}(\widehat{\pi}) - J(\widehat{\pi}) - J_{\beta,\widehat{r}}^{\chi_{\mathsf{mix}}}(\pi_{\mathsf{ref}}) + J(\pi_{\mathsf{ref}})$$

$$= \mathbb{E}_{\widehat{\pi},\pi_{\mathsf{ref}}}[\Delta^{\widehat{r}}(x,a,b) - \Delta^{r^\star}(x,a,b)] - \beta D_{\chi^2}(\widehat{\pi} \parallel \pi_{\mathsf{ref}}) - \beta D_{\mathsf{KL}}(\widehat{\pi} \parallel \pi_{\mathsf{ref}})$$

$$\leq \sqrt{(1 + D_{\chi^2}(\widehat{\pi} \parallel \pi_{\mathsf{ref}})) \cdot \varepsilon_{\mathsf{stat}}^2} - \beta \cdot D_{\chi^2}(\widehat{\pi} \parallel \pi_{\mathsf{ref}})$$

$$\lesssim \varepsilon_{\mathsf{stat}} + \underbrace{\beta^{-1} \varepsilon_{\mathsf{stat}}^2}_{\text{overoptimization error}}.$$

Above, the first inequality uses $D_{\mathsf{KL}}(\pi \parallel \pi_{\mathsf{ref}}) \geq 0$ and Lemma G.1, while the second inequality uses AM-GM. Critically, by using $\chi^2$-regularization, we are able to cancel the on-policy error term $\sqrt{(1 + D_{\chi^2}(\widehat{\pi} \parallel \pi_{\mathsf{ref}})) \cdot \varepsilon_{\mathsf{stat}}^2}$ that arises from change-of-measure, leading to a modest $\beta^{-1} \varepsilon_{\mathsf{stat}}^2$ penalty for overoptimization.

Combining these results, and recalling that $\mathcal{C}^\pi = 1 + 2 D_{\chi^2}(\pi \parallel \pi_{\mathsf{ref}})$, we conclude that

$$J(\pi^\star) - J(\widehat{\pi}) \lesssim \sqrt{\mathcal{C}^{\pi^\star} \cdot \varepsilon_{\mathsf{stat}}^2} + \underbrace{\beta \cdot \mathcal{C}^{\pi^\star}}_{\text{bias}} + \underbrace{\beta^{-1} \cdot \varepsilon_{\mathsf{stat}}^2}_{\text{overoptimization error}}.$$

The bias and overoptimization errors above arise from how well our chosen uncertainty quantifier, $\beta D_{\chi^2}(\pi \parallel \pi_{\mathsf{ref}})$, accounts for the on-policy statistical error $\sqrt{(1 + D_{\chi^2}(\pi \parallel \pi_{\mathsf{ref}})) \cdot \varepsilon_{\mathsf{stat}}^2}$ arising from Lemma G.1; this is controlled by the magnitude of the regularization parameter $\beta$. When $\beta$ is too large, the uncertainty quantifier is overly pessimistic about the quality of the reward model $\widehat{r}$ under $\pi^\star$, which increases the *bias* of $\chi$PO. In contrast, the *overoptimization error* increases when $\beta$ is too small. In this regime, $\widehat{\pi}$ overfits to $\widehat{r}$ because the regularizer under-evaluates the statistical error of the learned policy. In order to obtain tight statistical rates, the choice of regularization parameter $\beta$ must carefully balance its opposing effects on bias and overoptimization error. For a fixed $\pi^\star$, choosing $\beta \propto (\varepsilon_{\mathsf{stat}}^2/\mathcal{C}^{\pi^\star})^{1/2}$ results in the second claim in Theorem 3.1.

# H  PROOFS FOR SECTION 3

This section is organized as follows. First, in Appendix H.1, we analyze a more general version of $\chi$PO that mixes KL-regularization with $\chi^2$-regularization using a mixing parameter $\gamma \in (0, 1]$, and present its sample complexity guarantee in Theorem H.1. $\chi$PO is a special case with $\gamma = 1$, and Appendix H.2 shows (with a one-line proof) that Theorem 3.1 follows directly from Theorem H.1 with this parameter choice.

### H.1 GENERAL VERSION OF THEOREM 3.1

As previously described at the end of Section 3.3, $\chi$PO can be applied in a more general form where the KL-regularization is mixed with $\chi^2$-regularization using a weight parameter $\gamma \in (0, 1]$. In this section, we analyze the sample complexity for this form of the algorithm, of which $\chi$PO is a special case with $\gamma = 1$, which directly leads to the guarantee in Theorem 3.1.

Concretely, given regularization parameter $\beta > 0$ and weight parameter $\gamma \in (0, 1]$, we aim to solve the mixed $\chi^2$-regularized objective

$$\underset{\pi:\mathcal{X}\to\Delta(\mathcal{A})}{\mathrm{argmax}}\ J^{\chi_{\mathrm{mix}}}_{\beta,\gamma}(\pi) := \mathbb{E}_\pi[r^\star(x, a)] - \beta \cdot D_{\chi^2}(\pi \,\|\, \pi_{\mathrm{ref}}) - \beta\gamma \cdot D_{\mathrm{KL}}(\pi \,\|\, \pi_{\mathrm{ref}}). \tag{39}$$

The regularization term $D_{\chi^2}(\pi \,\|\, \pi_{\mathrm{ref}}) + \gamma \cdot D_{\mathrm{KL}}(\pi \,\|\, \pi_{\mathrm{ref}}) = D_{f_{\chi_{\mathrm{mix}},\gamma}}(\pi \,\|\, \pi_{\mathrm{ref}})$ is an $f$-divergence induced by the function $f_{\chi_{\mathrm{mix}},\gamma}(z) := \frac{1}{2}(z-1)^2 + \gamma z \log z$. Correspondingly, we replace the link function $\phi(\cdot)$ in $\chi$PO with

$$\phi_\gamma(z) := z + \gamma \log(z),$$

and output the policy

$$\widehat{\pi} \leftarrow \underset{\pi\in\Pi}{\mathrm{argmax}} \sum_{(x,a_+,a_-)\in\mathcal{D}_{\mathrm{pref}}} \log\left[\sigma\left(\mathsf{clip}_{2R_{\mathrm{max}}}\left[\beta\phi_\gamma\left(\frac{\pi(a_+ \mid x)}{\pi_{\mathrm{ref}}(a_+ \mid x)}\right) - \beta\phi_\gamma\left(\frac{\pi(a_- \mid x)}{\pi_{\mathrm{ref}}(a_- \mid x)}\right)\right]\right)\right]. \tag{40}$$

To give a sample complexity guarantee for Eq. (40), we require that $\Pi$ can express the optimal regularized policy for the objective $J^{\chi_{\mathrm{mix}}}_{\beta,\gamma}$ in Eq. (39). This generalizes Assumption 3.1 for $\chi$PO, which corresponds to the special case where $\gamma = 1$.

**Assumption H.1** (Policy realizability). *The policy class $\Pi$ satisfies $\pi^\star_{\beta,\gamma} \in \Pi$, where $\pi^\star_{\beta,\gamma}$ is the optimal policy under mixed $\chi^2$-regularization (Eq. (11)).*

We also assert that, analogous to Assumption 3.2, the "implicit" reward models induced by the policy class $\Pi$ and the link function $\phi_\gamma$ have bounded range.

**Assumption H.2** (Bounded implicit rewards). *For a parameter $V_{\mathrm{max}} \geq R_{\mathrm{max}}$, it holds that for all $\pi \in \Pi$, $x \in \mathcal{X}$, and $a, b \in \mathcal{A}$,*

$$\left|\beta\phi_\gamma\left(\frac{\pi(a \mid x)}{\pi_{\mathrm{ref}}(a \mid x)}\right) - \beta\phi_\gamma\left(\frac{\pi(b \mid x)}{\pi_{\mathrm{ref}}(b \mid x)}\right)\right| \leq V_{\mathrm{max}}. \tag{41}$$

We now state the sample complexity guarantee for the policy learned in Eq. (40). The first bound applies to general $\beta > 0$ and $\gamma \in (0, 1]$, while in the second we obtain a tight statistical rate by choosing the parameter $\beta$ as a function of the comparator policy $\pi^\star$.

**Theorem H.1** (General version of Theorem 3.1). *Suppose Assumptions H.1 and H.2 hold for some $\beta > 0$ and $\gamma \in (0, 1]$. With probability at least $1 - \delta$, the variant of $\chi$PO in Eq. (40) produces a policy $\widehat{\pi}$ such that for all policies $\pi^\star$ simultaneously, we have*

$$J(\pi^\star) - J(\widehat{\pi}) \leq 32V_{\mathrm{max}}e^{2R_{\mathrm{max}}} \cdot \sqrt{\frac{2\mathcal{C}^{\pi^\star}\log(|\Pi|/\delta)}{n}} + \beta(1+\gamma) \cdot \frac{\mathcal{C}^{\pi^\star}}{2} + \beta^{-1} \cdot \frac{256V^2_{\mathrm{max}}e^{4R_{\mathrm{max}}}\log(|\Pi|/\delta)}{n}.$$

*In particular, given any comparator policy $\pi^\star$, we can choose $\beta = 32V_{\mathrm{max}}e^{2R_{\mathrm{max}}}\sqrt{\frac{2\log(|\Pi|/\delta)}{n\mathcal{C}^{\pi^\star}}}$ to achieve*

$$J(\pi^\star) - J(\widehat{\pi}) \leq (64 + 4\gamma)V_{\mathrm{max}}e^{2R_{\mathrm{max}}} \cdot \sqrt{\frac{\mathcal{C}^{\pi^\star}\log(|\Pi|/\delta)}{n}}.$$

The bias-overoptimization tradeoffs induced by the choice of $\beta$ in Theorem H.1 are identical to those for Theorem 3.1 (and described there). Let us briefly discuss the influence of $\gamma$ on the sample complexity. We first observe that choice of $\gamma \in (0, 1]$ changes the bound by only a small multiplicative factor, which implies that $\gamma$ can be arbitrarily small as long as it is positive. For the analysis, this is natural because the KL-divergence is dominated by the $\chi^2$-divergence, and, as discussed in Section 3.2, KL-regularization is only needed to enable the DPO-style reparameterization trick for Eq. (40) (in

particular, the $\chi^2$-RLHF algorithm in Appendix C, which does not require reparameterization, obtains similar guarantees using pure $\chi^2$-regularization). It is worth noting, however, that the $\gamma$ parameter can implicitly influence the magnitude of $V_{\max}$, as well as the policy realizability condition. As such, practical consequences of this hyperparameter choice may not be fully captured by Theorem H.1.

**Proof of Theorem H.1.** Recall that the link function $\phi_\gamma$ induces a correspondence between policies in the class $\Pi$ and the implicit reward functions they induce (or, equivalently, between policies and the Bradley-Terry preference models they express). Our proof centers around the implicit reward model induced by the learned policy $\widehat\pi$,

$$\widehat{r}(x, a) := \beta \cdot \phi_\gamma\left( \frac{\widehat\pi(a \mid x)}{\pi_{\mathsf{ref}}(a \mid x)} \right),$$

which will allow us to move between the $\chi\mathsf{PO}$ objective (Eq. (40)) and the RLHF objective (Eq. (39)). In particular, we establish two key facts, which together show that Eq. (40) implicitly solves Eq. (39):

1. (Lemma H.3) The reward model $\widehat{r}$ is an accurate estimate of $r^\star$ on the distribution of $\pi_{\mathsf{ref}}$. Moreover, we can transfer this guarantee to the distribution of any policy $\pi$ by paying a multiplicative $(1 + 2D_{\chi^2}(\pi \| \pi_{\mathsf{ref}}))$-factor.

2. (Lemma H.2) $\widehat\pi$ maximizes the RLHF objective in Eq. (39) with reward model $\widehat{r}$, namely,

$$\widehat\pi = \operatorname*{argmax}_{\pi \in \Pi} \mathbb{E}_\pi[\widehat{r}(x, a)] - \beta \cdot D_{\chi^2}(\pi \| \pi_{\mathsf{ref}}) - \beta\gamma \cdot D_{\mathsf{KL}}(\pi \| \pi_{\mathsf{ref}}). \tag{42}$$

Establishing these relationships enables us to analyze the $\chi\mathsf{PO}$ policy $\widehat\pi$ defined in Eq. (40) through the RLHF formulation in Eq. (42), allowing us to appeal to pessimism-based arguments to show that $\chi\mathsf{PO}$ is insensitive to overoptimization error that might otherwise be encountered when learning a policy from off-policy data.

**Implicit reward model $\widehat{r}$.** The $\chi\mathsf{PO}$ objective in Eq. (40) is equivalent to maximum likelihood estimation with the Bradley-Terry preference model over the induced reward function class

$$\mathcal{R}_\Pi := \left\{ r(x, a) = \beta \cdot \phi_\gamma\left( \frac{\pi(a \mid x)}{\pi_{\mathsf{ref}}(a \mid x)} \right) : \pi \in \Pi \right\}.$$

Then, since $\widehat\pi$ is the maximizer in Eq. (40), we can equivalently write

$$\widehat{r} = \operatorname*{argmax}_{r \in \mathcal{R}_\Pi} \sum_{(x, a_+, a_-) \in \mathcal{D}_{\mathsf{pref}}} \log \sigma\big(\mathsf{clip}_{2R_{\max}}[r(a_+ \mid x) - r(a_- \mid x)]\big). \tag{43}$$

The following lemma, which builds on a standard MLE generalization bound (Lemma F.1) bounds the error of $\widehat{r}$ under the action distribution induced by $\pi_{\mathsf{ref}}$. Recall that we use $\mathbb{E}_{\pi, \pi'}[\cdot]$ as shorthand for $\mathbb{E}_{x \sim \rho, a \sim \pi(\cdot|x), b \sim \pi'(\cdot|x)}[\cdot]$.

**Lemma H.1.** *Suppose Assumption H.1 holds. Then with probability at least $1 - \delta$, the policy $\widehat\pi$ output by Eq. (40) satisfies*

$$\varepsilon_{\mathrm{stat}}^2 =: \mathbb{E}_{\pi_{\mathsf{ref}}, \pi_{\mathsf{ref}}}\left[ \big(\mathsf{clip}_{2R_{\max}}[\widehat{r}(x, a) - \widehat{r}(x, b)] - \mathsf{clip}_{2R_{\max}}[r^\star(x, a) - r^\star(x, b)]\big)^2 \right] \leq \frac{128 R_{\max}^2 e^{4R_{\max}} \log(|\Pi|/\delta)}{n}.$$

Lemma H.1, along with all further supporting lemmas, is proven in the sequel. This result measures the error of $\widehat{r}$ using the clipped differences of rewards for pairs of actions $(x, a, b)$ drawn from $\pi_{\mathsf{ref}}$. Clipping the range of the implicit/explicit reward functions to $2R_{\max}$ ensures that the statistical error does not depend on $V_{\max}$. One minor but important detail in the proof is showing that Assumption H.1 implies $\mathcal{R}_\Pi$ includes the true reward function $r^\star$ up to an action-independent shift, so that the true preference model is realizable.

**Implicit RLHF policy optimization.** Having established the accuracy of $\widehat{r}$, we now show that Eq. (40) finds the optimal policy to the RLHF objective in Eq. (42) when $\widehat{r}$ is used as the reward model, i.e.,

$$\widehat\pi = \operatorname*{argmax}_{\pi \in \Pi} J_{\beta, \gamma, \widehat{r}}^{\chi_{\mathsf{mix}}}(\pi) := \mathbb{E}_\pi[\widehat{r}(x, a)] - \beta \cdot D_{\chi^2}(\pi \| \pi_{\mathsf{ref}}) - \beta\gamma \cdot D_{\mathsf{KL}}(\pi \| \pi_{\mathsf{ref}}). \tag{44}$$

This is a direct consequence of the result in Lemma H.2, which shows that an analogous property holds for general $f$-divergences. In particular, for any convex function $f$ and policy $\pi$, the policy $\pi$ is itself the optimal solution to the $f$-divergence-regularized RLHF objective under the implicit reward model induced by $\pi$ with the link function $f'$.

**Lemma H.2.** *Let $f : (0, \infty) \to \mathbb{R}$ be a convex function with $f(1) = 0$. Further, $f$ is differentiable almost everywhere and $0 \notin \mathrm{dom}(f')$, where we define $f'(0) := \lim_{x \downarrow 0} \frac{f(x) - f(0)}{x}$ and $f(0) := \lim_{x \downarrow 0} f(x)$. Given any parameter $\beta > 0$ and valid policy $\bar{\pi} : \mathcal{X} \to \Delta(\mathcal{A})$, with $\bar{\pi}(a \mid x) \in \mathrm{dom}(f')$ for all $(x, a)$, let $\bar{r}(x, a) = \beta f'\left(\frac{\bar{\pi}(a|x)}{\pi_{\mathrm{ref}}(a|x)}\right)$ be the implicit reward model. Then*

$$\bar{\pi} \in \operatorname*{argmax}_{\pi : \mathcal{X} \to \Delta(\mathcal{A})} \mathbb{E}_\pi[\bar{r}(x, a)] - \beta D_f(\pi \,\|\, \pi_{\mathrm{ref}}).$$

Since $f'_{\chi_{\mathrm{mix}}, \gamma} = \phi_\gamma = x + \gamma \log x$ for $\gamma > 0$, clearly $0 \notin \mathrm{dom}(\phi_\gamma)$. Further, under Assumption H.2, $\pi(a \mid x) > 0$ for all $\pi \in \Pi$ (otherwise $V_{\mathrm{max}}$ would be undefined), thus $\pi(a \mid x) \in \mathrm{dom}(\phi_\gamma)$ for all $(x, a)$. The claim in Eq. (44) then directly follows.

**Estimation error translation.** To proceed, we will use condition on Lemma H.1 and use the event in this lemma to relate the estimated RLHF objective in Eq. (42) to the "true" RLHF objective that replaces $\hat{r}$ with $r^\star$. An immediate challenge is that the RLHF objective in Eq. (42) must evaluate $\mathbb{E}_\pi[\hat{r}(x, a)]$ for all $\pi \in \Pi$, and accuracy under $\pi_{\mathrm{ref}}$ does not immediately imply that $\hat{r}$ is accurate for other policies. The following bound quantifies the effects of this distribution shift using the $\chi^2$-divergence, and expresses how the estimation guarantee for $\hat{r}$ in Lemma H.1 transfers to other policies $\pi$ of interest.

**Lemma H.3.** *Suppose Assumption 3.1 holds. Then for any $\pi : \mathcal{X} \to \Delta(\mathcal{A})$, under the event in Lemma H.1, we have*

$$\mathbb{E}_{\pi, \pi_{\mathrm{ref}}}[|\hat{r}(x, a) - \hat{r}(x, b) - (r^\star(x, a) - r^\star(x, b))|] \leq \frac{2V_{\mathrm{max}}}{R_{\mathrm{max}}} \cdot \sqrt{\left(1 + 2D_{\chi^2}(\pi \,\|\, \pi_{\mathrm{ref}})\right) \cdot \varepsilon_{\mathrm{stat}}^2},$$

*where $\varepsilon_{\mathrm{stat}}^2$ is the off-policy estimation error defined in Lemma H.1.*

It is worth noting that Lemma H.3 bounds the *unclipped* on-policy estimation error (on the LHS) in terms of the *clipped* off-policy estimation error, and in making this translation we pay for $V_{\mathrm{max}}$. As we will see shortly, working with the unclipped $\hat{r}$ object is necessary for showing that Eq. (40) implicitly optimizes Eq. (42).

**Pessimism-based regret decomposition.** Equipped with the preceding lemmas, we can now bound the regret for $\chi$PO. We decompose the regret using the RLHF objective $J_{\beta, \gamma, \hat{r}}^{\chi_{\mathrm{mix}}}(\pi^\star)$ defined in Eq. (44). Fixing an arbitrary comparator policy $\pi^\star$, we have

$$\begin{aligned}
J(\pi^\star) - J(\hat{\pi}) &= \mathbb{E}_{\pi^\star}[r^\star(x, a)] - \mathbb{E}_{\hat{\pi}}[r^\star(x, a)] \\
&= \mathbb{E}_{\pi^\star}[r^\star(x, a)] - J_{\beta, \gamma, \hat{r}}^{\chi_{\mathrm{mix}}}(\pi^\star) + J_{\beta, \gamma, \hat{r}}^{\chi_{\mathrm{mix}}}(\pi^\star) - \mathbb{E}_{\hat{\pi}}[r^\star(x, a)] \\
&\leq \mathbb{E}_{\pi^\star}[r^\star(x, a)] - J_{\beta, \gamma, \hat{r}}^{\chi_{\mathrm{mix}}}(\pi^\star) + J_{\beta, \gamma, \hat{r}}^{\chi_{\mathrm{mix}}}(\hat{\pi}) - \mathbb{E}_{\hat{\pi}}[r^\star(x, a)],
\end{aligned}$$

where the last inequality uses the optimality of $\hat{\pi}$ for Eq. (44).

Expanding the expression for $J_{\beta, \gamma, \hat{r}}^{\chi_{\mathrm{mix}}}$, we can further bound this by

$$\begin{aligned}
J(\pi^\star) - J(\hat{\pi}) &\leq \mathbb{E}_{\pi^\star}[r^\star(x, a) - \hat{r}(x, a)] + \beta D_{\chi^2}(\pi^\star \,\|\, \pi_{\mathrm{ref}}) + \beta \gamma D_{\mathrm{KL}}(\pi^\star \,\|\, \pi_{\mathrm{ref}}) \\
&\quad + \mathbb{E}_{\hat{\pi}}[\hat{r}(x, a) - r^\star(x, a)] - \beta D_{\chi^2}(\hat{\pi} \,\|\, \pi_{\mathrm{ref}}) - \beta \gamma D_{\mathrm{KL}}(\hat{\pi} \,\|\, \pi_{\mathrm{ref}}) \\
&\leq \mathbb{E}_{\pi^\star}[r^\star(x, a) - \hat{r}(x, a)] + \beta(1 + \gamma) D_{\chi^2}(\pi^\star \,\|\, \pi_{\mathrm{ref}}) \\
&\quad + \mathbb{E}_{\hat{\pi}}[\hat{r}(x, a) - r^\star(x, a)] - \beta D_{\chi^2}(\hat{\pi} \,\|\, \pi_{\mathrm{ref}}). \quad (45)
\end{aligned}$$

In the last line, we use the fact that $0 \leq D_{\mathrm{KL}}(\pi \,\|\, \pi_{\mathrm{ref}}) \leq D_{\chi^2}(\pi \,\|\, \pi_{\mathrm{ref}})$ for any policy $\pi$ to consolidate the $f$-divergence terms. Specifically, this allows us to eliminate $D_{\mathrm{KL}}(\hat{\pi} \,\|\, \pi_{\mathrm{ref}})$, and combine $D_{\mathrm{KL}}(\pi^\star \,\|\, \pi_{\mathrm{ref}})$ and $D_{\chi^2}(\pi^\star \,\|\, \pi_{\mathrm{ref}})$.

In order to bound the reward estimation error terms in Eq. (45) using the guarantee we have previously established (Lemma H.3), we first center them using the return under the reference policy:

$$\begin{aligned}
&\mathbb{E}_{\pi^\star}[r^\star(x, a) - \hat{r}(x, a)] + \mathbb{E}_{\hat{\pi}}[\hat{r}(x, a) - r^\star(x, a)] \\
&= \mathbb{E}_{\pi^\star, \pi_{\mathrm{ref}}}[r^\star(x, a) - \hat{r}(x, a) - r^\star(x, b) + \hat{r}(x, b)] + \mathbb{E}_{\hat{\pi}, \pi_{\mathrm{ref}}}[\hat{r}(x, a) - r^\star(x, a) - \hat{r}(x, b) + r^\star(x, b)] \\
&= \mathbb{E}_{\pi^\star, \pi_{\mathrm{ref}}}\left[\Delta^\star(x, a, b) - \hat{\Delta}(x, a, b)\right] + \mathbb{E}_{\hat{\pi}, \pi_{\mathrm{ref}}}\left[\hat{\Delta}(x, a, b) - \Delta^\star(x, a, b)\right],
\end{aligned}$$

where $\Delta^\star(x, a, b) := r^\star(x, a) - r^\star(x, b)$ and $\widehat{\Delta}(x, a, b) := \widehat{r}(x, a) - \widehat{r}(x, b)$. Substituting this identity back into the regret decomposition in Eq. (45), we apply Lemma H.3 with $\varepsilon_{\text{stat}}^2 :=$ $128 R_{\max}^2 e^{4 R_{\max}} \frac{\log(|\Pi|/\delta)}{n}$ (from Lemma H.1) to obtain

$$J(\pi^\star) - J(\widehat{\pi}) \leq \mathbb{E}_{\pi^\star, \pi_{\text{ref}}} \Big[ \Delta^\star(x, a, b) - \widehat{\Delta}(x, a, b) \Big] + \beta(1 + \gamma) D_{\chi^2}(\pi^\star \| \pi_{\text{ref}})$$

$$+ \mathbb{E}_{\widehat{\pi}, \pi_{\text{ref}}} \Big[ \widehat{\Delta}(x, a, b) - \Delta^\star(x, a, b) \Big] - \beta D_{\chi^2}(\widehat{\pi} \| \pi_{\text{ref}})$$

$$\leq \frac{2 V_{\max}}{R_{\max}} \sqrt{\left(1 + 2 D_{\chi^2}(\pi^\star \| \pi_{\text{ref}})\right) \cdot \varepsilon_{\text{stat}}^2} + \beta(1 + \gamma) D_{\chi^2}(\pi^\star \| \pi_{\text{ref}})$$

$$+ \frac{2 V_{\max}}{R_{\max}} \sqrt{\left(1 + 2 D_{\chi^2}(\widehat{\pi} \| \pi_{\text{ref}})\right) \cdot \varepsilon_{\text{stat}}^2} - \beta D_{\chi^2}(\widehat{\pi} \| \pi_{\text{ref}})$$

$$= \frac{2 V_{\max}}{R_{\max}} \sqrt{\mathcal{C}^{\pi^\star} \cdot \varepsilon_{\text{stat}}^2} + \frac{\beta(1 + \gamma)}{2} \cdot \left(\mathcal{C}^{\pi^\star} - 1\right) + \frac{2 V_{\max}}{R_{\max}} \sqrt{\mathcal{C}^{\widehat{\pi}} \cdot \varepsilon_{\text{stat}}^2} - \frac{\beta}{2} \cdot \left(\mathcal{C}^{\widehat{\pi}} - 1\right)$$

$$\leq \frac{2 V_{\max}}{R_{\max}} \sqrt{\mathcal{C}^{\pi^\star} \cdot \varepsilon_{\text{stat}}^2} + \frac{\beta(1 + \gamma)}{2} \cdot \mathcal{C}^{\pi^\star} + \frac{2 V_{\max}}{R_{\max}} \sqrt{\mathcal{C}^{\widehat{\pi}} \cdot \varepsilon_{\text{stat}}^2} - \frac{\beta}{2} \cdot \mathcal{C}^{\widehat{\pi}},$$

since $\mathcal{C}^\pi = 1 + 2 D_{\chi^2}(\pi \| \pi_{\text{ref}})$, or equivalently $D_{\chi^2}(\pi \| \pi_{\text{ref}}) = \frac{1}{2}(\mathcal{C}^\pi - 1)$. Lastly, we use the AM-GM inequality to upper bound

$$\frac{2 V_{\max}}{R_{\max}} \sqrt{\mathcal{C}^{\widehat{\pi}} \cdot \varepsilon_{\text{stat}}^2} \leq \frac{2 V_{\max}^2 \varepsilon_{\text{stat}}^2}{R_{\max}^2 \beta} + \frac{\beta \mathcal{C}^{\widehat{\pi}}}{2},$$

allowing us to conclude that

$$J(\pi^\star) - J(\widehat{\pi}) \leq \frac{2 V_{\max}}{R_{\max}} \sqrt{\mathcal{C}^{\pi^\star} \cdot \varepsilon_{\text{stat}}^2} + \frac{\beta(1 + \gamma)}{2} \cdot \mathcal{C}^{\pi^\star} + 2 \beta^{-1} \cdot \frac{V_{\max}^2 \varepsilon_{\text{stat}}^2}{R_{\max}^2}.$$

Plugging in the expression for $\varepsilon_{\text{stat}}^2$ results in the first statement of Theorem H.1.

**Choosing $\beta$ for tight rates.** For the second statement, given a comparator policy $\pi^\star$, choosing $\beta = \frac{2 V_{\max}}{R_{\max}} \sqrt{\frac{\varepsilon_{\text{stat}}^2}{\mathcal{C}^{\pi^\star}}}$ gives

$$J(\pi^\star) - J(\widehat{\pi}) \leq \frac{2 V_{\max}}{R_{\max}} \sqrt{\mathcal{C}^{\pi^\star} \cdot \varepsilon_{\text{stat}}^2} + (1 + \gamma) \frac{V_{\max}}{R_{\max}} \sqrt{\mathcal{C}^{\pi^\star} \cdot \varepsilon_{\text{stat}}^2} + \frac{V_{\max}}{R_{\max}} \sqrt{\mathcal{C}^{\pi^\star} \cdot \varepsilon_{\text{stat}}^2}$$

$$= (4 + \gamma) \frac{V_{\max}}{R_{\max}} \sqrt{\mathcal{C}^{\pi^\star} \cdot \varepsilon_{\text{stat}}^2}.$$

$\square$

### H.1.1 PROOFS FOR SUPPORTING LEMMAS

**Proof of Lemma H.1.** Recall the reward-based MLE objective in Eq. (43),

$$\widehat{r} = \underset{r \in \mathcal{R}_\Pi}{\arg\max} \sum_{(x, a_+, a_-) \in \mathcal{D}_{\text{pref}}} \log \sigma \big( \text{clip}_{2 R_{\max}} [r(x, a_+) - r(x, a_-)] \big).$$

To leverage standard generalization bounds for MLE, we re-interpret this objective as maximum likelihood over a class of preference distributions under the Bradley-Terry model. For a reward function $r$, define for all $y \in \{+1, -1\}$ and $(x, a, b) \in \mathcal{X} \times \mathcal{A} \times \mathcal{A}$ its induced preference distribution:

$$P_r(y | x, a, b) = \mathbb{I}\{y = +1\} \cdot \sigma \big( \text{clip}_{2 R_{\max}} [r(x, a) - r(x, b)] \big) + \mathbb{I}\{y = -1\} \cdot \sigma \big( \text{clip}_{2 R_{\max}} [r(x, b) - r(x, a)] \big).$$

Consider the a class of preference models induced by $\mathcal{R}_\Pi$ under this definition, $\mathcal{P}_\Pi := \{P_r : r \in \mathcal{R}_\Pi\}$. We can equivalently write that

$$P_{\widehat{r}} = \underset{p \in \mathcal{P}_\Pi}{\arg\max} \sum_{(x, a_+, a_-) \in \mathcal{D}_{\text{pref}}} \log p(+1 | x, a_+, a_-),$$

or, interpreting each tuple $(x, a_+, a_-)$ in $\mathcal{D}_{\text{pref}}$ as being induced by a tuple $(x, a, \widetilde{a}, y)$ in which $(a_+, a_-) = (a, \widetilde{a})$ if $y = +1$ and $(a_+, a_-) = (\widetilde{a}, a)$ if $y = -1$,

$$P_{\widehat{r}} = \underset{p \in \mathcal{P}_\Pi}{\arg\max} \sum_{(x, a, \widetilde{a}, y) \in \mathcal{D}_{\text{pref}}} \log p(y | x, a, \widetilde{a}).$$

Next, we show that $P_{r^\star} \in \mathcal{P}_\Pi$, ie., the induced preference model class realizes the true distribution. For $\pi^\star_{\beta,\gamma}$, define the reward model

$$\widetilde{r}^\star(x, a) = \phi_\gamma\left(\frac{\pi^\star_{\beta,\gamma}(a \mid x)}{\pi_{\mathsf{ref}}(a \mid x)}\right),$$

which is equivalent to $r^\star$ up to an action-independent shift, namely, the normalization factor $\lambda^\star_{\beta,\gamma}$ in Lemma H.4. Since $\pi^\star_{\beta,\gamma} \in \Pi$ under Assumption H.1, we have $\widetilde{r}^\star \in \mathcal{R}_\Pi$, and for all $(x, a, b) \in \mathcal{X} \times \mathcal{A} \times \mathcal{A}$, it holds that

$$\mathsf{clip}_{2R_{\mathsf{max}}}[\widetilde{r}^\star(x, a) - \widetilde{r}^\star(x, b)] = \mathsf{clip}_{2R_{\mathsf{max}}}[r^\star(x, a) - r^\star(x, b)] = r^\star(x, a) - r^\star(x, b).$$

The first equality is because action-independent shift between $\widetilde{r}^\star$ and $r^\star$ is cancelled out when taking the difference of rewards, and the second equality is because, by assumption, $r^\star \in [0, R_{\mathsf{max}}]$. As a result, the reward difference is bounded in the same range and never clipped.

From this we conclude that $P_{\widetilde{r}^\star} = P_{r^\star} \in \mathcal{P}_\Pi$, and realizability is satisfied. Further, it is easy to see that $\mathcal{P}_\Pi$ contains only valid distributions. Thus, having satisfied the necessary preconditions, we can invoke Lemma F.1, which guarantees that with probability at least $1 - \delta$, we have

$$\mathbb{E}_{\pi_{\mathsf{ref}},\pi_{\mathsf{ref}}}\left[D_{\mathsf{H}}^2(P_{\widehat{r}}(\cdot \mid x, a, b), P_{r^\star}(\cdot \mid x, a, b))\right] \leq \frac{2\log(|\Pi|/\delta)}{n}.$$

To conclude, we extract a bound on reward estimation error from this Hellinger distance bound by using Lemma H.5 with $R = V = 2R_{\mathsf{max}}$, giving

$$\mathbb{E}_{\pi_{\mathsf{ref}},\pi_{\mathsf{ref}}}\left[\left(\mathsf{clip}_{2R_{\mathsf{max}}}[\widehat{r}(x, a) - \widehat{r}(x, b)] - \mathsf{clip}_{2R_{\mathsf{max}}}[r^\star(x, a) - r^\star(x, b)]\right)^2\right]$$
$$\leq 64e^{4R_{\mathsf{max}}}R_{\mathsf{max}}^2 \cdot \mathbb{E}_{\pi_{\mathsf{ref}},\pi_{\mathsf{ref}}}\left[D_{\mathsf{H}}^2(P_{\widehat{r}}(\cdot \mid x, a, b), P_{r^\star}(\cdot \mid x, a, b))\right]$$
$$\leq 128e^{4R_{\mathsf{max}}}R_{\mathsf{max}}^2 \cdot \frac{\log(|\Pi|/\delta)}{n}.$$

$\square$

**Proof of Lemma H.2.**

First we rewrite the objective as a minimization problem,

$$\operatorname*{argmin}_\pi \quad -\mathbb{E}_\pi[\bar{r}(x, a)] + \beta D_f(\pi \,\|\, \pi_{\mathsf{ref}})$$
$$\text{s.t.} \quad \rho(x)\sum_a \pi(a \mid x) = \rho(x) \qquad \forall x,$$
$$\rho(x)\pi(a \mid x) \geq 0 \qquad \forall x, a.$$

Here, $\pi$ is the primal variable, and denote the dual variables as $\lambda : \mathcal{X} \to \mathbb{R}$ and $\alpha : \mathcal{X} \times \mathcal{A} \to [0, \infty)$, which correspond to the first and second constraints, respectively. The Lagrangian form is then

$$\mathcal{L}(\pi, \lambda, \alpha) = -\mathbb{E}_\pi[\bar{r}(x, a)] + \beta D_f(\pi \,\|\, \pi_{\mathsf{ref}}) + \sum_x \rho(x)\lambda(x)\left(\sum_a \pi(a \mid x) - 1\right) - \sum_x \rho(x)\sum_a \alpha(x, a)\pi(a \mid x).$$

Slater's condition holds since $\bar{\pi}$ itself is a strictly feasible solution, and the objective is convex in $\pi(a \mid x)$. Then if $(\pi, \lambda, \alpha)$ satisfy the KKT conditions, they are the optimal primal and dual variables, which, overloading notation, we denote as $(\pi^\star, \lambda^\star, \alpha^\star)$.

We will demonstrate that setting $\pi^\star = \bar{\pi}$, $\lambda^\star = 0$, and $\alpha^\star = 0$ satisfies the KKT conditions. First, we observe that the proposed solutions are primal and dual feasible. Further, we have $\bar{\pi} > 0$ since $0 \notin \mathsf{dom}(f')$ and $\bar{\pi}(a \mid x) \in \mathsf{dom}(f')$. As a result, $\rho(x)\alpha^\star(x, a)\pi(a \mid x) = 0$ for all $x, a$, and complementary slackness is satisfied. Lastly, for stationarity,

$$\frac{\partial \mathcal{L}(\pi, \lambda, \alpha)}{\partial \pi(a \mid x)} = \rho(x)\left(-\bar{r}(x, a) + \beta f'\left(\frac{\bar{\pi}(a \mid x)}{\pi_{\mathsf{ref}}(a \mid x)}\right) + \lambda^\star(x) - \alpha^\star(x, a)\right)$$
$$= \rho(x)\left(-\bar{r}(x, a) + \beta f'\left(\frac{\bar{\pi}(a \mid x)}{\pi_{\mathsf{ref}}(a \mid x)}\right)\right)$$

$$= \rho(x)\left(-\beta f'\left(\frac{\bar{\pi}(a \mid x)}{\pi_{\mathsf{ref}}(a \mid x)}\right) + \beta f'\left(\frac{\bar{\pi}(a \mid x)}{\pi_{\mathsf{ref}}(a \mid x)}\right)\right)$$
$$= 0,$$

where in the second line we substitute $\lambda^\star = 0$ and $\alpha^\star = 0$, and in third line we have utilized the definition of $\bar{r}(x,a)$ from the lemma statement. $\qquad\square$

**Proof of Lemma H.3.** For a pair of policies $\pi, \pi'$ and $p \geq 1$, we define the norm $\|\cdot\|_{p,\pi\times\pi'} := (\mathbb{E}_{\rho,a\sim\pi,b\sim\pi'}[|\cdot|^p])^{1/p}$. In addition, for notational compactness, we abbreviate $\widehat{\Delta}(x,a,b) := \widehat{r}(x,a) - \widehat{r}(x,b)$, and $\Delta^\star(x,a,b) := r^\star(x,a) - r^\star(x,b)$.

Recall that our goal is to bound the (unclipped) reward estimation error under $\pi$ using the (clipped) reward estimation error $\pi_{\mathsf{ref}}$. We begin by decomposing

$$\left\|\Delta^\star - \widehat{\Delta}\right\|_{1,\pi\times\pi_{\mathsf{ref}}} = \left\|\Delta^\star - \mathsf{clip}_{2R_{\max}}\left[\widehat{\Delta}\right] + \mathsf{clip}_{2R_{\max}}\left[\widehat{\Delta}\right] - \widehat{\Delta}\right\|_{1,\pi\times\pi_{\mathsf{ref}}}$$
$$\leq \left\|\Delta^\star - \mathsf{clip}_{2R_{\max}}\left[\widehat{\Delta}\right]\right\|_{1,\pi\times\pi_{\mathsf{ref}}} + \left\|\left(\mathsf{clip}_{2R_{\max}}\left[\widehat{\Delta}\right] - \widehat{\Delta}\right) \cdot \mathbb{I}\left[\mathsf{clip}_{2R_{\max}}\left[\widehat{\Delta}\right] \neq \widehat{\Delta}\right]\right\|_{1,\pi\times\pi_{\mathsf{ref}}}$$
$$\leq \underbrace{\left\|\Delta^\star - \mathsf{clip}_{2R_{\max}}\left[\widehat{\Delta}\right]\right\|_{1,\pi\times\pi_{\mathsf{ref}}}}_{\text{(I) clipped on-policy estimation error}} + \underbrace{V_{\max} \cdot \mathbb{P}_{\pi,\pi_{\mathsf{ref}}}\left(\mathsf{clip}_{2R_{\max}}\left[\widehat{\Delta}\right] \neq \widehat{\Delta}\right)}_{\text{(II) bias from clipping}}.$$

This splits our bound into two terms. The first is the on-policy error of the clipped reward differences, and can be directly bounded by Lemma H.1 using a standard change-of-measure argument. The second expresses the error of translating the clipped estimates to the unclipped ones in our target bound. For the first term, using Cauchy-Schwarz gives

$$\text{(I)} = \left\|\Delta^\star - \mathsf{clip}_{2R_{\max}}\left[\widehat{\Delta}\right]\right\|_{1,\pi\times\pi_{\mathsf{ref}}} \leq \sqrt{\mathcal{C}^\pi \cdot \left\|\Delta^\star - \mathsf{clip}_{2R_{\max}}\left[\widehat{\Delta}\right]\right\|_{2,\pi_{\mathsf{ref}}\times\pi_{\mathsf{ref}}}^2}$$
$$= \sqrt{\mathcal{C}^\pi \cdot \left\|\mathsf{clip}_{2R_{\max}}[\Delta^\star] - \mathsf{clip}_{2R_{\max}}\left[\widehat{\Delta}\right]\right\|_{2,\pi_{\mathsf{ref}}\times\pi_{\mathsf{ref}}}^2},$$

where the last equality uses that $\Delta^\star \in [-R_{\max}, R_{\max}]$.

Next, for the second term, we again use Cauchy-Schwarz to change measure onto the offline distribution,

$$\text{(II)} = V_{\max} \cdot \mathbb{P}_{\pi\times\pi_{\mathsf{ref}}}\left(\mathsf{clip}_{2R_{\max}}\left[\widehat{\Delta}\right] \neq \widehat{\Delta}\right) \leq V_{\max} \cdot \sqrt{\mathcal{C}^\pi \cdot \mathbb{P}_{\pi_{\mathsf{ref}},\pi_{\mathsf{ref}}}\left(\mathsf{clip}_{2R_{\max}}\left[\widehat{\Delta}\right] \neq \widehat{\Delta}\right)}.$$

Further, using Markov's inequality along with the fact that $\Delta^\star \in [-R_{\max}, R_{\max}]$,

$$\mathbb{P}_{\pi_{\mathsf{ref}},\pi_{\mathsf{ref}}}\left(\mathsf{clip}_{2R_{\max}}\left[\widehat{\Delta}\right] \neq \widehat{\Delta}\right) \leq \mathbb{P}_{\pi_{\mathsf{ref}},\pi_{\mathsf{ref}}}\left(\left|\mathsf{clip}_{2R_{\max}}\left[\widehat{\Delta}\right]\right| = 2R_{\max}\right)$$
$$\leq \mathbb{P}_{\pi_{\mathsf{ref}},\pi_{\mathsf{ref}}}\left(\left|\mathsf{clip}_{2R_{\max}}\left[\widehat{\Delta}\right] - \mathsf{clip}_{2R_{\max}}[\Delta^\star]\right| \geq R_{\max}\right)$$
$$\leq \frac{1}{R_{\max}^2}\left\|\mathsf{clip}_{2R_{\max}}\left[\widehat{\Delta}\right] - \mathsf{clip}_{2R_{\max}}[\Delta^\star]\right\|_{2,\pi_{\mathsf{ref}}\times\pi_{\mathsf{ref}}}^2.$$

Combining inequalities, we obtain

$$\left\|\Delta^\star - \widehat{\Delta}\right\|_{1,\pi\times\pi_{\mathsf{ref}}} \leq \left(1 + \frac{V_{\max}}{R_{\max}}\right)\sqrt{\mathcal{C}^\pi \cdot \left\|\mathsf{clip}_{2R_{\max}}\left[\widehat{\Delta}\right] - \mathsf{clip}_{2R_{\max}}[\Delta^\star]\right\|_{2,\pi_{\mathsf{ref}}\times\pi_{\mathsf{ref}}}^2}$$
$$= \left(1 + \frac{V_{\max}}{R_{\max}}\right)\sqrt{\left(1 + 2D_{\chi^2}(\pi \parallel \pi_{\mathsf{ref}})\right) \cdot \varepsilon_{\mathrm{stat}}^2}$$
$$\leq \frac{2V_{\max}}{R_{\max}}\sqrt{\left(1 + 2D_{\chi^2}(\pi \parallel \pi_{\mathsf{ref}})\right) \cdot \varepsilon_{\mathrm{stat}}^2}.$$

In the second line we have used $\mathcal{C}^\pi = 1 + 2D_{\chi^2}(\pi \parallel \pi_{\mathsf{ref}})$ and the definition of $\varepsilon_{\mathrm{stat}}^2$ from Lemma H.1, and in the last line we use $V_{\max} \geq R_{\max}$.

$\qquad\square$

**Lemma H.4.** *When $\pi_{\text{ref}}(a \mid x) > 0$ for all $x \in \mathcal{X}$, the optimal policy $\pi^\star_{\beta,\gamma}$ for Eq. (39) satisfies*

$$r^\star(x, a) = \phi_\gamma\left(\frac{\pi^\star_{\beta,\gamma}(a \mid x)}{\pi_{\text{ref}}(a \mid x)}\right) + \lambda^\star_{\beta,\gamma}(x),$$

*where $\lambda^\star_{\beta,\gamma}$ is an optimal dual variable that normalizes $\pi^\star_{\beta,\gamma}$.*

**Proof of Lemma H.4.** It is easy to see that strong duality holds for Eq. (39), since it is convex and strictly feasible (e.g., for the policy $\pi_{\text{ref}}$). Thus, the KKT conditions give the optimal primal and dual solutions.

Since Eq. (39) is constrained optimization problem (over valid policies), we first define the dual variables. Below, $\lambda : \mathcal{X} \to \mathbb{R}$ corresponds to the equality constraint that $\sum_a \pi(a \mid x) = 1$ for all $x \in \mathcal{X}$, and $\alpha : \mathcal{X} \times \mathcal{A} \to \mathbb{R}_{\geq 0}$ corresponds to the inequality constraint that $\pi(a \mid x) \geq 0$ for all $(x, a) \in \mathcal{X} \times \mathcal{A}$. After converting Eq. (39) from maximization to minimization, we write Eq. (39) in Lagrangian form as

$$\mathcal{L}(\pi, \lambda, \alpha) = -\mathbb{E}_\pi[r^\star(x, a)] + \beta D_{f_{\chi_{\text{mix}},\gamma}}(\pi \parallel \pi_{\text{ref}}) + \sum_x \rho(x)\lambda(x)\left(\sum_a \pi(a \mid x) - 1\right) - \sum_x \rho(x)\sum_a \alpha(x, a)\pi(a \mid x),$$

since multiplying each of the solutions by $\rho(x)$ does not affect the value of the saddle-point problem. We denote the optimal primal variable as $\pi^\star_{\beta,\gamma}$, and optimal dual variables as $(\lambda^\star_{\beta,\gamma}, \alpha^\star_{\beta,\gamma})$.

From stationarity, the optimal primal and dual variables satisfy

$$r^\star(x, a) = \phi_\gamma\left(\frac{\pi^\star_{\beta,\gamma}(a \mid x)}{\pi_{\text{ref}}(a \mid x)}\right) + \lambda^\star_{\beta,\gamma}(x) - \alpha^\star_{\beta,\gamma}(x, a).$$

Next, for a function $g$ let $g^{-1}$ denote its left inverse, such that $g^{-1}(g(x)) = x$. Because $\phi_\gamma$ is injective (see proof of Lemma H.2), it has a left inverse $(\phi_\gamma)^{-1}$, and we can write

$$\pi^\star_{\beta,\gamma}(a \mid x) = \pi_{\text{ref}}(a \mid x) \cdot (\phi_\gamma)^{-1}\left(r^\star(x, a) - \lambda^\star_{\beta,\gamma}(x) + \alpha^\star_{\beta,\gamma}(x, a)\right).$$

Because $\phi_\gamma(z) = z + \gamma \log(z)$, $0 \notin \text{dom}(\phi_\gamma)$, and therefore $0 \notin \text{range}((\phi_\gamma)^{-1})$. Then from the above expression, we observe that $\pi^\star_{\beta,\gamma}(a \mid x) > 0$ since $\pi_{\text{ref}}(a \mid x) > 0$. It immediately follows that $\alpha^\star_{\beta,\gamma}(x, a) = 0$ for all $(x, a)$ from complementary slackness, which states that the optimal solutions satisfy $\pi^\star_{\beta,\gamma}(a \mid x) \cdot \alpha^\star_{\beta,\gamma}(x, a) = 0$ for all $x, a$. This allows us to reduce the expression for $r^\star$ to the stated result, that is,

$$r^\star(x, a) = \phi_\gamma\left(\frac{\pi^\star_{\beta,\gamma}(a \mid x)}{\pi_{\text{ref}}(a \mid x)}\right) + \lambda^\star_{\beta,\gamma}(x).$$

$\square$

**Lemma H.5.** *For $z \in [-R, R]$ and $z' \in [-V, V]$ where $V \geq R \geq 1$, we have*

$$|z - z'| \leq 4e^{2R}V \cdot |\sigma(z) - \sigma(z')|.$$

*Additionally, if we define the distribution $P_z(y) = \mathbb{I}\{y = +1\}\sigma(z) + \mathbb{I}\{y = -1\}\sigma(-z)$ for $y \in \{-1, +1\}$ and define $P_{z'}$ analogously, then*

$$|z - z'| \leq 4e^{2R}V \cdot D_{\text{H}}(P_z, P_{z'}).$$

**Proof of Lemma H.5.** We begin with the first statement, and write

$$|z - z'| = \frac{|z - z'|}{|\sigma(z) - \sigma(z')|} \cdot |\sigma(z) - \sigma(z')|.$$

Since $\sigma(z') \in (0, 1)$ but $z' \in [-V, V]$, it can be observed that the slope $\frac{|z-z'|}{|\sigma(z)-\sigma(z')|}$ is smallest where $z \approx z'$, and increases as we move away from this region in either direction. To better intuit the scaling of the slope in terms of $V$, we expand $|\sigma(z) - \sigma(z')|$ in the denominator to write

$$|z - z'| = \frac{|z - z'|(1 + e^z)(1 + e^{z'})}{|e^z - e^{z'}|} \cdot |\sigma(z) - \sigma(z')|.$$

This indicates that the slope should scale linearly (not exponentially) with the range of $z'$. For example, as $z' \to \infty$, $(1 + e^{z'})/|e^z - e^{z'}| = O(1)$.

To make this intuition precise, we split into two cases. First, whenever $e^{z'} \geq \frac{e^{R+z}+1}{e^R-1}$ or $e^{z'} \leq \frac{e^{R+z}-1}{e^R+1}$ (this constitutes the range where "$z' \approx z$"), we have $1 + e^{z'} \leq e^R |e^z - e^{z'}|$. Then in this region,

$$|z - z'| = \frac{|z - z'|(1 + e^z)(1 + e^{z'})}{|e^z - e^{z'}|} |\sigma(z) - \sigma(z')| \leq 2V(1 + e^R)e^R \cdot |\sigma(z) - \sigma(z')|.$$

Next, for $e^{z'} \in [\frac{e^{R+z}-1}{e^R+1}, \frac{e^{R+z}+1}{e^R-1}]$, we apply the mean value theorem. Since $\sigma'(x) = e^x(1 + e^{-x})^{-2}$,

$$\frac{|z - z'|}{|\sigma(z) - \sigma(z')|} \leq \sup_{\tilde{z} \in [\min\{z,z'\}, \max\{z,z'\}]} e^{\tilde{z}}(1 + e^{-\tilde{z}})^{-2}$$

$$\leq \sup_{e^{\tilde{z}} \in \left[\frac{e^{R+z}-1}{e^R+1}, \frac{e^{R+z}+1}{e^R-1}\right]} e^{\tilde{z}}(1 + e^{-\tilde{z}})^{-2}$$

$$\leq 4e^R.$$

In the second inequality, we use the fact that $e^{z'}, e^z \in [\frac{e^{R+z}-1}{e^R+1}, \frac{e^{R+z}+1}{e^R-1}]$, and in the third inequality we use the fact that $\sigma'(x)$ is increasing in $x$, and that $|z| \leq R$. Combining the inequalities for the two regions of $e^{z'}$ gives the result.

For the second statement, we use the fact that

$$2D_{\mathsf{H}}^2(P_z, P_{z'}) \geq \sum_{y \in \{+1, -1\}} \frac{(P_z(y) - P_{z'}(y))^2}{P_z(y) + P_{z'}(y)}.$$

As a result,

$$\sum_{y \in \{+1, -1\}} (P_z(y) - P_{z'}(y))^2 \leq 4D_{\mathsf{H}}^2(P_z, P_{z'}).$$

Since $P_z(y) = 1 - P_z(-y)$ and $P_z(+1) = \sigma(z)$,

$$\sum_{y \in \{+1, -1\}} (P_z(y) - P_{z'}(y))^2 = 2(\sigma(z) - \sigma(z'))^2,$$

and therefore $(\sigma(z) - \sigma(z'))^2 \leq 2D_{\mathsf{H}}^2(P_z, P_{z'})$. The result follows from taking the square root of both sides and combining with the first statement in the lemma. $\qquad \square$

## H.2 PROOF OF THEOREM 3.1

**Proof of Theorem 3.1.** The policy optimization in Line 2 of Algorithm 1 is a special case of Eq. (40) with $\gamma = 1$. As a result, Theorem 3.1 follows directly from Theorem H.1 when instantiated with $\gamma = 1$. $\qquad \square$

## H.3 PROOF OF COROLLARY 3.1

**Proof of Corollary 3.1.** Recall that for any $\beta > 0$, Theorem 3.1 (Eq. (13)) with the policy class $\Pi_{\mathcal{R}}$ ensures that with probability at least $1 - \delta$, for all $\pi^\star$,

$$J(\pi^\star) - J(\hat{\pi}) \leq c_1 R_{\max} e^{2R_{\max}} \cdot \sqrt{\frac{\mathcal{C}^{\pi^\star} \log(|\mathcal{R}|/\delta)}{n}} + c_2 \beta \mathcal{C}^{\pi^\star} + c_3 \beta^{-1} \frac{R_{\max}^2 e^{4R_{\max}} \log(|\mathcal{R}|/\delta)}{n} \tag{46}$$

for absolute constants $c_1, c_2, c_3 > 0$. Let us invoke this result with

$$\beta^\star = \operatorname*{argmax}_{\beta > 0} \max_{\pi^\star} \left\{ J(\pi^\star) - c_1 R_{\max} e^{2R_{\max}} \cdot \sqrt{\frac{\mathcal{C}^{\pi^\star} \log(|\mathcal{R}|/\delta)}{n}} - c_2 \beta \mathcal{C}^{\pi^\star} - c_3 \beta^{-1} \frac{R_{\max}^2 e^{4R_{\max}} \log(|\mathcal{R}|/\delta)}{n} \right\}.$$

Then Eq. (46) implies that

$$\max_{\pi^\star} \left\{ J(\pi^\star) - c_1 R_{\max} e^{2R_{\max}} \cdot \sqrt{\frac{\mathcal{C}^{\pi^\star} \log(|\mathcal{R}|/\delta)}{n}} - c_2 \beta^\star \mathcal{C}^{\pi^\star} - c_3 (\beta^\star)^{-1} \frac{R_{\max}^2 e^{4R_{\max}} \log(|\mathcal{R}|/\delta)}{n} \right\} - J(\widehat{\pi}) \le 0,$$

so that by the definition of $\beta^\star$,

$$\max_{\beta > 0} \max_{\pi^\star} \left\{ J(\pi^\star) - c_1 R_{\max} e^{2R_{\max}} \cdot \sqrt{\frac{\mathcal{C}^{\pi^\star} \log(|\mathcal{R}|/\delta)}{n}} - c_2 \beta \mathcal{C}^{\pi^\star} - c_3 \beta^{-1} \frac{R_{\max}^2 e^{4R_{\max}} \log(|\mathcal{R}|/\delta)}{n} \right\} - J(\widehat{\pi}) \le 0,$$

or equivalently

$$J(\pi^\star) - J(\widehat{\pi}) \le c_1 R_{\max} e^{2R_{\max}} \cdot \sqrt{\frac{\mathcal{C}^{\pi^\star} \log(|\mathcal{R}|/\delta)}{n}} + c_2 \beta \mathcal{C}^{\pi^\star} + c_3 \beta^{-1} \frac{R_{\max}^2 e^{4R_{\max}} \log(|\mathcal{R}|/\delta)}{n} \quad \forall \pi^\star, \forall \beta > 0.$$

It follows that for all comparator policies $\pi^\star$, we have

$$J(\pi^\star) - J(\widehat{\pi}) \lesssim R_{\max} e^{2R_{\max}} \cdot \sqrt{\frac{\mathcal{C}^{\pi^\star} \log(|\mathcal{R}|/\delta)}{n}}$$

by choosing $\beta \propto \sqrt{\frac{R_{\max}^2 e^{4R_{\max}} \log(|\mathcal{R}|/\delta)}{\mathcal{C}^{\pi^\star} n}}$ above.

$\square$

# I  PROOFS FOR APPENDIX B

**Proof of Proposition B.1.**  To see that $\phi$ and $\phi^{-1}$ are strictly increasing, we note that $\phi'(z) = 1 + \frac{1}{z} > 0$ for all $z > 0$.

We now bound the inverse function $\phi^{-1}$. We will use the fact that $z \mapsto W_0(z)$ is increasing over $z \ge 0$ throughout. We first consider the regime where $z \ge 1$. Since $W_0(\cdot)$ is increasing, we have that $\phi^{-1}(z) = W_0(e^z) \le z$ if and only if $e^z \le ze^z$, which is clearly true for $z \ge 1$. On the other hand, for $c > 0$ we have $\phi^{-1}(z) = W_0(e^z) \ge c \cdot z$ if and only if $e^z \ge cze^{cz}$; setting $c = 1/2$ is clearly sufficient.

We now consider the regime where $z \le 1$. Here, we see that $\phi^{-1}(z) = W(e^z) \le e^z$ if and only if $e^z \le e^z e^{e^z}$, which holds for all $z \in \mathbb{R}$. On the other hand have that $\phi^{-1}(z) = W(e^z) \ge e^{-e} e^z$ if and only if $e^z \ge e^{-e} e^z e^{e^{-e} e^z}$. Since $z \le 1$, we have

$$e^{-e} e^z e^{e^{-e} e^z} \le e^{-e} e^z e^{e^z} \le e^{-e} e^z e^e = e^z,$$

which establishes the result.

$\square$

**Proof of Proposition B.2.** Recall that the optimal policy satisfies

$$r(x, a) = \beta \phi \left( \frac{\pi_\beta^\star(a \mid x)}{\pi_{\mathsf{ref}}(a \mid x)} \right) + Z_{\beta, r}(x), \tag{47}$$

where $Z_{\beta, r}(x)$ is a normalization constant chosen such that $\pi_\beta^\star(\cdot \mid x)$ is a valid probability distribution.

We begin by bounding $Z_{\beta, r}(x)$. We will use that $r(x, a) \in [0, R_{\max}]$. Let $x \in \mathcal{X}$ be fixed. By averaging Eq. (47) over $a \sim \pi_\beta^\star(x)$, we have

$$\mathbb{E}_{a \sim \pi_\beta^\star(x)}[r(x, a)] = \beta \, \mathbb{E}_{a \sim \pi_\beta^\star(x)} \left[ \frac{\pi_\beta^\star(a \mid x)}{\pi_{\mathsf{ref}}(a \mid x)} \right] + \beta D_{\mathsf{KL}} \left( \pi_\beta^\star \, \| \, \pi_{\mathsf{ref}} \right) + Z_{\beta, r}(x) \ge Z_{\beta, r}(x),$$

so $Z_{\beta, r}(x) \le R_{\max}$. On the other hand, averaging over $a \sim \pi_{\mathsf{ref}}(x)$, we have

$$\mathbb{E}_{a \sim \pi_\beta^\star(x)}[r(x, a)] = \beta \, \mathbb{E}_{a \sim \pi_{\mathsf{ref}}(x)} \left[ \frac{\pi_\beta^\star(a \mid x)}{\pi_{\mathsf{ref}}(a \mid x)} \right] - \beta D_{\mathsf{KL}} \left( \pi_{\mathsf{ref}} \, \| \, \pi_\beta^\star \right) + Z_{\beta, r}(x)$$

$$\leq \beta + Z_{\beta,r}(x),$$

so $Z_{\beta,r}(x) \geq -\beta$.

Having established that $Z_{\beta,r}(x) \in [-\beta, R_{\max}]$, we will use that $\phi\left(\frac{\pi_\beta^\star(a|x)}{\pi_{\text{ref}}(a|x)}\right) = \beta^{-1}(r(x,a) - Z_{\beta,r}(x))$, so that our bound on $Z_{\beta,r}$ implies that

$$-\beta^{-1}R_{\max} \leq \phi\left(\frac{\pi_\beta^\star(a \mid x)}{\pi_{\text{ref}}(a \mid x)}\right) \leq 1 + \beta^{-1}R_{\max},$$

or, since $\phi^{-1}$ is increasing,

$$e^{-e} \cdot e^{-\beta^{-1}R_{\max}} \leq \phi^{-1}(-\beta^{-1}R_{\max}) \leq \frac{\pi_\beta^\star(a \mid x)}{\pi_{\text{ref}}(a \mid x)} \leq \phi^{-1}(1 + \beta^{-1}R_{\max}) \leq 1 + \beta^{-1}R_{\max},$$

where we have used that $\phi^{-1}(z) \leq z$ for $z \geq 1$ and $\phi^{-1}(z) \geq e^{z-e}$ for $z \leq 1$ (by Proposition B.1). $\qquad\square$

## J  PROOFS FOR APPENDIX D

### J.1  PROOF OF THEOREM D.1

**Proof of Theorem D.1.** We consider a family of instances in which there is a single context (prompt) $\mathcal{X} = \{\varnothing\}$ and four actions (responses) $\mathcal{A} = \{a, b, c, d\}$. We consider the reference policy $\pi_{\text{ref}}$ given by

$$\pi_{\text{ref}}(a' \mid x) = \begin{cases} \frac{1}{C}, & \text{if } a' = a \text{ or } a' = b, \\ 1 - \frac{2}{C}, & \text{if } a' = c. \end{cases}$$

We consider a preference model class $\mathscr{P} = \{\mathcal{P}^1, \mathcal{P}^2\}$ in which

$$\mathcal{P}^i(a^0 \succ a^1 \mid x) = (1 + \ell^i(x, a^0, a^1))/2$$

for a function $\ell^i(x, a^0, a^1) \in [-1, +1]$. The functions $\ell^1$ and $\ell^2$ are defined as follows (we omit the dependence on $x$, since there is a single context):

$$\ell^1(a^0, a^1) = \ell^2(a^0, a^1) = 0, \quad \forall a^0 \in \mathcal{A}, a^1 \in \{a, b, c\},$$
$$\ell^1(a, d) = 0, \quad \ell^1(b, d) = -1, \quad \ell^1(c, d) = 1$$
$$\ell^2(a, d) = -1, \quad \ell^2(b, d) = 0, \quad \ell^2(c, d) = -1.$$

Note that both functions are skew-symmetric in the sense that $\ell(x, a', a') = 0$ and $\ell(x, a^0, a^1) + \ell(x, a^1, a^0) = 0$ for all $x \in \mathcal{X}$ and $a^0, a^1 \in \mathcal{A}$.

It is straightforward to see that the deterministic policies $\pi_{\text{MW}}^1(x) = a$ and $\pi_{\text{MW}}^2(x) = b$ are minimax winners for $\ell^1$ and $\ell^2$ respectively. Observe that for both policies, we have

$$\mathcal{C}_\infty^{\pi_{\text{MW}}^1} = \mathcal{C}_\infty^{\pi_{\text{MW}}^2} = C.$$

To proceed, we compute duality gap an arbitrary policy $\pi$ under $\mathcal{P}^1$ and $\mathcal{P}^2$. Let $\text{DG}(\pi; \mathcal{P})$ denote the value of $\text{DG}(\pi)$ when $\mathcal{P}$ is the true preference model. Then we have:

$$\max_{q \in \Delta(\mathcal{A})} l(q, \pi) = \max_{q \in \Delta(\mathcal{A})} -q(b)\pi(d) + q(c)\pi(d) + q(d)\pi(b) - q(d)\pi(c),$$
$$\min_{q \in \Delta(\mathcal{A})} l(\pi, q) = \min_{q \in \Delta(\mathcal{A})} -\pi(b)q(d) + \pi(c)q(d) + \pi(d)q(b) - \pi(d)q(c),$$
$$= -\max_{q \in \Delta(\mathcal{A})} -q(b)\pi(d) + q(c)\pi(d) + q(d)\pi(b) - q(d)\pi(c).$$

Therefore we know

$$\text{DG}(\pi; \mathcal{P}^1) = 2 \max_{q \in \Delta(\mathcal{A})} q(d)(\pi(b) - \pi(c)) - \pi(d)(q(b) - q(c))$$

Following similar computations, we have

$$\text{DG}(\pi; \mathcal{P}^2) = 2 \max_{q \in \Delta(\mathcal{A})} q(d)(\pi(a) + \pi(c)) - \pi(d)(q(a) + q(c)).$$

We aim to show that for all policies $\pi$, $\text{DG}(\pi; \mathcal{P}^1) + \text{DG}(\pi; \mathcal{P}^2) \geq \frac{1}{2}$. To do so, we consider two cases. Going forward, we will use that $\text{DG}(\pi; \mathcal{P}^i) \geq 0$.

**Case (1):** $\pi(a) + \pi(c) \geq \frac{1}{2}$. In this case, we have $\mathsf{DG}(\pi; \mathcal{P}^2) \geq \frac{1}{2}$, and thus $\mathsf{DG}(\pi; \mathcal{P}^1) + \mathsf{DG}(\pi; \mathcal{P}^2) \geq \frac{1}{2}$.

**Case (2):** $\pi(a) + \pi(c) < \frac{1}{4}$. In this case, let $\theta := \pi(b) - \pi(c)$. Then we have $\mathsf{DG}(\pi; \mathcal{P}^1) \geq 2 \max\{\theta, \pi(d)\}$. We observe that $\theta + \pi(d) = \pi(b) + \pi(d) - \pi(c) > \frac{3}{4} - \frac{1}{4} = \frac{1}{2}$. This implies that $\mathsf{DG}(\pi; \mathcal{P}^1) > \frac{1}{2}$, and thus $\mathsf{DG}(\pi; \mathcal{P}^1) + \mathsf{DG}(\pi; \mathcal{P}^2) \geq \frac{1}{2}$.

Having established that all $\pi$ satisfy $\mathsf{DG}(\pi; \mathcal{P}^1) + \mathsf{DG}(\pi; \mathcal{P}^2) \geq \frac{1}{2}$ we can apply the Le Cam two-point method (specifically, the variant based on the Bretagnolle-Huber inequality (e.g., Theorem 14.2 in Lattimore and Szepesvári (2020))), which leads to the following inequality

$$\inf_{\text{Alg}} \sup_{\mathcal{P} \in \mathscr{P}} \mathbb{E}_{\mathcal{D}_{\text{pref}}}[\mathsf{DG}(\widehat{\pi}; \mathcal{P})] \geq \frac{1}{8} \exp\left(-n \cdot D_{\mathsf{KL}}\left(\rho \otimes \pi_{\text{ref}} \otimes \pi_{\text{ref}} \otimes \mathcal{P}^1 \,\|\, \rho \otimes \pi_{\text{ref}} \otimes \pi_{\text{ref}} \otimes \mathcal{P}^2\right)\right).$$

It can be observed that $D_{\mathsf{KL}}\left(\rho \otimes \pi_{\text{ref}} \otimes \pi_{\text{ref}} \otimes \mathcal{P}^1 \,\|\, \rho \otimes \pi_{\text{ref}} \otimes \pi_{\text{ref}} \otimes \mathcal{P}^2\right) = 0$, since $\ell^1(a^0, a^1) = \ell^2(a^0, a^1) = 0$ for all $a^0, a^1 \in \{a, b, c\}$, and $\pi_{\text{ref}}$ is supported on $\{a, b, c\}$. We conclude that any policy derived from $\mathcal{D}_{\text{pref}}$ must have

$$\mathbb{E}\left[\mathsf{DG}(\widehat{\pi}; \mathcal{P}^i)\right] \geq \frac{1}{8}$$

for some $i$. □

## J.2 PROOF OF THEOREM D.2

**Proof of Theorem D.2.** Let $\widetilde{\pi}$ be the global best response of $\widehat{\pi}$:

$$\widetilde{\pi} = \operatorname*{argmax}_{\pi \in \Pi} \mathbb{E}_{x \sim \rho, a \sim \pi(x), b \sim \widehat{\pi}(x)}\left[\ell^\star(x, a, b)\right],$$

and let $\widetilde{\pi}_C$ be the best response within $\Pi_C$ of $\widehat{\pi}$ where $C \geq 1$ (recall that $\Pi_C := \{\pi : \max_{x \in \mathcal{X}} D_{\chi^2}(\pi(x) \,\|\, \pi_{\text{ref}}(x)) \leq C\}$ denotes the set of policies with bounded $\chi^2$-divergence w.r.t. $\pi_{\text{ref}}$):

$$\widetilde{\pi}_C = \operatorname*{argmax}_{\pi \in \Pi_C} \mathbb{E}_{x \sim \rho, a \sim \pi(x), b \sim \widehat{\pi}(x)}\left[\ell^\star(x, a, b)\right].$$

Recall that $\overline{r}^t(x, a) := \mathbb{E}_{b \sim \pi^t(x)}[\widehat{\ell}(x, a, b)]$. Then we know

$$\ell^\star(\widetilde{\pi}, \widehat{\pi}) = \mathsf{subopt}(\widehat{\pi}, C) + \underbrace{\frac{1}{T} \sum_{t=1}^{T} \left(\widehat{r}^t(\widetilde{\pi}_C) - \widehat{r}^t(\pi^t)\right)}_{(1)} + \underbrace{\frac{1}{T} \sum_{t=1}^{T} \left(\ell^\star(\widetilde{\pi}_C, \pi^t) - \widehat{\ell}(\widetilde{\pi}_C, \pi^t)\right)}_{(2)}$$

$$+ \underbrace{\frac{1}{T} \sum_{t=1}^{T} (\overline{r}^t(\widetilde{\pi}_C) - \widehat{r}^t(\widetilde{\pi}_C))}_{(3)} + \underbrace{\frac{1}{T} \sum_{t=1}^{T} (\widehat{r}^t(\pi^t) - \overline{r}^t(\pi^t))}_{(4)}, \tag{48}$$

where $r(\pi) := \mathbb{E}_{x \sim \rho, a \sim \pi(x)}[r(x, a)]$. The decomposition utilizes the fact that $\overline{r}^t(\pi^t) = 0$ and $\overline{r}^t(\widetilde{\pi}_C) = \widehat{\ell}(\widetilde{\pi}_C, \pi^t)$. This implies that we only need to bound term (1)(2)(3)(4) in Eq. (48) to upper bound the gap of $\widehat{\pi}$.

**Bounding term (1).** Let $g_x(p)$ to denote the mixed divergence $\beta D_{f_{\chi_{\text{mix}}}}(p(x) \,\|\, \pi_{\text{ref}}(x))$. Then we have the following guarantee on regularized policy mirror descent:

**Lemma J.1.** *For any $C \geq 0$, we have for all policy $\pi \in \Pi_C$ that*

$$\frac{1}{T} \sum_{t=1}^{T} \left(\widehat{r}^t(\pi) - \widehat{r}^t(\pi^t)\right) \leq \frac{2\beta C}{\eta T} + 2\beta C - \frac{1}{T} \sum_{t=1}^{T+1} \mathbb{E}_{x \sim \rho}[g_x(\pi^t)]$$

$$+ \frac{\eta}{2\beta} + \frac{1}{T} \sum_{t=1}^{T} \mathbb{E}_{x \sim \rho}\left[\langle \widehat{r}^t(x, \cdot) - G^t(\pi^{t+1}, x, \cdot), \pi(x) - \pi^{t+1}(x)\rangle\right],$$

*where $G^t(\pi, x, a) := \beta\left((1 + \frac{1}{\eta})\phi\left(\frac{\pi(a|x)}{\pi_{\text{ref}}(a|x)}\right) - \frac{1}{\eta}\phi\left(\frac{\pi^t(a|x)}{\pi_{\text{ref}}(a|x)}\right)\right)$ for all $\pi \in \Pi, x \in \mathcal{X}, a \in \mathcal{A}$.*

To simplify writing, we use $\overline{\pi}^{t+1}$ to denote the minimizer of the following regularized RL objective:

$$\overline{\pi}^{t+1}(x) := \arg \min_{p \in \Delta(\mathcal{X})} \left\langle -\widehat{r}^t(x, \cdot), p \right\rangle + \beta D_{f_{\chi_{\mathsf{mix}}}}(p \,\|\, \pi_{\mathsf{ref}}(x)) + \frac{\beta}{\eta} B_x(p, \pi^t), \qquad \forall x \in \mathcal{X}.$$

Then Assumption D.2 indicates that $\overline{\pi}^{t+1} \in \Pi$ for all $t \in [T]$. In addition, by introducing Lagrangian multipliers into the above optimization problem and following similar arguments in the proof of Lemma H.4, we know

$$f^{\beta, \eta}_{\overline{\pi}^{t+1}, \pi^t}(x, a, b) - (\widehat{r}^t(x, a) - \widehat{r}^t(x, b)) = 0, \qquad \forall x \in \mathcal{X}, a, b \in \mathcal{A}. \tag{49}$$

Recall that by definition $f^{\beta, \eta}_{\pi, \pi^t}(x, a, b) = G^t(\pi, x, a) - G^t(\pi, x, b)$ for all policies $\pi \in \Pi$. This implies that we have

$$\begin{aligned}
&\mathbb{E}_{x \sim \rho}\left[\left\langle \widehat{r}^t(x, \cdot) - G^t(\pi^{t+1}, x, \cdot), \pi(x) - \pi^{t+1}(x) \right\rangle\right] \\
=&\mathbb{E}_{x \sim \rho}\left[\left\langle \widehat{r}^t(x, \cdot) - G^t(\pi^{t+1}, x, \cdot), \pi(x) - \pi_{\mathsf{ref}}(x) \right\rangle\right] + \mathbb{E}_{x \sim \rho}\left[\left\langle \widehat{r}^t(x, \cdot) - G^t(\pi^{t+1}, x, \cdot), \pi_{\mathsf{ref}}(x) - \pi^{t+1}(x) \right\rangle\right] \\
=&\underbrace{(f^{\beta, \eta}_{\overline{\pi}^{t+1}, \pi^t} - f^{\beta, \eta}_{\pi^{t+1}, \pi^t})(\rho, \pi, \pi_{\mathsf{ref}})}_{(5)} + \underbrace{(f^{\beta, \eta}_{\pi^{t+1}, \pi^t} - f^{\beta, \eta}_{\overline{\pi}^{t+1}, \pi^t})(\rho, \pi^{t+1}, \pi_{\mathsf{ref}})}_{(6)},
\end{aligned}$$

where we use $f(\rho, \pi, \pi')$ to denote the expectation $\mathbb{E}_{x \sim \rho, a \sim \pi(x), b \sim \pi'(x)}[f(x, a, b)]$ and the last step utilizes Eq. (49). Therefore, to bound term (1), we need to bound term (5) and (6) respectively. To simplify writing, we define $L(\pi, \pi', \pi'')$ as follows:

$$L(\pi, \pi', \pi'') := \mathbb{E}_{x \sim \rho, a \sim \pi_{\mathsf{ref}}(x), b \sim \pi_{\mathsf{ref}}(x)}\left[\left(\mathsf{clip}_4(f^{\beta, \eta}_{\pi, \pi''}(x, a, b)) - \mathsf{clip}_4(f^{\beta, \eta}_{\pi', \pi''}(x, a, b))\right)^2\right],$$

Note that we have the following guarantee of least squares regression from the literature (Lemma 15 in Song et al. (2022))

**Lemma J.2** (least squares regression). *Let $\{(y_i, z_i)\}_{i=1}^K$ be a dataset of $K$ points where each point are independently sampled from $y_i \sim \mu$ and $z_i \sim p(\cdot|y_i) := h^*(y_i) + \varepsilon_i$. Let $\mathcal{H} : \mathcal{Y} \to [-R, R]$ be a real valued functions where $h^* \in \mathcal{H}$ and $R > 0$. Then if $\{\varepsilon_i\}_{i=1}^K$ are independent random variables such that $\mathbb{E}[z_i|y_i] = h^*(y_i)$, the least squares solution $\widehat{h} = \arg\min_{h \in \mathcal{H}} \sum_{i=1}^K (h(y_i) - z_i)^2$ satisfies with probability at least $1 - \delta$ that*

$$\mathbb{E}_{x \sim \mu}[(\widehat{h}(y) - h^*(y))^2] \lesssim \frac{R^2 \log(|\mathcal{H}|/\delta)}{K}.$$

The proof of the above lemma is omitted. Applying Lemma J.2 to the least sqaures solution $\pi^{t+1}$, we have the following concentration lemma:

**Lemma J.3** (concentration in optimization). *Suppose Assumption D.2 and Assumption D.3 hold. Then with probability at least $1 - \delta/4$, we have for all policy $t \in [T]$ that*

$$L(\pi^{t+1}, \overline{\pi}^{t+1}, \pi^t) \leq \frac{C_{\mathsf{con}} \log(|\Pi|/\delta)}{m} := \varepsilon_{\mathsf{md}}^2,$$

*where $C_{\mathsf{con}} > 0$ is a universal constant.*

In the following discussion, we use $\mathcal{E}_1$ to denote the event in Lemma J.3. Then under $\mathcal{E}_1$, by following the same arguments in the proof of Lemma H.3, we have the following bound on $\|f^{\beta, \eta}_{\overline{\pi}^{t+1}, \pi^t} - f^{\beta, \eta}_{\pi^{t+1}, \pi^t}\|_{1, \pi \times \pi_{\mathsf{ref}}}$:

$$\|f^{\beta, \eta}_{\overline{\pi}^{t+1}, \pi^t} - f^{\beta, \eta}_{\pi^{t+1}, \pi^t}\|_{1, \pi \times \pi_{\mathsf{ref}}} \leq V_{\mathsf{max}} \sqrt{\left(1 + 2D_{\chi^2}(\pi \,\|\, \pi_{\mathsf{ref}})\right) \varepsilon_{\mathsf{md}}^2}, \qquad \forall \pi \in \Pi, t \in [T]. \tag{50}$$

Therefore, with Eq. (50) we know that conditioned on $\mathcal{E}_1$, for any policy $\pi \in \Pi_C$ we have

$$(5) \leq V_{\mathsf{max}} \sqrt{3C\varepsilon_{\mathsf{md}}^2}, \quad (6) \leq V_{\mathsf{max}} \sqrt{\left(1 + 2D_{\chi^2}(\pi^{t+1} \,\|\, \pi_{\mathsf{ref}})\right) \varepsilon_{\mathsf{md}}^2} \leq \frac{V_{\mathsf{max}}^2 \varepsilon_{\mathsf{md}}^2}{\beta} + \frac{1}{2} \mathbb{E}_{x \sim \rho}[g_x(\pi^{t+1})] + V_{\mathsf{max}} \varepsilon_{\mathsf{md}},$$

where we use AM-GM inequality in the last step, the definition of $g_x(\pi) := \beta D_{f_{\chi_{\text{mix}}}}(\pi(\cdot|x) \,\|\, \pi_{\text{ref}}(\cdot|x))$, and $D_{f_{\chi_{\text{mix}}}}(p(x) \,\|\, \pi_{\text{ref}}(x)) \geq D_{\chi^2}(p(x) \,\|\, \pi_{\text{ref}}(x))$ since KL is non-negative

In summary, conditioned on $\mathcal{E}_1$, we have

$$(1) \leq \frac{2\beta C}{\eta T} + 2\beta C - \frac{1}{2T} \sum_{t=1}^{T+1} \mathbb{E}_{x\sim\rho}[g_x(\pi^t)] + \frac{\eta}{2\beta} + V_{\max}\sqrt{4C\varepsilon_{\text{md}}^2} + \frac{V_{\max}^2\varepsilon_{\text{md}}^2}{\beta}. \qquad (51)$$

**Bounding term (2).** From Cauchy-Schwartz's inequality, we have

$$\ell^\star(\widetilde{\pi}_C, \pi^t) - \widehat{\ell}(\widetilde{\pi}_C, \pi^t)$$
$$\leq \sqrt{\mathbb{E}_{x\sim\rho, a\sim\pi_{\text{ref}}(x), b\sim\pi_{\text{ref}}(x)}[(\ell^\star(x,a,b) - \widehat{\ell}(x,a,b))^2]\left(1 + 2D_{\chi^2}(\rho\otimes\widetilde{\pi}_C\otimes\pi^t \,\|\, \rho\otimes\pi_{\text{ref}}\otimes\pi_{\text{ref}})\right)},$$

where $\rho\otimes\pi_1\otimes\pi_2$ denotes the joint distribution of $(x,a,b)$ where $x\sim\rho, a\sim\pi_1(x), b\sim\pi_2(x)$ for all $\pi_1, \pi_2 \in \Pi$. Applying the guarantee of least squares regression (Lemma J.2) to the least squares solution $\widehat{\ell}$, we have under Assumption D.1, with probability at least $1 - \delta/4$, the following event holds:

$$\mathbb{E}_{x\sim\rho, y^0\sim\pi_{\text{ref}}(x), y^1\sim\pi_{\text{ref}}(x)}\left[\left(\widehat{\ell}(x,y^0,y^1) - \ell^\star(x,y^0,y^1)\right)^2\right] \leq O\left(\frac{\ln(|\mathcal{L}|/\delta)}{n}\right) := \varepsilon_{\text{general}}^2. \quad (52)$$

Denote the event in Eq. (52) by $\mathcal{E}_2$. On the other hand, we can obtain that:

$$1 + 2D_{\chi^2}\left(\rho\otimes\widetilde{\pi}_C\otimes\pi^t \,\|\, \rho\otimes\pi_{\text{ref}}\otimes\pi_{\text{ref}}\right) = \sum_x \rho(x)\sum_a \frac{(\widetilde{\pi}_C(a|x))^2}{\pi_{\text{ref}}(a|x)}\sum_b \frac{(\pi^t(b|x))^2}{\pi_{\text{ref}}(b|x)}$$
$$= \sum_x \rho(x)\left(1 + 2D_{\chi^2}(\widetilde{\pi}_C(x) \,\|\, \pi_{\text{ref}}(x))\right)\left(1 + 2D_{\chi^2}(\pi^t(x) \,\|\, \pi_{\text{ref}}(x))\right)$$
$$\leq 6C\left(\mathbb{E}_{x\sim\rho}\left[D_{\chi^2}(\pi^t(x) \,\|\, \pi_{\text{ref}}(x))\right] + 1\right)$$

where the last step is due to $\widetilde{\pi}_C \in \Pi_C$. Therefore, conditioned on $\mathcal{E}_2$, we have

$$\ell^\star(\widetilde{\pi}_C, \pi^t) - \widehat{\ell}(\widetilde{\pi}, \pi^t) \leq \sqrt{6C\mathbb{E}_{x\sim\rho}\left[D_{\chi^2}(\pi^t(x) \,\|\, \pi_{\text{ref}}(x))\right]\varepsilon_{\text{general}}^2} + \sqrt{6C\varepsilon_{\text{general}}^2}$$
$$\leq \frac{1}{2}\mathbb{E}_{x\sim\rho}[g_x(\pi^t)] + \frac{3C\varepsilon_{\text{general}}^2}{\beta} + \sqrt{6C\varepsilon_{\text{general}}^2}.$$

In summary, we have

$$\frac{1}{T}\sum_{t=1}^T \ell^\star(\widetilde{\pi}_C, \pi^t) - \widehat{\ell}(\widetilde{\pi}, \pi^t) \leq \frac{1}{2T}\sum_{t=1}^T \mathbb{E}_{x\sim\rho}[g_x(\pi^t)] + \frac{3C\varepsilon_{\text{general}}^2}{\beta} + \sqrt{6C\varepsilon_{\text{general}}^2}. \qquad (53)$$

**Bounding term (3).** Recall that $\widehat{r}^t(x,a) = \widehat{\ell}(x,a,b_t)$ where $b_t \sim \pi^t(x)$ is an unbiased estimator of $\overline{r}^t$. Fix any policy $\pi \in \Pi$, then from Azuma-Hoeffding's inequality, we have with probability at least $1 - \delta'$ that

$$\left|\sum_{t=1}^T \widehat{r}^t(\pi) - \sum_{t=1}^T \overline{r}^t(\pi)\right| \lesssim \sqrt{T\log(1/\delta')}.$$

By union bound, with probability at least $1 - \delta/4$ we have that for all $\pi \in \Pi$:

$$\left|\sum_{t=1}^T \widehat{r}^t(\pi) - \sum_{t=1}^T \overline{r}^t(\pi)\right| \lesssim \sqrt{T\log(|\Pi|/\delta)}.$$

Therefore, specifically for $\widetilde{\pi}_C$, we have

$$(3) \lesssim \sqrt{\frac{\log(|\Pi|/\delta)}{T}}. \qquad (54)$$

**Bounding term (4).** From Azuma-Hoeffding's inequality, we have with probability at least $1 - \delta/4$ that

$$\left| \sum_{t=1}^{T} \widehat{r}^t(\pi^t) - \sum_{t=1}^{T} \overline{r}^t(\pi^t) \right| \lesssim \sqrt{T \log(1/\delta')}.$$

Therefore, we have

$$(4) \lesssim \sqrt{\frac{\log(1/\delta)}{T}}. \tag{55}$$

**Putting everything together.** Substituting Eq. (51)(53)(54)(55) into (48), we have with probability at least $1 - \delta$ that

$$\ell^\star(\widetilde{\pi}, \widehat{\pi}) \lesssim \mathsf{subopt}(\widehat{\pi}, C) + \frac{C\beta}{\eta T} + C\beta + \frac{\eta}{\beta} + V_{\mathsf{max}} \sqrt{C\varepsilon_{\mathsf{md}}^2} + \frac{V_{\mathsf{max}}^2 \varepsilon_{\mathsf{md}}^2}{2\beta}$$

$$+ \frac{C\varepsilon_{\mathsf{general}}^2}{\beta} + \sqrt{C\varepsilon_{\mathsf{general}}^2} + \sqrt{\frac{\log \frac{|\Pi|}{\delta}}{T}}.$$

By selecting

$$T = \frac{mn}{nV_{\mathsf{max}}^2 + m}, \qquad \beta = \frac{1}{\sqrt{T}}, \qquad \eta = \frac{1}{T},$$

we have with probability at least $1 - \delta$ that

$$\ell^\star(\widetilde{\pi}, \widehat{\pi}) \lesssim \mathsf{subopt}(\widehat{\pi}, C) + C \left( \frac{V_{\mathsf{max}} \log(|\Pi|/\delta)}{\sqrt{m}} + \frac{\log(|\Pi||\mathcal{L}|/\delta)}{\sqrt{n}} \right)$$

Note that due to the skew symmetry of $\ell^\star$, we have:

$$\min_{\pi \in \Pi} \mathbb{E}_{x \sim \rho, a \sim \widehat{\pi}(x), b \sim \pi(x)} \left[ \ell^\star(x, a, b) \right] = -\max_{\pi \in \Pi} \mathbb{E}_{x \sim \rho, a \sim \pi(x), b \sim \widehat{\pi}(x)} \left[ \ell^\star(x, a, b) \right] = -\ell^\star(\widetilde{\pi}, \widehat{\pi}).$$

This implies that $\mathsf{DG}(\widehat{\pi}) \leq 2\ell^\star(\widetilde{\pi}, \widehat{\pi})$, which concludes our proof. $\qquad\square$

### J.3 Proofs for Supporting Lemmas

**Proof of Lemma J.1.** First for all $t \in [T], s \in \mathcal{S}$ and any policy $\pi \in \Pi_C$, we have

$$\langle \eta \widehat{r}^t(x), \pi(x) - \pi^t(x) \rangle + \eta g_x(\pi^t) - \eta g_x(\pi)$$

$$= \langle \eta \widehat{r}^t(x) - (1 + \eta)\nabla g_x(\pi^{t+1}) + \nabla g_x(\pi^t), \pi(x) - \pi^{t+1}(x) \rangle$$

$$+ \underbrace{\langle \nabla g_x(\pi^{t+1}) - \nabla g_x(\pi^t), \pi(x) - \pi^{t+1}(x) \rangle}_{(7)} + \underbrace{\langle \eta \widehat{r}^t(x), \pi^{t+1}(x) - \pi^t(x) \rangle}_{(8)}$$

$$+ \underbrace{\langle \eta \nabla g_x(\pi^{t+1}), \pi(x) - \pi^{t+1}(x) \rangle + \eta g_x(\pi^t) - \eta g_x(\pi)}_{(9)},$$

Note that we have

$$\langle \eta \widehat{r}^t(x) - (1 + \eta)\nabla g_x(\pi^{t+1}) + \nabla g_x(\pi^t), \pi(x) - \pi^{t+1}(x) \rangle = \eta \langle \widehat{r}^t(x, \cdot) - G^t(\pi^{t+1}, x, \cdot), \pi(x) - \pi^{t+1}(x) \rangle$$

Next we bound the term (7)(8)(9) respectively.

**Bounding term (7).** Note that we have the following three point lemma:

**Lemma J.4** (three point lemma). *For any $p_1, p_2, p_3 : \mathcal{X} \mapsto \Delta(\mathcal{Y})$, we have for all $x \in \mathcal{X}$*

$$\frac{1}{\beta} \langle \nabla g_x(p_1) - \nabla g_x(p_2), p_3(x) - p_1(x) \rangle = B_x(p_3, p_2) - B_x(p_3, p_1) - B_x(p_1, p_2).$$

**Proof.** By definition, we know

$$\beta B_x(p, p') = g_x(p) - g_x(p') - \langle \nabla g_x(p'), p - p' \rangle.$$

Substitute the definition into Lemma J.4 and we can prove the lemma. $\qquad\square$

From Lemma J.4, we can rewrite (7) as follows:

$$(7) = \beta \left( B_x(\pi, \pi^t) - B_x(\pi, \pi^{t+1}) - B_x(\pi^{t+1}, \pi^t) \right).$$

**Bounding term (8).** From Cauchy-Schwartz inequality, we have

$$(8) \leq \sum_{a \in \mathcal{A}} \frac{\beta(\pi^{t+1}(a|x) - \pi^t(a|x))^2}{2\pi_{\mathsf{ref}}(a|x)} + \frac{\pi_{\mathsf{ref}}(a|x)\eta^2(\widehat{r}^t(x,a))^2}{2\beta} \leq \beta B_x(\pi^{t+1}, \pi^t) + \frac{\eta^2}{2\beta},$$

where the last step comes from the definition of $B_x$.

**Bounding term (9).** Since $g_x$ is convex, we know

$$\langle \eta \nabla g_x(\pi^{t+1}), \pi - \pi^{t+1} \rangle \leq \eta g_x(\pi) - \eta g_x(\pi^{t+1}).$$

This implies that

$$(3) \leq \eta \left( g_x(\pi^t) - g_x(\pi^{t+1}) \right).$$

In summary, for all $t \in [T]$, $s \in \mathcal{S}$ and any policy $\pi \in \Pi_C$, we have

$$\langle \eta \widehat{r}^t(x), \pi(x) - \pi^t(x) \rangle + \eta g_x(\pi^t) - \eta g_x(\pi) \leq \beta \left( B_x(\pi, \pi^t) - B_x(\pi, \pi^{t+1}) \right)$$
$$+ \eta \left( g_x(\pi^t) - g_x(\pi^{t+1}) \right) + \frac{\eta^2}{2\beta} + \eta \langle \widehat{r}^t(x, \cdot) - G^t(\pi^{t+1}, x, \cdot), \pi(x) - \pi^{t+1}(x) \rangle.$$

This implies that for any policy $\pi \in \Pi_C$:

$$\sum_{t=1}^T \left( \widehat{r}^t(\pi) - \widehat{r}^t(\pi^t) \right) \leq T\mathbb{E}_{x\sim\rho}[g_x(\pi)] - \sum_{t=1}^{T+1} \mathbb{E}_{x\sim\rho}[g_x(\pi^t)] + \frac{\beta}{\eta}\mathbb{E}_{x\sim\rho}\left[ B_x(\pi, \pi^1) \right] + \frac{\eta T}{2\beta}$$

$$+ \sum_{t=1}^T \mathbb{E}_{x\sim\rho}\left[ \langle \widehat{r}^t(x, \cdot) - G^t(\pi^{t+1}, x, \cdot), \pi(x) - \pi^{t+1}(x) \rangle \right]$$

$$\leq 2TC\beta - \sum_{t=1}^{T+1} \mathbb{E}_{x\sim\rho}[g_x(\pi^t)] + \frac{2C\beta}{\eta} + \frac{\eta T}{2\beta}$$

$$+ \sum_{t=1}^T \mathbb{E}_{x\sim\rho}\left[ \langle \widehat{r}^t(x, \cdot) - G^t(\pi^{t+1}, x, \cdot), \pi(x) - \pi^{t+1}(x) \rangle \right]$$

Here the last step uses the fact that $B_x(\cdot, \pi_{\mathsf{ref}}) = \frac{1}{\beta}g_x(\cdot)$ and $\pi \in \Pi_C$. This concludes our proof. $\qquad\square$

**Proof of Lemma J.3.** Let $\widehat{L}(\pi, \pi', \pi'')$ denote the empirical squared loss:

$$\widehat{L}(\pi, \pi', \pi'') := \sum_{(\overline{x},\overline{a},\overline{b})} \left( \mathsf{clip}_4(f_{\pi,\pi''}^{\beta,\eta}(\overline{x},\overline{a},\overline{b})) - \mathsf{clip}_4(f_{\pi',\pi''}^{\beta,\eta}(\overline{x},\overline{a},\overline{b})) \right)^2.$$

Fix any $\pi', \pi'' \in \Pi$ and consider the following LSR problems:

$$\pi(\pi', \pi'') := \operatorname*{argmin}_{\pi \in \Pi} \widehat{L}(\pi, \pi', \pi'').$$

Then from Lemma J.2, we know with probability at least $1 - \delta'$ that

$$L(\pi(\pi', \pi''), \pi', \pi'') \lesssim \frac{\log(|\Pi|/\delta')}{M}.$$

Therefore, by union bound, we know with probability at least $1 - \delta'$ that for all $\pi', \pi'' \in \Pi$:

$$L(\pi(\pi', \pi''), \pi', \pi'') \lesssim \frac{\log(|\Pi|/\delta')}{M}.$$

The proof is concluded by noticing that $\pi^{t+1} = \operatorname{argmin}_{\pi\in\Pi} \widehat{L}(\pi, \overline{\pi}^{t+1}, \pi^t)$ under Assumption D.2. $\qquad\square$

## K   PROOFS FOR APPENDIX C

The section contains the proofs for the main guarantee $\chi^2$-RLHF in Appendix C (Theorem C.1). We first prove two results, Theorem K.1 and Corollary K.1, which correspond to exact (i.e., including precise constants) versions of the two statements in Theorem C.1. We also analyze $\chi^2$-RLHF with $\eta = 0$ in Corollary K.2.

Throughout this section, we make use of the following $\eta$-smoothed version of the $L_1$ concentrability coefficient:

$$\mathcal{C}_\eta^\pi := \mathbb{E}_\pi \left[ \frac{\pi(a \mid x)}{\pi_{\mathsf{ref}}(a \mid x) + \eta \pi(a \mid x)} \right].$$

It is easy to see that for any $\eta \geq 0$ we have $\mathcal{C}_\eta^\pi \leq \mathcal{C}^\pi$, as well as $\mathcal{C}_\eta^\pi \leq \eta^{-1}$.

**Theorem K.1** (General regret bound for Algorithm 2). *Suppose Assumption C.1 and Assumption C.2 hold for parameters $\beta > 0$ and $\eta \in \left[ 0, \frac{\beta}{8R_{\max}} \right]$. Then with probability at least $1 - \delta$, the policy $\widehat{\pi}$ produced by $\chi^2$-RLHF (Algorithm 2) satisfies*

$$J(\pi^\star) - J(\widehat{\pi}) \leq 2\sqrt{\mathcal{C}_\eta^{\pi^\star} \cdot \varepsilon_{\mathrm{stat}}^2} + 2\beta \cdot \mathcal{C}_\eta^{\pi^\star} + 4\beta^{-1} \cdot \varepsilon_{\mathrm{stat}}^2$$

$$+ 4\beta \cdot \left( \min\left\{ \mathcal{C}_\infty^{\pi^\star}, \eta^{-1} \right\} + \min\left\{ \max_{\pi \in \Pi} \mathcal{C}_\infty^\pi, \eta^{-1} \right\} \right) \varepsilon_{\mathsf{x}}^2 + 2R_{\max}\varepsilon_{\mathsf{x}}.$$

*where $\varepsilon_{\mathrm{stat}}^2 = \frac{32 R_{\max}^2 e^{4R_{\max}} \log(3|\mathcal{R}|/\delta)}{n}$ and $\varepsilon_{\mathsf{x}} = \sqrt{\frac{\log(3|\Pi|/\delta)}{2n_{\mathsf{x}}}}$.*

The following results are immediate consequences of Theorem K.1.

**Corollary K.1** (Smoothed $\chi^2$-regularization). Given $\pi^\star$, let $\eta = \frac{\beta}{8R_{\max}}$ and $\beta = 2\sqrt{\frac{32 R_{\max}^2 e^{4R_{\max}} \log(3|\mathcal{R}|/\delta)}{n \mathcal{C}^{\pi^\star}}}$. Then under the preconditions of Theorem K.1, with probability at least $1 - \delta$, the policy $\widehat{\pi}$ produced by $\chi^2$-RLHF (Algorithm 2) satisfies

$$J(\pi^\star) - J(\widehat{\pi}) \leq 20 R_{\max} e^{2R_{\max}} \sqrt{\frac{2\mathcal{C}^{\pi^\star} \log(3|\mathcal{R}|/\delta)}{n}} + R_{\max}\sqrt{\frac{2\log(3|\Pi|/\delta)}{n_{\mathsf{x}}}} + \frac{32 R_{\max} \log(3|\Pi|/\delta)}{n_{\mathsf{x}}}.$$

**Corollary K.2** (Non-smoothed $\chi^2$-regularization). Given $\pi^\star$, let $\eta = 0$ and $\beta = 2\sqrt{\frac{32 R_{\max}^2 e^{4R_{\max}} \log(3|\mathcal{R}|/\delta)}{n \mathcal{C}^{\pi^\star}}}$. Then under the preconditions of Theorem K.1, with probability at least $1 - \delta$, the policy $\widehat{\pi}$ produced by $\chi^2$-RLHF (Algorithm 2) satisfies

$$J(\pi^\star) - J(\widehat{\pi}) \leq 20 R_{\max} e^{2R_{\max}} \sqrt{\frac{2\mathcal{C}^{\pi^\star} \log(3|\mathcal{R}|/\delta)}{n}} + R_{\max}\sqrt{\frac{2\log(3|\Pi|/\delta)}{n_{\mathsf{x}}}}$$

$$+ 32 \left( \mathcal{C}_\infty^{\pi^\star} + \max_{\pi \in \Pi} \mathcal{C}_\infty^\pi \right) \cdot \frac{\log(3|\Pi|/\delta)}{n_{\mathsf{x}}} \cdot \sqrt{\frac{2\log(3|\mathcal{R}|/\delta)}{n}}.$$

**Proof of Theorem K.1.**   The proof follows largely the same lines of analyses as the proof of Theorem H.1. One difference is that in Algorithm 2, we approximate the RLHF objective using contexts are sampled from $\mathcal{D}_{\mathsf{x}}$, so we require additional concentration arguments to show that the empirical objective approximates its population counterpart.

**Basic concentration results.**   We begin by stating the two concentration inequalities, which, given the reward model $\widehat{r}$ produced in Eq. (26), bound the error between $\widehat{J}_{\beta,\eta}^{\widehat{r}}$ and its the population version $J_{\beta,\eta}^{\widehat{r}}$.

We will handle the return and regularization terms separately, which will later allow us to obtain tighter bounds. Define

$$\widehat{J}(\pi) := \frac{1}{n_{\mathsf{x}}} \sum_{x \in \mathcal{D}_{\mathsf{x}}} \mathbb{E}_\pi[\widehat{r}(x, a) \mid x],$$

and

$$\widehat{\mathcal{C}}_\eta^\pi(\pi) := \frac{1}{n_\mathsf{x}} \sum_{x \in \mathcal{D}_\mathsf{x}} \mathbb{E}_\pi \left[ \sum_a \frac{\pi^2(a \mid x)}{\pi_{\mathsf{ref}}(a \mid x) + \eta\pi(a \mid x)} \;\Big|\; x \right],$$

so that $\widehat{J}_{\beta,\eta}^{\widehat{r}}(\pi) = \widehat{J}(\pi) - \beta\widehat{\mathcal{C}}_\eta^\pi(\pi)$.

Fix $\delta' \in (0, 1]$, which we will specify at the end of this proof. Since $\max_x \mathbb{E}_\pi[\widehat{r}(x, a) \mid x] \leq R_{\mathsf{max}}$, a straightforward application of Hoeffding's inequality guarantees that with probability at most $1 - \delta'$, for all $\pi \in \Pi$ we have that

$$\left| \widehat{J}(\pi) - \mathbb{E}_\pi[\widehat{r}(x, a)] \right| \leq R_{\mathsf{max}} \sqrt{\frac{\log(2|\Pi|/\delta')}{2n_\mathsf{x}}}. \tag{56}$$

Next, we consider the regularization term. Since $\sum_a \frac{\pi^2(a|x)}{\pi_{\mathsf{ref}}(a|x)+\eta\pi(a|x)} \leq \min\{\mathcal{C}_\infty^\pi, \eta^{-1}\}$ for any $x \in \mathcal{X}$, we use Bernstein's inequality to derive the following result.

**Lemma K.1.** *With probability at least $1 - \delta$, for any $\pi \in \Pi$, we have*

$$\left| \widehat{\mathcal{C}}_\eta^\pi - \mathcal{C}_\eta^\pi \right| \leq \frac{\mathcal{C}^\pi}{2} + \frac{2\min\{\mathcal{C}_\infty^\pi, \eta^{-1}\}\log(2|\Pi|/\delta)}{n_\mathsf{x}}.$$

Define $\varepsilon_\mathsf{x} := \sqrt{\frac{\log(2|\Pi|/\delta')}{2n_\mathsf{x}}}$. The above lemma implies that for all $\pi \in \Pi$, we have

$$\widehat{\mathcal{C}}_\eta^\pi \leq \frac{3\mathcal{C}^\pi}{2} + 4\min\{\mathcal{C}_\infty^\pi, \eta^{-1}\} \cdot \varepsilon_\mathsf{x}^2, \quad \text{and} \quad \widehat{\mathcal{C}}_\eta^\pi \geq \frac{\mathcal{C}^\pi}{2} - 4\min\{\mathcal{C}_\infty^\pi, \eta^{-1}\} \cdot \varepsilon_\mathsf{x}^2.$$

Together with Eq. (56), this implies that for all $\pi \in \Pi$,

$$\widehat{J}_{\beta,\eta}^{\widehat{r}}(\pi) = \widehat{J}(\pi) - \beta\widehat{\mathcal{C}}_\eta^\pi \leq \mathbb{E}_\pi[\widehat{r}(x, a)] - \frac{\beta\mathcal{C}_\eta^\pi}{2} + 4\beta\min\{\mathcal{C}_\infty^\pi, \eta^{-1}\}\varepsilon_\mathsf{x}^2 + R_{\mathsf{max}}\varepsilon_\mathsf{x}, \tag{57}$$

and

$$\widehat{J}_{\beta,\eta}^{\widehat{r}}(\pi) = \widehat{J}(\pi) - \beta\widehat{\mathcal{C}}_\eta^\pi \geq \mathbb{E}_\pi[\widehat{r}(x, a)] - \frac{3\beta\mathcal{C}_\eta^\pi}{2} - 4\beta\min\{\mathcal{C}_\infty^\pi, \eta^{-1}\}\varepsilon_\mathsf{x}^2 - R_{\mathsf{max}}\varepsilon_\mathsf{x}. \tag{58}$$

**Estimation error bounds.** Next, we state the following off- and on-policy reward estimation error bounds for the reward model $\widehat{r}$, analogous to Lemma H.1 and Lemma H.3 for $\chi$PO.

**Lemma K.2.** *Suppose Assumption C.1 holds. Then with probability at least $1 - \delta$, the reward model $\widehat{r}$ learned in Eq. (26) satisfies*

$$\varepsilon_{\mathsf{stat}}^2 =: \mathbb{E}_{\pi_{\mathsf{ref}}, \pi_{\mathsf{ref}}} \left[ ((\widehat{r}(x, a) - \widehat{r}(x, b)) - (r^\star(x, a) - r^\star(x, b)))^2 \right] \leq \frac{32R_{\mathsf{max}}^2 e^{4R_{\mathsf{max}}} \log(|\Pi|/\delta)}{n}.$$

**Lemma K.3.** *Under the event in Lemma K.2, we have that for all $\pi : \mathcal{X} \to \Delta(\mathcal{A})$,*

$$\mathbb{E}_{\pi, \pi_{\mathsf{ref}}}[|(\widehat{r}(x, a) - \widehat{r}(x, b)) - (r^\star(x, a) - r^\star(x, b))|] \leq 2\sqrt{\mathcal{C}_\eta^\pi \varepsilon_{\mathsf{stat}}^2} + 2\mathcal{C}_\eta^\pi R_{\mathsf{max}}\eta,$$

*where $\varepsilon_{\mathsf{stat}}^2$ is defined in Lemma K.2.*

**Regret decomposition.** Equipped with these concentration and estimation error bounds, we now bound the regret of Algorithm 2 using a pessimism-based analysis similar to the proof of Theorem H.1. Condition on the events in Eq. (56), Lemma K.1, and Lemma K.2, which hold together with probability at least $1 - 3\delta'$. We decompose the regret of $\widehat{\pi}$ using $\widehat{J}_{\beta,\eta}^{\widehat{r}}$, then leverage the inequalities in Eq. (57) and Eq. (58):

$$\begin{aligned}
J(\pi^\star) - J(\widehat{\pi}) &= J(\pi^\star) - \widehat{J}_{\beta,\eta}^{\widehat{r}}(\pi^\star) + \widehat{J}_{\beta,\eta}^{\widehat{r}}(\pi^\star) - J(\widehat{\pi}) \\
&\leq J(\pi^\star) - \widehat{J}_{\beta,\eta}^{\widehat{r}}(\pi^\star) + \widehat{J}_{\beta,\eta}^{\widehat{r}}(\widehat{\pi}) - J(\widehat{\pi})
\end{aligned}$$

$$\leq J(\pi^\star) - \mathbb{E}_{\pi^\star}[\widehat{r}(x,a)] + \frac{3\beta \mathcal{C}_\eta^{\pi^\star}}{2} + 4\beta \min\{\mathcal{C}_\infty^{\pi^\star}, \eta^{-1}\}\varepsilon_{\mathrm{x}}^2 + R_{\mathsf{max}}\varepsilon_{\mathrm{x}}$$

$$+ \mathbb{E}_{\widehat{\pi}}[\widehat{r}(x,a)] - \frac{\beta \mathcal{C}_\eta^{\widehat{\pi}}}{2} + 4\beta \min\{\mathcal{C}_\infty^{\widehat{\pi}}, \eta^{-1}\}\varepsilon_{\mathrm{x}}^2 + R_{\mathsf{max}}\varepsilon_{\mathrm{x}} - J(\widehat{\pi})$$

$$= \mathbb{E}_{\pi^\star, \pi_{\mathrm{ref}}}[\Delta^\star(x,a,b) - \widehat{\Delta}(x,a,b)] + \frac{3\beta \mathcal{C}_\eta^{\pi^\star}}{2} + \mathbb{E}_{\widehat{\pi}, \pi_{\mathrm{ref}}}[\widehat{\Delta}(x,a,b) - \Delta^\star(x,a,b)] - \frac{\beta \mathcal{C}_\eta^{\widehat{\pi}}}{2}$$

$$+ 4\beta \varepsilon_{\mathrm{x}}^2 \Big( \min\{\mathcal{C}_\infty^{\pi^\star}, \eta^{-1}\} + \min\{\mathcal{C}_\infty^{\widehat{\pi}}, \eta^{-1}\} \Big) + 2R_{\mathsf{max}}\varepsilon_{\mathrm{x}}.$$

In the last line above, we have introduced the notation $\Delta^\star(x,a,b) = r^\star(x,a) - r^\star(x,b)$ and $\widehat{\Delta}(x,a,b) = \widehat{r}(x,a) - \widehat{r}(x,b)$, and centered the returns. Next, applying Lemma K.3 to bound the reward estimation error above, we have

$$J(\pi^\star) - J(\widehat{\pi}) \leq 2\sqrt{\mathcal{C}_\eta^{\pi^\star} \varepsilon_{\mathrm{stat}}^2} + 2\eta R_{\mathsf{max}} \mathcal{C}_\eta^{\pi^\star} + \frac{3\beta \mathcal{C}_\eta^{\pi^\star}}{2}$$

$$+ 2\sqrt{\mathcal{C}_\eta^{\widehat{\pi}} \varepsilon_{\mathrm{stat}}^2} + 2\eta R_{\mathsf{max}} \mathcal{C}_\eta^{\widehat{\pi}} - \frac{\beta \mathcal{C}_\eta^{\widehat{\pi}}}{2}$$

$$+ 4\beta \varepsilon_{\mathrm{x}}^2 \Big( \min\{\mathcal{C}_\infty^{\pi^\star}, \eta^{-1}\} + \min\{\mathcal{C}_\infty^{\widehat{\pi}}, \eta^{-1}\} \Big) + 2R_{\mathsf{max}}\varepsilon_{\mathrm{x}}.$$

Applying the AM-GM inequality to $2\sqrt{\mathcal{C}_\eta^{\widehat{\pi}} \varepsilon_{\mathrm{stat}}^2}$ for $\eta \in \left[0, \frac{\beta}{4R_{\mathsf{max}}}\right]$, we have

$$2\sqrt{\mathcal{C}_\eta^{\widehat{\pi}} \varepsilon_{\mathrm{stat}}^2} = \sqrt{(\beta - 4\eta R_{\mathsf{max}})\mathcal{C}_\eta^{\widehat{\pi}} \cdot \frac{4\varepsilon_{\mathrm{stat}}^2}{(\beta - 4\eta R_{\mathsf{max}})}}$$

$$\leq \frac{\beta \mathcal{C}_\eta^{\widehat{\pi}}}{2} - 2\eta R_{\mathsf{max}} \mathcal{C}_\eta^{\widehat{\pi}} + \frac{2\varepsilon_{\mathrm{stat}}^2}{\beta - 4\eta R_{\mathsf{max}}}$$

$$\leq \frac{\beta \mathcal{C}_\eta^{\widehat{\pi}}}{2} - 2\eta R_{\mathsf{max}} \mathcal{C}_\eta^{\widehat{\pi}} + \frac{4\varepsilon_{\mathrm{stat}}^2}{\beta},$$

where in the last line we use the fact that $\eta \leq \frac{\beta}{8R_{\mathsf{max}}}$ so $4\eta R_{\mathsf{max}} \leq \frac{\beta}{2}$. Then plugging this back into our regret decomposition cancels out the $\mathcal{C}_\eta^{\widehat{\pi}}$ terms to give

$$J(\pi^\star) - J(\widehat{\pi}) \leq 2\sqrt{\mathcal{C}_\eta^{\pi^\star} \varepsilon_{\mathrm{stat}}^2} + 2\eta R_{\mathsf{max}} \mathcal{C}_\eta^{\pi^\star} + \frac{3\beta \mathcal{C}_\eta^{\pi^\star}}{2} + \frac{4\varepsilon_{\mathrm{stat}}^2}{\beta}$$

$$+ 4\beta \varepsilon_{\mathrm{x}}^2 \Big( \min\{\mathcal{C}_\infty^{\pi^\star}, \eta^{-1}\} + \min\{\mathcal{C}_\infty^{\widehat{\pi}}, \eta^{-1}\} \Big) + 2R_{\mathsf{max}}\varepsilon_{\mathrm{x}}$$

$$\leq 2\sqrt{\mathcal{C}_\eta^{\pi^\star} \varepsilon_{\mathrm{stat}}^2} + 2\beta \mathcal{C}_\eta^{\pi^\star} + \frac{4\varepsilon_{\mathrm{stat}}^2}{\beta}$$

$$+ 4\beta \varepsilon_{\mathrm{x}}^2 \Big( \min\{\mathcal{C}_\infty^{\pi^\star}, \eta^{-1}\} + \min\{\mathcal{C}_\infty^{\widehat{\pi}}, \eta^{-1}\} \Big) + 2R_{\mathsf{max}}\varepsilon_{\mathrm{x}},$$

where in the last line we consolidate $\mathcal{C}_\eta^{\pi^\star}$ terms by again using $4\eta R_{\mathsf{max}} \leq \frac{\beta}{2}$. Plugging in $\delta' = \delta/3$ and the values for $\varepsilon_{\mathrm{stat}}^2$ and $\varepsilon_{\mathrm{x}}$ results in the theorem statement.

$\square$

**Proof of Corollary K.1.** When $\eta = \frac{\beta}{8R_{\mathsf{max}}}$, Theorem K.1 states that

$$J(\pi^\star) - J(\widehat{\pi}) \leq 2\sqrt{\mathcal{C}_\eta^{\pi^\star} \varepsilon_{\mathrm{stat}}^2} + 2\beta \mathcal{C}_\eta^{\pi^\star} + \frac{4\varepsilon_{\mathrm{stat}}^2}{\beta} + 4\beta \varepsilon_{\mathrm{x}}^2 \cdot \left( \min\left\{\mathcal{C}_\infty^{\pi^\star}, \eta^{-1}\right\} + \min\left\{\max_{\pi \in \Pi} \mathcal{C}_\infty^{\pi}, \eta^{-1}\right\} \right) + 2R_{\mathsf{max}}\varepsilon_{\mathrm{x}}$$

$$\leq 2\sqrt{\mathcal{C}_\eta^{\pi^\star} \varepsilon_{\mathrm{stat}}^2} + 2\beta \mathcal{C}_\eta^{\pi^\star} + \frac{4\varepsilon_{\mathrm{stat}}^2}{\beta} + 8\beta \varepsilon_{\mathrm{x}}^2 \cdot \eta^{-1} + 2R_{\mathsf{max}}\varepsilon_{\mathrm{x}}$$

$$= 2\sqrt{\mathcal{C}_\eta^{\pi^\star} \varepsilon_{\mathrm{stat}}^2} + 2\beta \mathcal{C}_\eta^{\pi^\star} + \frac{4\varepsilon_{\mathrm{stat}}^2}{\beta} + 64 R_{\mathsf{max}} \varepsilon_{\mathrm{x}}^2 + 2R_{\mathsf{max}}\varepsilon_{\mathrm{x}}.$$

Setting $\beta = 2\sqrt{\frac{\varepsilon_{\text{stat}}^2}{\mathcal{C}^{\pi^\star}}}$, we obtain

$$J(\pi^\star) - J(\widehat{\pi}) \leq 5\sqrt{\mathcal{C}_\eta^{\pi^\star}\varepsilon_{\text{stat}}^2} + 64R_{\max}\varepsilon_{\text{x}}^2 + 2R_{\max}\varepsilon_{\text{x}}.$$

$\square$

**Proof of Corollary K.2.** When $\eta = 0$, Theorem K.1 states that

$$J(\pi^\star) - J(\widehat{\pi}) \leq 2\sqrt{\mathcal{C}^{\pi^\star}\varepsilon_{\text{stat}}^2} + 2\beta\mathcal{C}^{\pi^\star} + \frac{4\varepsilon_{\text{stat}}^2}{\beta} + 4\beta\varepsilon_{\text{x}}^2 \cdot \left(\mathcal{C}_\infty^{\pi^\star} + \max_{\pi \in \Pi}\mathcal{C}_\infty^{\pi}\right) + 2R_{\max}\varepsilon_{\text{x}}$$

Setting $\beta = 2\sqrt{\frac{\varepsilon_{\text{stat}}^2}{\mathcal{C}^{\pi^\star}}}$, we obtain

$$J(\pi^\star) - J(\widehat{\pi}) \leq 5\sqrt{\mathcal{C}^{\pi^\star}\varepsilon_{\text{stat}}^2} + 8\varepsilon_{\text{stat}}\varepsilon_{\text{x}}^2 \cdot \left(\mathcal{C}_\infty^{\pi^\star} + \max_{\pi \in \Pi}\mathcal{C}_\infty^{\pi}\right) + 2R_{\max}\varepsilon_{\text{x}}.$$

$\square$

**Proof of Lemma K.2.** We use similar reasoning and notation to the proof of Lemma H.1. Since $r^\star \in \mathcal{R}$ under Assumption C.1, Lemma F.1 guarantees that with probability at least $1 - \delta$ we have

$$\mathbb{E}_{\pi_{\text{ref}},\pi_{\text{ref}}}\left[D_{\mathsf{H}}^2(P_{\widehat{r}}(\cdot \mid x, a, b), P_{r^\star}(\cdot \mid x, a, b))\right] \leq \frac{2\log(|\mathcal{R}|/\delta)}{n}.$$

Since $|r(x, a) - r(x, b)| \leq R_{\max}$ for all $r \in \mathcal{R}$ under Assumption C.1, we then apply Lemma H.5 with $R = V = R_{\max}$.

$$\mathbb{E}_{\pi_{\text{ref}},\pi_{\text{ref}}}\left[(\widehat{r}(x, a) - \widehat{r}(x, b) - (r^\star(x, a) - r^\star(x, b)))^2\right]$$
$$\leq 16e^{4R_{\max}}R_{\max}^2 \cdot \mathbb{E}_{\pi_{\text{ref}},\pi_{\text{ref}}}\left[D_{\mathsf{H}}^2(P_{\widehat{r}}(\cdot \mid x, a, b), P_{r^\star}(\cdot \mid x, a, b))\right]$$
$$\leq 32e^{4R_{\max}}R_{\max}^2 \cdot \frac{\log(|\mathcal{R}|/\delta)}{n}.$$

$\square$

**Proof of Lemma K.3.** Abbreviate $\Delta^\star(x, a, b) = r^\star(x, a) - r^\star(x, b)$, and $\widehat{\Delta}(x, a, b) = \widehat{r}(x, a) - \widehat{r}(x, b)$. For a pair of policies $\pi, \pi'$ and $p \geq 1$, we define the norm $\|\cdot\|_{p, \pi \times \pi'} := (\mathbb{E}_{\rho, a \sim \pi, b \sim \pi'}[|\cdot|^p])^{1/p}$, so that $\mathbb{E}_{\pi,\pi_{\text{ref}}}\left[\left|\Delta^\star(x, a, b) - \widehat{\Delta}(x, a, b)\right|\right] = \left\|\Delta^\star - \widehat{\Delta}\right\|_{1, \pi \times \pi_{\text{ref}}}$. Then via Cauchy-Schwarz,

$$\left\|\Delta^\star - \widehat{\Delta}\right\|_{1, \pi \times \pi_{\text{ref}}} \leq \sqrt{\mathbb{E}_\rho\left[\sum_{a,b}\frac{\pi^2(a \mid x)\pi_{\text{ref}}^2(b \mid x)}{(\pi_{\text{ref}}(a \mid x) + \eta\pi(a \mid x))\pi_{\text{ref}}(b \mid x)}\right]}$$

$$\cdot \sqrt{\mathbb{E}_\rho\left[\sum_{a,b}(\pi_{\text{ref}}(a \mid x) + \eta\pi(a \mid x))\pi_{\text{ref}}(b \mid x)\left(\Delta^\star(x, a, b) - \widehat{\Delta}(x, a, b)\right)^2\right]}$$

$$= \sqrt{\mathcal{C}_\eta^{\pi} \cdot \left(\left\|\Delta^\star - \widehat{\Delta}\right\|_{2, \pi_{\text{ref}} \times \pi_{\text{ref}}}^2 + \eta\left\|\Delta^\star - \widehat{\Delta}\right\|_{2, \pi \times \pi_{\text{ref}}}^2\right)}$$

$$\leq \sqrt{\mathcal{C}_\eta^{\pi} \cdot \left\|\Delta^\star - \widehat{\Delta}\right\|_{2, \pi_{\text{ref}} \times \pi_{\text{ref}}}^2} + \sqrt{2\eta R_{\max}\mathcal{C}_\eta^{\pi} \cdot \left\|\Delta^\star - \widehat{\Delta}\right\|_{1, \pi \times \pi_{\text{ref}}}}.$$

Applying the AM-GM inequality to the second term, we obtain

$$\left\|\Delta^\star - \widehat{\Delta}\right\|_{1, \pi \times \pi_{\text{ref}}} \leq \sqrt{\mathcal{C}_\eta^{\pi} \cdot \left\|\Delta^\star - \widehat{\Delta}\right\|_{2, \pi_{\text{ref}} \times \pi_{\text{ref}}}^2} + \eta R_{\max}\mathcal{C}_\eta^{\pi} + \frac{1}{2}\left\|\Delta^\star - \widehat{\Delta}\right\|_{1, \pi \times \pi_{\text{ref}}}.$$

Rearranging,

$$\left\| \Delta^\star - \widehat{\Delta} \right\|_{1, \pi \times \pi_{\mathrm{ref}}} \leq 2 \sqrt{ \mathcal{C}_\eta^\pi \cdot \left\| \Delta^\star - \widehat{\Delta} \right\|_{2, \pi_{\mathrm{ref}} \times \pi_{\mathrm{ref}}}^2 } + 2\eta R_{\mathsf{max}} \mathcal{C}_\eta^\pi.$$

$\square$

