# OpenReview forum: "Correcting the Mythos of KL-Regularization: Direct Alignment without Overoptimization via Chi-Squared Preference Optimization"
_ICLR.cc/2025/Conference — ICLR 2025 Spotlight_

### Official Review · Reviewer_r2Qj · 2024-10-20

**Soundness:** 2
**Presentation:** 2
**Contribution:** 3
**Rating:** 6
**Confidence:** 3

**Summary:**

This work proposes a simple variant of DPO called $\chi$PO, by adding $\chi^2$-divergence regularizer to DPO to encourage pessimism, and proves that $\chi$PO has improved generalization error bound of DPO with all-policy concentrability replaced by single-policy concentrability.

**Strengths:**

Adding $\chi^2$ divergence is novel in DPO. Theorem 3.1 well supports the good generalization ability of $\chi$PO, and it also does not require convex set assumption in existing pessimistic approach called DPO+SFT. The presentation before Section 4 is clear.

**Weaknesses:**

Some points could be clarified as listed in the questions below. The experiments only demonstrate the advantage of $\chi$PO over DPO but not DPO+SFT.

**Questions:**

(1) $\widehat{\pi}$ is defined by different objective functions (7) and (9). However, it seems that only (9) is used for algorithm, Theorem 3.1 and analysis in Section 4. Is it more clear to only introduce (9) not (7), and state your novelty as adding $\chi^2$-divergence to the original DPO objective, not replacing the KL divergence?

(2) You could give the full name of "SFT" at its first appearance.

(3) At the end of Section 2.1, I think (4) is $-\infty$ if $\pi(a_+|x)\ll\pi _ {\rm ref}(a_+|x)$ for some data point $x,a_+$.

(4) In Eq. (5), is $n$ the size of $\mathcal{D}_{\rm off}$?

(5) Theorem H.1 (General version of Theorem 3.1) and Corollary 3.1 seem to use the same choice of $\beta$ relying on $\pi^*$. In this case, I am not sure why do we need Corollary 3.1?

(6) You could define $\mathcal{C}_{\infty}^{\pi}$ at its first appearance.

(7) At the beginning of Section 4.3, why can Assumption 3.2 about two-point difference imply single-point bound $\big\|\frac{\pi}{\pi _ {\rm ref}}\big\| _ {\infty} \lesssim \frac{V _ {\max}}{\beta}$ for all $\pi\in\Pi$?

(8) In line 507, should it be $\frac{\pi_{\beta;{\rm KL}}^{\star}(a \mid x)}{\pi_{\rm ref}(a \mid x)} \gtrsim \exp\big(-\frac{R_{\max }}{\beta}\big)$ (as you said in line 449)? If so, the optimal mixed $\chi^2$-regularized policy also has this lower bound and thus also cannot simultaneously satisfy the two properties you said in line 509 (change the second to $\exp\big(-\frac{R_{\max }}{\beta}\big)$).

(9) Is Section 4.3 necessary, as it seems to only give a looser bound than Theorem 3.1 which already shows the better generalization ability of $\chi$-PO over DPO by replacing $\max_{\pi\in\Pi}\mathcal{C}^{\pi}$ with $\mathcal{C}^{\pi^*}$? Also, I feel it will be more clear to conclude the benefits of $\chi$-PO over DPO, for example, at the beginning or end of Section 4 or its subsections.

(10) About experiments: The link function $\tilde{\phi}(z):=\exp \big(\operatorname{clip} _ {[-88,20]}(\alpha \cdot z)\big)+\gamma\log z$ looks far away from $\phi$ since $\exp \big(\operatorname{clip} _ {[-88,20]}(\alpha \cdot z)\big)$ is exponential in a large range $[-88,20]$ which is far away from $z$. Since you said $\tilde{\phi}(z)$ is has better stability and performance in practice, do you think we can obtain its generalization bound? It is also better to compare with existing pessimistic approaches such as DPO+SFT. Also, in Section E.2, we could use $\alpha\in${$\frac{1}{4},1$} and $\gamma\in${$0.1,1$}.

---

> ### Author Response · Authors · 2024-11-22
> **Response re: questions 1**
>
> Thank you very much for your careful attention to our work and interesting questions! In follow-up empirical work, we do intend to compare with more baselines like DPO+SFT (see joint response for more details).
>
> > $\widehat\pi$  is defined by different objective functions (7) and (9). However, it seems that only (9) is used for algorithm, Theorem 3.1 and analysis in Section 4. Is it more clear to only introduce (9) not (7), and state your novelty as adding χ2-divergence to the original DPO objective, not replacing the KL divergence?
>
> This is a slightly subtle point. We believe the idea of using chi-squared regularization for alignment in general is a novel conceptual contribution that goes beyond just comparisons to DPO. In particular, it can be used to implement pessimistic algorithms (with strong theoretical guarantees) in an efficient way that goes beyond just χPO, thanks to the structure of language models.
>
> For example, replacing KL divergence with chi-squared divergence entirely is sufficient to derive guarantees for an RLHF version of the algorithm (see Appendix C), and in fact the guarantees for χPO hold even if we use an arbitrarily small (but positive) coefficient for KL (which we show in Theorem H.1). Therefore, we believe that characterizing our novelty as “replacing” KL divergence is more accurate, but we are happy to include more discussion around this point in the final version of the paper.
>
> > You could give the full name of "SFT" at its first appearance.
>
> We will be sure to do this in the final version of the paper.
>
> > At the end of Section 2.1, I think (4) is −∞if  π(a+|x)≪πref(a+|x) for some data point x,a+.
>
> We will update this in the final version of the paper.
>
> > In Eq. (5), is $n$ the size of Doff?
>
> Yes.
>
> > Theorem H.1 (General version of Theorem 3.1) and Corollary 3.1 seem to use the same choice of β relying on π∗. In this case, I am not sure why do we need Corollary 3.1?
>
> Corollary 3.1 only serves to highlight the sample complexity guarantee that is achievable by optimal tuning. Indeed, it is an immediate consequence of Theorem H.1.
>
> > You could define C∞π at its first appearance.
>
> We will be sure to move the definition to the first appearance in the final version of the paper.

---

> ### Author Response · Authors · 2024-11-22
> **Response re: questions 2**
>
> > At the beginning of Section 4.3, why can Assumption 3.2 about two-point difference imply single-point bound $\|\pi / \piref\|_\infty \l V_{\max}/\beta$?
>
> We are happy to include a proof of this fact in the final version of the paper. Briefly, the argument is as follows. If we consider the tightest possible value for $V_{\max}$, the inequality in Assumption 3.2 will be tight for a pair of actions $(a,b)$ where $\pi(a) > \pi_{\mathsf{ref}}(a)$ but $b$ has $\pi(b) < \pi_{\mathsf{ref}}(b)$. This is because for action $b$, the $\log$ will dominate the $\phi(\pi(b) / \pi_{\mathsf{ref}}(b))$ term which will be negative; whilst the corresponding term for $a$ will be positive. Thus the two-point difference implies a single-point bound that $\phi(\pi(a) / \pi_{\mathsf{ref}}(a)) \le V_{\max}/\beta$, and because the policy ratio is larger than 1, this in turn implies the single-point bound (via Proposition B.1).
>
> > Is Section 4.3 necessary, as it seems to only give a looser bound than Theorem 3.1 which already shows the better generalization ability of χPO over DPO by replacing maxπ∈ΠCπ with Cπ∗? Also, I feel it will be more clear to conclude the benefits of χPO over DPO, for example, at the beginning or end of Section 4 or its subsections.
>
> Indeed, the discussion in Section 4.3 is not “strictly necessary” to show that χPO has better generalization guarantees than DPO, for which Theorem 3.1 is sufficient. Rather, Section 4.3 simply serves to build further intuition about how the policy learned by χPO (and, more generally, under $\chi^2$-regularization) compares to the one learned by DPO.
>
> Let us mention in passing that our broader goal in Section 4.3 was to make a point about the benefits of $\chi^2$-generalization in terms of the representational conditions it places on policy classes and on the learned policies, and it appears we can improve our presentation here to make the message more clear. We are happy to consider reorganizing our paper based on your feedback.
>
> > About experiments: The link function ϕ~(z):=exp⁡(clip[−88,20](α⋅z))+γlog⁡z looks far away from ϕ since exp⁡(clip[−88,20](α⋅z)) is exponential in a large range [−88,20] which is far away from z. Since you said $\tilde\phi(z)$ has better stability and performance in practice, do you think we can obtain its generalization bound? It is also better to compare with existing pessimistic approaches such as DPO+SFT.
>
> This is actually a typo, it is supposed to be $\widetilde\phi(z) := \exp⁡(\mathsf{clip}(\alpha\cdot\log(z)))+\gamma\log(⁡z)$. Without clipping, this link function corresponds to mixing KL-divergence with the so-called $\alpha$-divergence given by $D_{1+\alpha}(\pi, \pi_{\mathsf{ref}}) = c \cdot E_{\pi}[(\pi/\pi_{\mathsf{ref}})^{\alpha}] + c’$, which is a generalization of the $\chi^2$-divergence that is equivalent to setting $\alpha = 1$.
>
> It is very straightforward to generalize our statistical analysis to handle general values of $\alpha > 0$ (with a slightly different rate), and we are happy to include this in the final version of the paper.

---

> > ### Comment · Reviewer_r2Qj · 2024-11-22
> > **I'm satisfied with the answers and will keep rating 6.**
> >
> > I'm satisfied with the answers.
> >
> > Since DPO+SFT is not included in the experiment, I will keep rating 6.
> >
> > Based on your answer to Q1, you may consider different notations of $\widehat{\pi}$ for Eqs. (7) and (9).

---

### Official Review · Reviewer_rd5w · 2024-11-01

**Soundness:** 3
**Presentation:** 3
**Contribution:** 3
**Rating:** 6
**Confidence:** 3

**Summary:**

This paper introduces Chi-Squared Preference Optimization ($\chi$PO) as an alternative to KL-based regularization in preference alignment. $\chi$PO replaces the KL divergence with Chi-Squared divergence, leveraging its ability to better quantify uncertainty and provide stronger, more reliable regularization against overoptimization. The authors demonstrate that $\chi$PO aligns language models more effectively by enhancing sample efficiency and robustness, grounded in single-policy concentrability—a key offline reinforcement learning standard. This theoretical shift offers a novel perspective on controlling overoptimization, simplifying the DPO framework with a mathematically sound alternative that generalizes well across alignment tasks.

**Strengths:**

1. $\chi$PO provides stronger regularization than KL-divergence by better accounting for uncertainty, thus significantly reducing the risk of overoptimization when aligning language models with human preferences.

2. $\chi$PO achieves sample-complexity guarantees through single-policy concentrability, meaning it requires fewer samples to achieve reliable performance, making it particularly effective in offline alignment tasks.

3. The approach requires minimal modifications to the DPO framework, making it both practical and easy to implement in existing systems.

4. By grounding $\chi$PO in well-established principles from offline reinforcement learning, the authors provide a robust theoretical foundation that enhances the credibility and potential generalizability of the method.

**Weaknesses:**

1. Limited empirical validation: The paper only provide theoretical insights, not sure about the gap between the analysis and the real practice.

2. The gap between theoretical/implementable algorithm: the additional KL term for KKT.

**Questions:**

1. In order to obtain the practical algorithm, the authors added the KL term, do you have more explanation about the gap and more intuitions to deliver the implementable algorithm?

2. Have you tried to perform empirical experiments?

---

> ### Author Response · Authors · 2024-11-22
> **Response re: weaknesses**
>
> > Limited empirical validation: The paper only provide theoretical insights, not sure about the gap between the analysis and the real practice
>
> Preliminary experiments are included in Appendix E. Please see also our joint response for more details.
>
> > The gap between theoretical/implementable algorithm: the additional KL term for KKT.
>
> We implement the algorithm (including KL), so respectfully disagree that the KL-regularization creates a gap between the theoretical and implementable algorithm.

---

> ### Author Response · Authors · 2024-11-22
> **Response re: questions**
>
> > Have you tried to perform empirical experiments?
>
> Yes, we provide an initial proof-of-concept experiment in Appendix E. Our experiments in Appendix implement χPO (Algorithm 1) the TL;DR summarization task, which has been used frequently to evaluate RLHF methods. Figure 4 shows that, compared to DPO, χPO does not exhibit severe decrease in win-rate over training, and mitigates deviation from $\pi_{\mathsf{ref}}$ much more effectively.
>
> We do plan to perform a more extensive empirical evaluation of χPO, but we view this as follow-up work to this theoretical investigation. Indeed, we would like to briefly emphasize that this is a theoretical paper submitted to the learning theory category. The main result in the paper is Theorem 3.1, which is the first theoretical sample complexity result which offers a provably efficient direct alignment algorithm that converges to an optimal LLM policy under single-policy concentrability (with minimal additional assumptions). We believe this result on its own represents a significant advance in the theory of reinforcement learning, and is sufficient to merit acceptance.
>
> > Limited empirical validation & Gap between analysis and real practice…Gap between theoretical/implementable algorithm
>
> We emphasize that there is no gap between the algorithm we analyze and the algorithm we use in our experiments. The χPO algorithm (Algorithm 1)---including KL divergence—is exactly the algorithm that we show obtains single-policy concentrability in Theorem 3.1 (our main theoretical result), and is exactly the algorithm that is used for the empirical results in Appendix E (namely, Figure 4).
>
> > In order to obtain the practical algorithm, the authors added the KL term, do you have more explanation about the gap and more intuitions to deliver the implementable algorithm?
>
> We emphasize that there is no gap between the algorithm we analyze and the algorithm we use in our experiments; please refer to the response to the question above.

---

> > ### Comment · Reviewer_rd5w · 2024-11-24
> >
> > Thanks for the feedback.
> >
> > In general, this paper provides some very interesting and nice theories. The empirical experiments are preliminary but still helpful to readers. As I already provided a positive score, I will maintain my evaluation.

---

### Official Review · Reviewer_6mx8 · 2024-11-01

**Soundness:** 3
**Presentation:** 3
**Contribution:** 2
**Rating:** 6
**Confidence:** 3

**Summary:**

This paper studies overoptimization problem in RLHF where language models trend to overfit to the learned reward model which often results in degraded performance. The authors analyze the weaknesses of commonly used KL regularization from RL theoretic perspectives and claim its insufficiency to prevent overoptimization.
They propose to replace KL regularization with $\chi^2$-regularization to achieve theoretical guarantees on single-policy concentrability. Finally, the authors derive a variant of offline alignment objective function using $\chi^2$-regularization with minimal change to DPO.

**Strengths:**

- The proposed $\chi$PO algorithm and its theoretical analysis are novel contributions to alleviate overoptimization issue for offline alignment.
- $\chi$PO has better provable robustness to overoptimization and well-supported by empirical experiments.
- The theoretical analysis of $\chi$PO and its properties is sound and comprehensive.

**Weaknesses:**

- In addition to the comprehensive and rigorous analysis, it could be great to support the claim with more empirical experiments. For example, it would be nice to add additional benchmarks (e.g., MMLU, GSM8K) against common baselines (e.g. DPO, KPO) for empirical validations.
- As offline alignment has recently been rapidly growing, the empirical experiments could include a few more latest baselines (e.g. DPO, KPO, IPO, ...) with ablations over key hyperparameters such as $\beta$.
- As a comment: I think it would be nice to have experiments in Appendix E.2 included in main text more extensively for broader audience.

**Questions:**

- Although the paper explains the necessity from theoretical perspective to include KL term in mixed regularization, what would be the practical implications to have only $\chi^2$ regularization? How about a comparison among PPO-Mix (paper), PPO-KL (standard RLHF), and PPO-Chi for online setting to validate how well the theory and practice are matching.
- What would be training dynamics between two divergence regularizers? How separate coefficient would affect the interaction between KL and $\chi^2$ regularizers?
- More broadly, in standard RL policy optimization, PPO-Clip objective also does not have log-term. Could there be a meaningful link between $\chi^2$ regularization in general policy optimization? e.g. utilize $\chi^2$-divergence to constraint trust region during policy optimization.

---

> ### Author Response · Authors · 2024-11-22
> **Response re: weaknesses**
>
> We thank the reviewer for their careful attention to our work!
>
> > More benchmarks and baselines
>
> We do plan to perform a more extensive empirical evaluation of χPO, but we view this as follow-up work to this theoretical investigation. Indeed, we would like to emphasize that this is a theoretical paper submitted to the learning theory category. While we included encouraging initial empirical results that are consistent with our analysis, the main result in the paper is Theorem 3.1, which is the first theoretical sample complexity result which offers a provably efficient direct alignment algorithm that converges to an optimal LLM policy under single-policy concentrability (with minimal additional assumptions). We believe this result on its own represents a significant advance in the theory of reinforcement learning, and is sufficient to merit acceptance.
>
> That said, we agree that evaluation with more extensive benchmarks, baselines, and ablations are important, and we are excited to pursue this direction in follow-up work. Please also see our joint response also for more details, and we would be happy to re-organize our paper based on your comments.

---

> ### Author Response · Authors · 2024-11-22
> **Response re: questions**
>
> > Although the paper explains the necessity from theoretical perspective to include KL term in mixed regularization, what would be the practical implications to have only χ2 regularization?
>
> We briefly tried using smaller regularization parameters for the KL-divergence (and our theory handles this, see Appendix H). So far it seems that using regularization parameters of similar magnitude for KL and $\chi^2$ works best, and we see no real need to remove the KL-divergence yet. Overall, this is an interesting question that we are indeed investigating this further.
>
> Thank you for your suggestions of baselines, we agree that this comparison would be interesting and important.
>
> > What would be training dynamics between two divergence regularizers? How separate coefficient would affect the interaction between KL and χ2 regularizers?
>
> In general, the KL and $\chi^2$ regularizers play different roles, so we are not concerned with different coefficients causing poor interactions between the two terms; this is accounted for in our theoretical guarantees for χPO (e.g., Theorem 3.1). The $\chi^2$-divergence dominates the KL in terms of keeping $\pi$ close to $\pi_{\mathsf{ref}}$, and this effect can be seen empirically in Figure 4 (right), where χPO controls the deviation (as measured by KL-divergence) more strongly.  The primary purpose of the KL term is to keep $\pi > 0$ where $\pi_{\mathsf{ref}} > 0$, which enables the reward-to-policy substitution central to direct policy optimization methods (as described in Section 3.2).
>
> For our empirical results in Appendix E, we experimented with separate coefficients and found that, over a magnitude, a variety of regularization parameters work reasonably well (see Table 1).
>
> > More broadly, in standard RL policy optimization, PPO-Clip objective also does not have log-term. Could there be a meaningful link between χ2 regularization in general policy optimization? e.g. utilize χ2-divergence to constraint trust region during policy optimization.
>
> This is a very interesting question. In fact, in ongoing work, we are able to show, using technical arguments similar to those in the proof for χPO, that a clipping objective inspired by PPO can provably implement pessimism in a similar fashion to χPO, albeit with a slightly worse rate compared to Theorem 3.1. We plan to explore this more deeply in followup work.

---

### Official Review · Reviewer_Zwhj · 2024-11-02

**Soundness:** 4
**Presentation:** 3
**Contribution:** 4
**Rating:** 8
**Confidence:** 3

**Summary:**

Existing offline alignment algorithms based on KL regularization, such as direct preference optimization (DPO) and DPO+SFT, suffer from reward overoptimization in experiments.
This work aims to address this issue.
The authors first observed that reward overoptimization is related to the dependence of the regret bound on the coverage coefficient.
To address this, they proposed $\chi$PO, a variant of DPO based on $\chi^2$ regularization.
This algorithm is the first general method with a sample complexity bound of $O(\sqrt{C^{\pi^\star}/n})$, where $C^{\pi^\star}$ is the coverage coefficient, $n$ is the sample size, and $\pi^\star$ is the policy to which we wish to compare.
This bound can be significantly smaller than existing bounds that depend on $\max_{\pi}C^{\pi}$.
Numerical experiments demonstrate the robustness of $\chi$PO against overoptimization.

**Strengths:**

I have read the main paper, Appendix A, B, C, E, F, and have skimmed through Appendix G and H.

This work is solid and generally well-written.
Related works are adequately cited.
Theoretical results are sound.
It is surprising to me that a small tweak to DPO can significantly improve the theoretical guarantee and the empirical results.
I appreciate this work.

**Weaknesses:**

I did not spot any major weaknesses. Below are some minor issues related to clarity.

1. In Appendix E.2, it is mentioned that $\chi$PO was not used in the experiments because it may be unstable during training. Therefore, another link function was used. I suggest mentioning this in the captions (Table 1, Figure 4, Table 2) to avoid misunderstanding and, if possible, presenting the experimental results of $\chi$PO.
2. In line 131, it should be $x \in \mathcal{X}$.
3. In line 138, it should be $y \sim \mathbb{P}(a \succ b | x)$.

**Questions:**

1. The explanation of the non-triviality in Section 4.3 is a bit confusing to me. My understanding is that we can use the all-policy concentrability and Assumption 3.2 to obtain a sample complexity guarantee for $\chi$PO. Nevertheless, this does not yield a guarantee based on the single-policy concentrability, so the sample complexity guarantee of $\chi$PO (Theorem 3.1) is non-trivial. Is my understanding correct?
2. The sample complexity guarantee of $\chi$PO (Theorem 3.1) has an exponential dependence on the maximal reward $R_\max$. The algorithm of Xie et al. (2024) also has this exponential dependence. Is this exponential dependence unavoidable, or can it be improved? Also, is this exponential dependence observed in experiments?
3. The $\chi$PO algorithm uses a regularizer that combines the $\chi^2$ divergence and the KL divergence. In Theorem H.1 (a general version of Theorem 3.1), it is mentioned that we can make the weight of the KL divergence arbitrarily close to $0$ while still maintaining the sample complexity guarantee. As the weight of the KL divergence decreases to $0$, does the empirical performance of $\chi$PO improve?
4. Following question 3, I wonder if the algorithm and the proof would still hold if we removed the KL divergence. Furthermore, does this suggest that the technical assumption $0 \notin \text{dom}(f')$ in Wang et al. (2023) might be removable?

References:
- T. Xie et al. Exploratory preference optimization: Harnessing implicit Q*-approximation for sample-efficient RLHF. *arXiv*, 2024.
- C. Wang et al. Beyond reverse KL: Generalizing direct preference optimization with diverse divergence constraints. *arXiv*, 2023.

---

> ### Author Response · Authors · 2024-11-22
> **Response re: weaknesses**
>
> Thank you for your insightful comments and appreciation of our work!
>
> > In Appendix E.2, it is mentioned that χPO was not used in the experiments because it may be unstable during training. Therefore, another link function was used. I suggest mentioning this in the captions (Table 1, Figure 4, Table 2) to avoid misunderstanding and, if possible, presenting the experimental results of χPO.
>
> To clarify, we did use χPO (with the link function from Algorithm 1) for the experiments displayed in Figure 4, which shows that its winrate does not decrease over training and KL-divergence is much better controlled compared to DPO. For Table 2, we experimented with a slightly more general link function corresponds to the so-called alpha-divergence given by $D_{\alpha}(\pi, \pi_{\mathsf{ref}}) = c \cdot E_{\pi}{}[(\pi / \pi_{\mathsf{ref}})^{\alpha-1}] + c'$ for different values of $\alpha$, which is a slight generalization of the chi-squared divergence (the special case where $\alpha=2$).
>
> We will make this more clear in the caption. Also let us mention that it is straightforward to generalize our statistical analysis to handle general values of $\alpha > 1$ (with a slightly different rate), and we are happy to include this in the final version of the paper.

---

> ### Author Response · Authors · 2024-11-22
> **Response re: questions 1**
>
> > Explanation of non-triviality in Section 4.3
>
> Yes, your understanding is correct, and it results in a bound that is looser than the one we establish in Theorem 3.1 (as you mentioned).
>
> In addition, Section 4.3 shows that the same argument applied to DPO yields a substantially worse conclusion. DPO requires a similar assumption to Assumption 3.2 (bounded implicit rewards) as a consequence of the reward-to-policy substitution. However, the same computation reveals that the induced all-policy concentrability is exponentially larger (scaling with $\exp(R_{\max}/\beta)$) compared to that of $\chi\mathsf{PO}$ (which scales with $R_{\max}/\beta$).
>
> > The sample complexity guarantee of χPO (Theorem 3.1) has an exponential dependence on the maximal reward Rmax. The algorithm of Xie et al. (2024) also has this exponential dependence. Is this exponential dependence unavoidable, or can it be improved? Also, is this exponential dependence observed in experiments?
>
> This dependence is intrinsic and results from the sigmoid function in the Bradley-Terry model. We typically treat $R_{\max} = O(1)$, so even its exponential should not be the dominating term in the bound. For example, publicly available reward models for the TL;DR task [2] fall in the range [-5, 5].  We emphasize that dependence on this parameter is found in all prior work on RLHF under the Bradley-Terry model [3,4,5] (including for vanilla DPO itself), as well as throughout the literature on the closely related problem of logistic bandits (e.g., [6,7]), where it is known to be unavoidable. We would be happy to include a simple lower bound showing that it is information-theoretically necessary in the RLHF setting as well.
>
> The underlying intuition for the exponential in Rmax is as follows: Suppose we compare a pair of reward values $r_1$  and $r_2$ through the sigmoid of their difference with some baseline value $r_0$; that is, we measure $r_1−r_2$  (usually not available for direct measuring) using $\sigma(r_1−r_0)−\sigma(r_2−r_0)$ as a proxy. In this case, the difference between $r_1$ and $r_2$ becomes exponentially difficult to detect as both $r_1−r_0$ and $r_2−r_0$ grow larger. In the worst-case scenario, this can be  $|r_1−r_2|\approx\exp⁡(R_{\max})|\sigma(r_1−r_0)−\sigma(r_2−r_0)|$.
>
> Let us mention in passing that if we consider the classical RL setting where rewards are observed directly, then by replacing the maximum likelihood loss with a square loss (e.g., as in [1], we can remove the $\exp(R_{\max})$ dependence.
>
> Regarding experiments: It is not obvious to us how to evaluate the exponential dependence directly in experiments since we use real preference data (i.e., there is no “true” underlying reward model that we can observe), but, even so, there is no reason to expect that our method show suffer more than DPO or any other direct alignment method.
>
> **References**
>
> [1] Gao, Z., Chang, J. D., Zhan, W., Oertell, O., Swamy, G., Brantley, K., ... & Sun, W. (2024). Rebel: Reinforcement learning via regressing relative rewards. arXiv preprint arXiv:2404.16767.
>
> [2] https://huggingface.co/cleanrl/EleutherAI_pythia-1b-deduped__reward__tldr
>
> [3] Liu, Z., Lu, M., Zhang, S., Liu, B., Guo, H., Yang, Y., ... & Wang, Z. (2024). Provably mitigating overoptimization in rlhf: Your sft loss is implicitly an adversarial regularizer. arXiv preprint arXiv:2405.16436.
>
> [4] Xie, T., Foster, D. J., Krishnamurthy, A., Rosset, C., Awadallah, A., & Rakhlin, A. (2024). Exploratory Preference Optimization: Harnessing Implicit Q*-Approximation for Sample-Efficient RLHF. arXiv preprint arXiv:2405.21046.
>
> [5] Ji, X., Kulkarni, S., Wang, M., & Xie, T. (2024). Self-Play with Adversarial Critic: Provable and Scalable Offline Alignment for Language Models. arXiv preprint arXiv:2406.04274.
>
> [6] Faury, L., Abeille, M., Calauzènes, C., & Fercoq, O. (2020, November). Improved optimistic algorithms for logistic bandits. In International Conference on Machine Learning (pp. 3052-3060). PMLR.
>
> [7] Liu, S., Ayoub, A., Sentenac, F., Tan, X., & Szepesvári, C. (2024). Almost Free: Self-concordance in Natural Exponential Families and an Application to Bandits. arXiv preprint arXiv:2410.01112.

---

> ### Author Response · Authors · 2024-11-22
> **Response re: questions 2**
>
> > The χPO algorithm uses a regularizer that combines the χ2 divergence and the KL divergence. In Theorem H.1 (a general version of Theorem 3.1), it is mentioned that we can make the weight of the KL divergence arbitrarily close to 0 while still maintaining the sample complexity guarantee. As the weight of the KL divergence decreases to 0, does the empirical performance of χPO improve?
>
> So far, we have found that χPO is more stable with KL divergence when the regularization parameters have a similar magnitude, although we did not do extensive ablations. We hypothesize that this is because KL regularization implicitly leads to better conditioning of the χPO loss through the magnitude of the parameter $V_{\max}$ in Assumption 3.2. It would be interesting to perform a more extensive study of this phenomenon in future work.
>
> > Following question 3, I wonder if the algorithm and the proof would still hold if we removed the KL divergence. Furthermore, does this suggest that the technical assumption 0∉dom(f′) in Wang et al. (2023) might be removable?
>
> Our sample complexity results still hold when the regularization parameter multiplying the KL-divergence is an arbitrarily small constant, but not necessarily when the KL-divergence is removed.  In particular, Appendix H analyzes a more general version of χPO, where the KL divergence is modulated by a different regularization parameter $\beta \gamma$, with $\gamma$ acting as a multiplier. Theorem H.1 shows that this algorithm achieves single-policy concentrability as long as $\gamma \in (0,1]$ (or $\beta\gamma \in (0, \beta]$), and some $\gamma = O(1)$ that’s larger than 1 also works.
>
> Obtaining single-policy concentrability when $\gamma = 0$ does not immediately follow from a limiting argument because the realizability assumption (Assumption H.1) for the optimal policy under mixed regularization depends on $\gamma$. That is, we can not simply take $\gamma \rightarrow 0$ while keeping the class $\Pi$ fixed.
>
> We have thought for some time about whether it is possible to remove the technical assumption from Wang et al (2023), but we now believe the KL regularization plays an important role as a “barrier” function that ensures the optimal regularized policy lies in the interior of the simplex. This enables the reward-to-policy substitution in direct policy optimization methods by ensuring that all action-dependent dual variables are 0 in the complementary slackness KKT condition.

---

> ### Comment · Reviewer_Zwhj · 2024-11-26
>
> Thank you for the detailed response. I have raised my confidence to 3.
>
> I appreciate the explanation of the exponential dependence on $R_{\max}$. Additionally, I would suggest not including more theoretical results, as they may not receive further review.

---

### Official Review · Reviewer_nK5T · 2024-11-08

**Soundness:** 3
**Presentation:** 3
**Contribution:** 2
**Rating:** 6
**Confidence:** 3

**Summary:**

This work focuses on the issue of over-optimization in RLHF, where over-optimization refers to the issue of overfitting of language model to an imperfect offline reward model. Authors use the principle of pessimism in reinforcement learning to tackle the over-optimization, and they propose to replace the KL divergence-based regularization in RLHF with the more conservative $\chi^2$ divergence. A single policy concentration convergence result is derived to show the sample-efficiency of the proposed algorithm and a summarization task is used for numerical validation.

**Strengths:**

1. Over-optimization to offline reward model has drawn a lot of attention recently and deserves more research attention.
2. The proposed algorithm is simple to implement with a one-line change compared to the classical RLHF/DPO
3. Convergence results in terms of single policy concentration are provided and detailed comparison with DPO in terms of bias/over-optimization trade-off is provided.

**Weaknesses:**

Empirical evaluation is the main weakness of the paper. Authors only compare with one task (the summarization task) and one baseline method (DPO). Considering the recent large body of work studying the topic, it is desirable to compare with more baseline methods such as the 'DPO+SFT' approach. Furthermore,  the improvement of XPO compared to DPO is not significant (roughly one point when comparing the best results of the two) and XPO still suffers from over-optimization when the training epochs are increased. Finally, can you show the plots of win-logps vs epochs? Does it increase or decrease as training goes.

**Questions:**

Can you elaborate more in terms why the mix $\chi^2$ regularization obtains single-policy concentration while the KL regularization not? What roles the pessimism plays in the proof?

---

> ### Author Response · Authors · 2024-11-22
> **Response re: weaknesses**
>
> > Empirical evaluation is the main weakness of the paper.
>  Authors only compare with one task (the summarization task) and one baseline method (DPO).
> Considering the recent large body of work studying the topic, it is desirable to compare with more baseline methods such as the 'DPO+SFT' approach.
>
> We would like to emphasize that this is a theoretical paper submitted to the learning theory category. While we included encouraging initial empirical results that are consistent with our analysis, the main result in the paper is Theorem 3.1, which is the first theoretical sample complexity result that offers a provably-efficient direct alignment algorithm, which converges to an optimal LLM policy under single-policy concentrability (with minimal additional assumptions). We believe this result on its own represents a significant advance in the theory of reinforcement learning, and is sufficient to merit acceptance.
>
> That said, we agree that evaluation with more extensive benchmarks, baselines (e.g., the DPO+SFT method), and ablations are important, and we are excited to pursue this direction in follow-up work. Please see our joint response also for more details.
>
> > Improvement is not significant
>
> We believe that the small improvement in overall performance, in combination with the robustness over different training epochs and regularization parameters, is a promising early result that is consistent with our theoretical analysis. It is possible that more interesting trends and separations will arise after evaluating χPO on more tasks and parameters (e.g., clipping variations, different KL and $\chi^2$ regularization).
>
> > $\chi\mathsf{PO}$ still suffers from over-optimization when the training epochs are increased
>
> We are not exactly sure which empirical observation this refers to, since Figure 4 shows that χPO performance gradually improves over the course of 2 epochs of training, and the KL-divergence is significantly better controlled compared to DPO.
>
> The reviewer may be referring to the fact that, in Table 1, the χPO winrate is lower after 4 epochs of training than 1 or 2? Here, we find that while χPO has a slight decrease in win-rate, the decrease is far smaller than that of DPO. Further, χPO exhibits strong robustness over the number of training epochs (and regularization parameter choice) where DPO has virtually none. Notably, from 1 to 4 training epochs, for $\beta = 0.05$, χPO drops 6% while DPO drops 14%, and χPO hardly changes in performance for $\beta = 0.005$, whereas DPO performance is virtually destroyed. We argue that this is one way in which the mitigation of over-optimization manifests empirically.
>
> > Finally, can you show the plots of win-logps vs epochs? Does it increase or decrease as training goes.
>
> We did not evaluate the model’s log-probabilities on chosen responses for the validation preference dataset, and this is not something that we’re able to produce during the discussion period due to limited access to computational resources.
>
> However, during training we did record the chosen response log-ratio $\pi(a_+\mid{}x) / \pi_{\mathsf{ref}}(a_+\mid{}x)$, a related quantity that may help answer your question. We have uploaded a screenshot anonymously here: https://imgur.com/a/mf7U372. We observe that the log-ratio of the chosen responses for χPO are frequently larger than 1 and significantly larger than those from DPO; they rebound after an initial decrease (while the probabilities for DPO continue decreasing).
>
> We are happy to evaluate this more extensively and include the figure in the future.

---

> ### Author Response · Authors · 2024-11-22
> **Response re: questions**
>
> > What roles the pessimism plays in the proof?
>
> At a high level, pessimism prevents the learned policy from overfitting to an (implicit or explicit) reward model learned from offline data, and it does this by preventing the algorithm from erroneously over-estimating the reward for responses that are poorly-represented by the offline preference dataset.
>
> In the proof, this manifests itself through the inequality (II) that appears on line 1908, where we bound the error of the algorithm in terms of the estimation error for the (implicit reward) model. To relate these terms, a non-pessimistic algorithm would incur dependence on two coverage coefficients, $C_{\pi^\star}$ and $C_{\widehat\pi}$; the latter is an algorithm-dependent quantity and can be as large as $\max_{\pi}C_{\pi}$ in general (e.g., this occurs in Proposition A.1, illustrated in Figure 1). We show that the chi-squared regularizer, through pessimism, cancels off the latter quantity, leaving us with only $C_{\pi^\star}$ as desired.
>
> Please refer to the proof sketch in Appendix G for a more detailed overview.
>
> > Can you elaborate more in terms why the mix χ2 regularization obtains single-policy concentration while the KL regularization not?
>
> The chi-squared regularization term serves to cancel/negate errors that can arise from erroneously over-estimating the true reward function. This is because the chi-squared divergence $D_{\chi^2}(\pi, \pi_\mathsf{ref})$ has the unique statistical property that it bounds the mismatch between expected error of a reward model under $\pi$ and the expected error under $\pi_\mathsf{ref}$; see Lemma G.1 for a formal statement. This means that chi-squared divergence meaningfully bounds the “uncertainty” in the reward model, allowing us to formalize the argument that chi-squared regularization implements pessimism. For KL-divergence, an analogous inequality to Lemma G.1 does not hold in general without paying for an exponential loss in accuracy, which is in fact tight (see Proposition A.1, illustrated in Figure 1).

---

### Author Response · Authors · 2024-11-22
**General response to all reviewers**

We would like to thank all reviewers for their comments, and we look forward to improving our paper based on the feedback we have received. We are glad to see a general appreciation for the simplicity and implementability of our algorithm, in combination with the strength of its theoretical guarantees.

A common limitation mentioned by reviewers was the limited empirical evaluation of χPO, and we agree that this is an important direction that we intend to pursue in future work.

That said, we would like to emphasize that our submission is a theoretical paper submitted to the learning theory category. While the appendix includes initial empirical results that are consistent with our analysis, our main contributions are the following theoretical results:
- $\chi^2$-regularization obtains strong guarantees through pessimism and can be implemented efficiently in LLMs (Theorem 3.1)
- KL-regularized algorithms are provably inefficient statistically (Proposition A.1)

We believe these results represent a significant advance in the theory of reinforcement learning and will be of interest to the community, and thus are sufficient to merit acceptance on their own.

The preliminary evaluations of χPO on the TL;DR summarization task in Appendix E are encouraging, and demonstrate consistency with our theoretical predictions. Reflecting the above, they also raise interesting follow-up questions surrounding the effects of optimization and statistical error in offline alignment methods. We are actively analyzing χPO, as well as other pessimistic algorithms inspired by the techniques within this paper, using more tasks and baselines, which we hope to release as a follow-up work in the near future.

---

### Meta-Review · Area_Chair_VjeK · 2024-12-17

**Metareview:**

The reviewers unanimously appreciate the intellectual contribution of the work. My personal assessment is that the linkages between pessimism, overparameterization, and more standard notions of trust region-based regularization of KL are novel, and will be impactful to the emerging area of AI alignment. Therefore, I recommend this work be accepted.

**Additional Comments On Reviewer Discussion:**

NA

---

### Decision · Program_Chairs · 2025-01-22

Accept (Spotlight)